# KOR-BENCH: BENCHMARKING LANGUAGE MODELS ON KNOWLEDGE-ORTHOGONAL REASONING TASKS

**Kaijing Ma**[1,5*], **Xinrun Du**[1,3*], **Yunran Wang**[6*], **Haoran Zhang**[1,7], **Zhoufutu Wen**[1], **Xingwei Qu**[1,8],
**Jian Yang**[1], **Jiaheng Liu**[1,9], **Minghao Liu**[1,4], **Xiang Yue**[1,10], **Wenhao Huang**[1 2 3†], **Ge Zhang**[1 2 3†]

[1]Multimodal Art Projection Research Community, [2]ByteDance.Inc, [3]01.AI, [4]2077.AI,
[5]Tongji University, [6]École Polytechnique, [7]University of Illinois at Urbana-Champaign,
[8]University of Manchester, [9]Nanjing University, [10]Carnegie Mellon University

`mkj3085003@gmail.com, duxinrun2000@gmail.com, gezhang@umich.edu`

`https://kor-bench.github.io/`

## ABSTRACT

In this paper, we introduce KNOWLEDGE-ORTHOGONAL REASONING (KOR), a concept aimed at minimizing reliance on domain-specific knowledge, enabling more accurate evaluation of models' reasoning abilities in out-of-distribution settings. Based on this concept, we propose the KNOWLEDGE-ORTHOGONAL REASONING BENCHMARK (KOR-BENCH), encompassing five task categories: Operation, Logic, Cipher, Puzzle, and Counterfactual. KOR-Bench emphasizes models' effectiveness in applying new rule descriptions to solve novel rule-driven questions. O1-Preview and O1-Mini achieve accuracies of 72.88% and 70.16%, surpassing Claude-3.5-Sonnet and GPT-4o (58.96% and 58.00%), highlighting the effectiveness of KOR-Bench. We perform detailed analyses, identifying bottlenecks in the Cipher task with Stepwise Prompting, where two rounds of Self-Correction yield optimal results. We evaluate performance across three integrated tasks, explore the impact of Tricks on the Puzzle task, and visualize rule-focused attention. Additionally, we conduct an ablation study on dataset size, benchmark correlations, and zero-shot and three-shot "only questions" experiments. KOR-Bench aims to enhance reasoning evaluation and support further research in this area.

## 1 INTRODUCTION

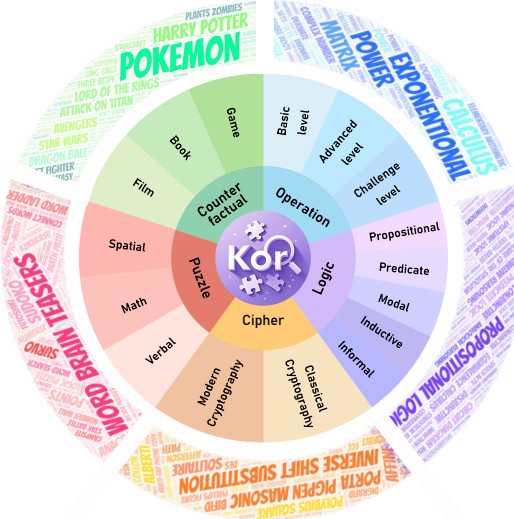

Figure 1: Overview of KOR-Bench.

Reasoning is a fundamental aspect of human intelligence, and research indicates that when models reach a sufficient scale, they exhibit emergent behaviors—including advanced reasoning capabilities such as understanding complex scenarios, strategic planning, and multi-step execution, making this capability a crucial indicator of an intelligent system's ability to handle complex tasks (Huang & Chang, 2022; Gui et al., 2024).

When learning new tasks and solving new problems, humans are never "starting from scratch"; rather, they are "nearly starting from scratch". This phenomenon is evident in various scenarios: by understanding game rules, humans can quickly master the gameplay (Nam & McClelland, 2024); by learning the basic rules of addition, humans can easily solve the problem of adding two numbers of any length (Hu et al., 2024); by giving restrictions and constraints, humans can apply thoughtful methods such as

---

* Equal Technical Contributions.
† Corresponding Authors.

*Reductio ad absurdum* and *Elimination* to solve puzzles (Bill Yuchen Lin, 2024). Human society is abundant with OOD (Out-of-Distribution) tasks (Liu et al., 2021b)—those that are novel and undefined—requiring continuous adaptation and the ability to navigate new paradigms. Humans have abilities like abstract, rule-based, and explanatory reasoning, enabling them to learn rules efficiently and adapt quickly to specific areas.

Similarly, we expect models to develop similar capabilities so that they can still effectively handle OOD tasks when encountering unfamiliar rules and frameworks, and generate results that conform to specific rules or settings in real-world applications (Sun et al., 2024). Despite the models' remarkable achievements on certain reasoning tasks, the study Mondorf & Plank (2024) points out that they are still challenged by conceptual errors and limitations when dealing with scenarios beyond the training data. While the incorporation of large amounts of code and data during model training improves the performance of a given task, this improvement is based more on the model's memory of the patterns of the training data than on its increased ability to follow rules or reason. This reliance on in-domain knowledge limits the effectiveness of existing evaluation benchmarks in accurately measuring a model's reasoning ability (Wu et al., 2023; Zhang et al., 2023; Dziri et al., 2023). Therefore, there is an urgent need to establish more comprehensive and effective evaluation benchmark to measure the ability of models to understand, follow new rules and solve problems efficiently, while reducing the reliance on pre-trained knowledge.

Inspired by a deeper understanding of the human learning process, we propose the concept of "Knowledge-Orthogonal Reasoning" (KOR) to explore a model's capabilities in reading comprehension, immediate learning, knowledge transfer, logical reasoning, and problem-solving, while reducing the reliance on the existing knowledge base. Knowledge Orthogonality, formally defined in Appendix A, refers to the independence between background/domain-specific knowledge (e.g., general knowledge or skills acquired during pre-training) and the rules explicitly defined to solve a particular task. It ensures that task-solving relies on understanding and reasoning about the task rules, while background knowledge only aids the reasoning process. "Knowledge-Orthogonal Reasoning Benchmark" (KOR-Bench) focuses on evaluating how models apply newly-defined rules to solve new rule-driven questions, rather than relying on data retrieval or information memorization.

Specifically, we design a series of tasks to challenge and demonstrate the model's reasoning ability by introducing new elements and rules. These tasks are divided into five categories, each based on one of the following new elements: new symbols, new concepts, new execution rules, new problem-solving frameworks, and new story-context settings. The specific categories are as follows:

- **Operation Reasoning Task**: Understand new definitions of mathematical symbols and apply this knowledge to perform calculations in mathematical reasoning tasks.
- **Logic Reasoning Task**: Reason and solve problems based on new logical rules and newly categorized logical concepts in logical reasoning tasks.
- **Cipher Reasoning Task**: Perform encryption and decryption operations according to new execution rules in cryptography reasoning tasks.
- **Puzzle Reasoning Task**: Solve puzzles and intellectual games based on newly defined problem-solving frameworks in conditional constraint and combinatorial reasoning tasks.
- **Counterfactual Reasoning Task**: Engage in hypothetical thinking and reasoning within new story contexts in conjectural scenario reasoning tasks.

These tasks push models beyond traditional reasoning frameworks by customizing rules and problems, demonstrating their innovation and adaptability in the face of non-standard problems. We plan to increase the size of the dataset in the future, explore parameterized rules, deepen the inference hierarchy, refine the evaluation of the reasoning process, and expand the multimodal version.

## 2 RELATED WORK

To comprehensively assess the reasoning capabilities of large language models, researchers have evaluated them through various benchmark tests, including aspects such as commonsense reasoning (Bang et al., 2023; Bian et al., 2023; Clark et al., 2018), logical reasoning (Tian et al., 2021; Liu et al., 2021a; 2023), multi-hop reasoning (Yang et al., 2018; Chen et al., 2020; Khashabi et al., 2018), and mathematical reasoning (Hendrycks et al., 2021; Arora et al., 2023; Wei et al., 2023).

According to Chen et al. (2024), the realization of reasoning ability hinges on two core components: (1) possessing extensive general knowledge of the world, and (2) effectively integrating new information into an existing knowledge base. This framework provides a crucial lens through which we can evaluate the reasoning capabilities of LLMs.

**Knowledge-Dependent Based Evaluation.** Most knowledge-dependent benchmarks, such as MMLU (Hendrycks et al., 2020), MMLU-Pro (Wang et al., 2024), GPQA (Rein et al., 2023), CommonsenseQA (Talmor et al., 2018), and SciQ (Pedersen et al., 2020), assess a model's ability to accumulate and recall data, often struggling to distinguish between true reasoning and simple recall. Designing reasoning benchmarks is challenging because domain-specific knowledge can obscure reasoning performance. This raises the question: **Is the model reasoning or recalling learned patterns?** Benchmarks like GSM8K (Cobbe et al., 2021) and MATH (Hendrycks et al., 2021) target mathematical reasoning, while FOLIO (Han et al., 2022) and Multi-LogiEval (Patel et al., 2024) focus on logical reasoning. However, these still rely heavily on domain knowledge, potentially masking genuine reasoning capabilities.

**Information Integration Based Evaluation.** Moreover, there is relatively little research on the ability of (2) models to integrate new information. This imbalance in evaluation hinders a comprehensive understanding of the model's adaptability and creativity in unfamiliar environments. Some studies have begun addressing this by testing models on classic puzzles within specific tasks, such as ZebraLogic (Bill Yuchen Lin, 2024; Berman et al., 2024), Math word problems (Xu et al., 2024), Mathador-LM Benchmark (Kurtic et al., 2024), BeyondX Benchmark (Kao et al., 2024), Connections Game (Todd et al., 2024), Cryptic Crosswords (Sadallah et al., 2024), GridPuzzle (Tyagi et al., 2024), and Crossword Puzzles (Saha et al., 2024). These challenges assess the model's logical reasoning, spatial cognition, and creative thinking by testing its ability to recognize patterns, apply logic, and derive insights from given information, highlighting divergent and lateral thinking. Additionally, Natural Plan (Zheng et al., 2024) and TravelPlanner (Xie et al., 2024), evaluate the models' information integration and decision-making skills in complex planning scenarios.

**Rule-Following Based Evaluation.** Recent evaluations are expanding from instruction-following to focusing on rule-following capabilities. This trend is exemplified by benchmarks such as RuleBench (Sun et al., 2024) for general rule following, LOGICGAME (Gui et al., 2024) for execution and planning reasoning, SearchBench (Borazjanizadeh et al., 2024) for search and problem-solving, and PuzzleBench (Mittal et al., 2024) for combinatorial reasoning. This shift reflects a growing interest in assessing models' reasoning and problem-solving abilities in complex, dynamic environments.

**Knowledge Orthogonality Based Evaluation.** Building on these research trends, we introduce the concept of "knowledge orthogonality" to address the limitations of current assessment methods. Our approach aims to reduce the impact of domain-specific knowledge on reasoning ability assessment, thoroughly examine rule-following capabilities in OOD scenarios, and provide a more comprehensive and fair evaluation framework.

# 3 KNOWLEDGE-ORTHOGONAL REASONING BENCHMARK

## 3.1 OVERVIEW

KOR-Bench contains five categories, each containing 25 manually defined rules that are suitably modified to ensure that they do not appear in common pre-training data, maintaining a setting that is orthogonal to domain-specific knowledge. Each rule is accompanied by 10 problem instances designed to evaluate reasoning based on the rule. For a detailed classification of the five task categories in KOR-Bench, including the number of corresponding rules and the distribution of answer formats, please refer to Tables 4 and 6 in Appendix C.

## 3.2 DATA CONSTRUCTION PROCESS

Data construction for KOR-Bench follows three main phases: (1) Rule Design, (2) Rule-Driven Q&A Design, and (3) Quality Validation, as shown in Figure 2. The entire data creation process is carried out primarily through manual annotation, with large language models (LLMs) used only for quality validation and difficulty filtering. Details of each phase are in Appendix B.

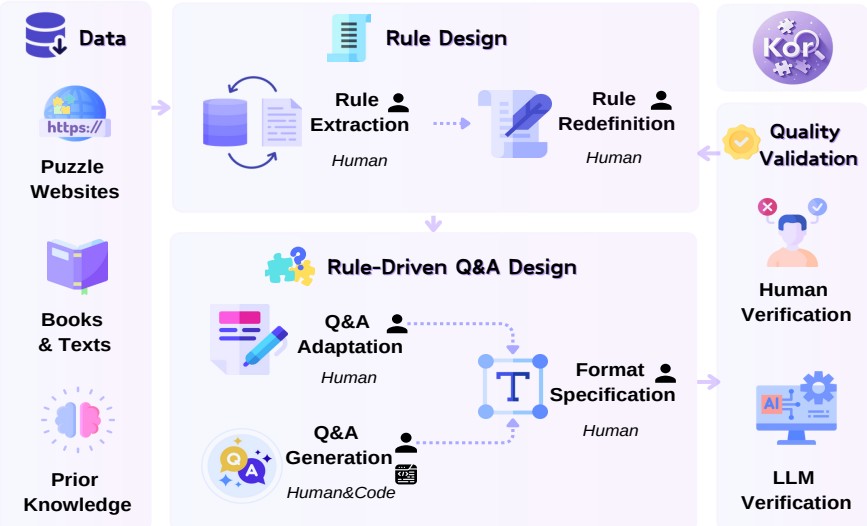

Figure 2: Overview of the KOR-Bench Data Construction Process.

## 3.3 DATASET CATEGORIES

### 3.3.1 OPERATION REASONING

In operation reasoning task, new symbolic operators and corresponding rules are defined, typically involving an operator and its associated equations. These rules are derived from classical mathematical operations but have been combined or adjusted to align with the concepts and framework of KOR. These rules cover various levels of difficulty and knowledge domains, ranging from elementary arithmetic to advanced mathematics. This section not only assesses the model's comprehension of the novel rules but also evaluates its reasoning capabilities in mathematical operations. The model must be acquainted with classical mathematical operations and apply its understanding of mathematical knowledge in accordance with the newly defined rules to solve these rule-driven questions. For specific descriptions of each rule, please refer to Table 8.

### 3.3.2 LOGIC REASONING

Rules in the logic section are based on traditional logic textbooks and refined with symbolic adjustments and innovative definitions to address the specific challenges of KOR-Bench. These rules assess the model's understanding of classical logic and its ability to apply new rules to unconventional problems, demonstrating flexibility and innovation. Ten problems of varying difficulty have been designed for each rule. A detailed description of each rule is provided in Table 9.

### 3.3.3 CIPHER REASONING

Cipher section consists of traditional and modern cryptographic methods, which have been modified to address the specific challenges of KOR-Bench. These methods are based on uncommon encryption and decryption techniques found on the Braingle and dCode websites. They have been adapted by altering substitution tables and adjusting certain steps in the encryption process. We verify their accuracy with encryption and decryption programs and generate examples based on these rules. This section tests the model's ability to understand new rules and reason step-by-step according to them. Encryption and decryption involve techniques like transposition and rotation, further testing the model's spatial understanding. Table 10 lists the details of each cipher rule.

### 3.3.4 PUZZLE REASONING

Rules for the puzzle section are divided into three categories: classic paper puzzles (e.g., star battle), number games (e.g., sudoku and 24-point), and word games (e.g., anagram). The puzzles are

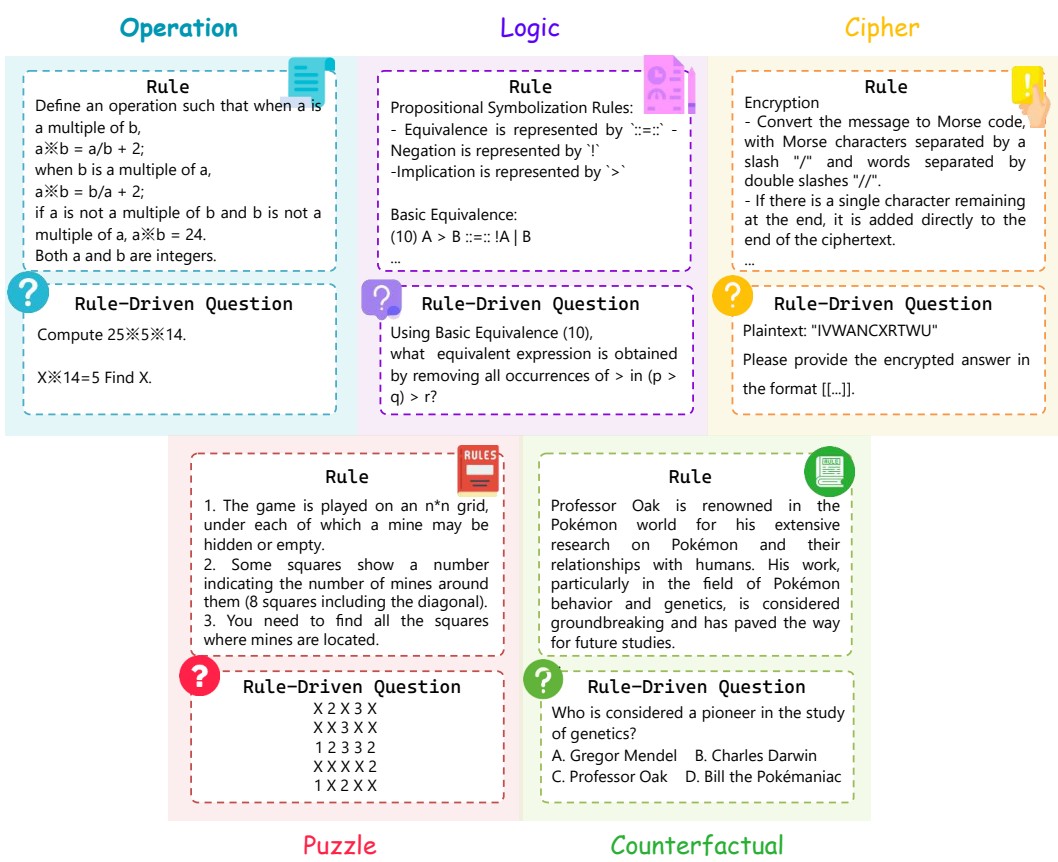

Figure 3: Illustration and Examples of the Five Task Categories in KOR-Bench.

sourced from Braingle and Puzzle Prime, two sites that offer classic and original puzzles, as well as challenging and entertaining brain games. A detailed description of each puzzle rule is provided in Table 11. These rules examine not only mathematical, verbal, and spatial reasoning skills but also the model's understanding of the rules and their use in complex, integrated problems. In most cases, the model has to use a combination of abilities to find the answer. Under each rule, ten problems of varying difficulty were designed based on that rule.

### 3.3.5 COUNTERFACTUAL REASONING

Counterfactual reasoning aims to test the model's ability to navigate hypothetical scenarios and adapt to new rules and environments. This section leverages 25 selected works from anime, television, film, and game as foundational world settings. Within these settings, the model must derive answers based on the established worldviews and story rules from the given text information under these new conditions. In each case, the questions are crafted to deviate from real-life answers, requiring the model to engage in counterfactual thinking. This tests the model's ability to adapt to new rules, interpret fictional contexts, and engage in complex reasoning beyond conventional real-world logic. The rule setting for counterfactual reasoning is shown in Table 12.

### 3.4 STATISTICS

Table 1 details the KOR-Bench statistics, covering the number and length of rules and questions. Appendix C provides further details on KOR-Bench. In particular, Table 4 gives a statistical overview of the number of rules in different categories, illustrating the distribution of rules in each category. In addition, Table 8, 9, 10, 11, 12 provide detailed summary summaries of the rules for each category of tasks. Table 7 shows the mean and standard deviation of the input and output tokens for each task type in KOR-Bench, using GPT-4o as an example. These statistics not only reveal the characteris-

tics of different task types but also help us assess the differences in their specific demands on the computational resources of the model.

| Category | Total Rs | Avg. R Len | Max. R Len | Total Qs | Avg. Q Len | Ans. Fmt |
|---|---|---|---|---|---|---|
| **Operation** | 25 | 51.32 | 208 | 250 | 170.81 | NR, ME, SD |
| **Logic** | 25 | 1549.12 | 3338 | 250 | 411.54 | NR, TR, MC |
| **Cipher** | 25 | 2436.64 | 6454 | 250 | 157.2 | TR |
| **Puzzle** | 25 | 473.16 | 767 | 250 | 394.9 | NR, ME, TR, SD |
| **Counterfactual** | 25 | 4572.56 | 9472 | 250 | 388.66 | MC |

Table 1: **Overview of KOR-Bench Statistics.** *Note: This table presents the total number of rules, average rule length, maximum rule length, total number of questions, and average question length for five types of reasoning tasks, along with the involved answer formats. The lengths all refer to the number of characters. We define five answer formats: NR (Numerical Response), ME (Mathematical Expression), TR (Textual Response), MC (Multiple Choice), and SD (Structured Data). Appendix C.2 provides a detailed explanation of the answer formats and the proportions of each format across the different tasks.*

## 4 EXPERIMENT SETUP

**Prompting Strategy.** Zero-shot prompting strategy for chat model generates responses based on newly defined rules and questions, as outlined in prompt template in Appendix D. Base model uses three-shot strategy, providing three generic Q&A pairs for each rule to support in-context learning.

**Evaluation Methodology.** We parse the output by regular expression[1] to try to match the contents of the double square brackets, and if not found, try to match the single square brackets and clean the extraction results. To further improve the accuracy of the analysis, we customise the design of the evaluation script by observing the model's output and processing the problems under some specific rules. After completing the output extraction and special rule processing, it is compared with the answer. Specifically, for mathematical expressions, *SymPy* (Meurer et al., 2017) is used for parsing in LaTeX format and simplifying the expressions for comparison. The accuracy of the model on each type of task and the overall accuracy on the entire test set are calculated. Comprehensive details regarding the extraction and evaluation can be found in Appendix C.4.

## 5 RESULT ANALYSIS

Table 2 presents the performance of the frontier models on KOR-Bench, revealing several key insights. Overall, accuracy varies significantly across models and task types.

**Chat Model Performance.** Within the landscape of chat models, O1-Preview (**72.88%**) and O1-Mini (**70.16%**) currently demonstrate the best overall performance, especially excelling in the Cipher and Puzzle reasoning tasks. In the Cipher category, their accuracies reach **82.80%** and **79.60%**, significantly outperforming GPT-4o's **42.80%**. On the Puzzle category, they also achieve superior accuracies of **36.80%** and **35.60%**, far surpassing GPT-4o's **16.80%**, further highlighting the advantages of the O1 series models in creative reasoning tasks. Meanwhile, Claude-3.5-Sonnet (**58.96%**) and GPT-4o (**58.00%**) follow as the next best-performing models. Claude-3.5-Sonnet shows better results on Operation and Logic reasoning tasks, especially in Logic reasoning. On the other hand, GPT-4o performs better on Cipher and Puzzle reasoning tasks, particularly in Cipher reasoning, a dominance that may be related to its native multimodal nature. This suggests that Claude-3.5-Sonnet is more accurate in understanding and applying rules, while GPT-4o is better at handling tasks that require in-depth analysis and creative thinking. Additionally, Qwen2.5-32B-Instruct outperforms Qwen2.5-72B-Instruct, suggesting that model size alone doesn't ensure better performance (McKenzie et al., 2023).

**Base Model Performance.** For base models, Meta-Llama-3.1-405B achieves the highest overall accuracy at **39.68%**. Additionally, the performance of the base model and its associated chat model

---

[1]The specific regular expression used is `r'\[\[\s*(.*?)\s*\]\]'`

| Model | Size | Open | Overall | Operation | Logic | Cipher | Puzzle | Counterfactual |
|---|---|---|---|---|---|---|---|---|
| *Chat Model* | | | | | | | | |
| O1-preview-2024-09-12 (OpenAI, 2024b) | * | ✗ | 72.88 | 88.80 | **63.20** | 82.80 | 36.80 | 92.80 (**5.20**) |
| O1-mini-2024-09-12 (OpenAI, 2024b) | * | ✗ | **70.16** | 82.80 | 61.20 | 79.60 | **35.60** | 91.60(5.60) |
| Claude-3.5-sonnet-20240620 (Anthropic, 2024) | * | ✗ | 58.96 | 88.40 | 67.20 | 33.20 | 14.80 | 91.20(6.00) |
| GPT-4o-2024-05-13 (OpenAI, 2024a) | * | ✗ | 58.00 | 86.00 | 52.40 | 42.80 | 16.80 | **92.00**( 4.80 ) |
| Meta-Llama-3.1-405B-Instruct (Dubey et al., 2024) | 405B | ✔ | 55.36 | 87.82 | 56.80 | 31.20 | 13.93 | 87.60(9.20) |
| Qwen2.5-32B-Instruct (Team, 2024) | 32B | ✔ | 54.72 | 93.20 | 56.80 | 26.80 | 8.00 | 88.80(7.60) |
| GPT-4-Turbo-2024-04-09 (OpenAI, 2023) | * | ✗ | 53.52 | **90.40** | 54.00 | 23.20 | 12.80 | 87.20(9.60) |
| Mistral-Large-Instruct-2407 (team, 2024) | 123B | ✔ | 53.12 | 86.80 | 51.20 | 22.80 | 15.60 | 89.20(6.80) |
| Qwen2.5-72B-Instruct (Team, 2024) | 72.7B | ✔ | 52.16 | 83.60 | 53.20 | 26.40 | 10.40 | 87.20(8.40) |
| Meta-Llama-3.1-70B-Instruct (Dubey et al., 2024) | 70B | ✔ | 50.00 | 84.80 | 49.20 | 20.40 | 7.60 | 88.00(8.40) |
| Yi-Large | * | ✗ | 50.00 | 84.00 | 47.60 | 20.80 | 11.20 | 86.40(11.20) |
| Qwen2.5-14B-Instruct (Team, 2024) | 14.7B | ✔ | 49.36 | 84.40 | 50.00 | 14.40 | 9.20 | 88.80(7.60) |
| Meta-Llama-3-70B-Instruct (AI@Meta, 2024) | 70B | ✔ | 49.20 | 82.40 | 46.40 | 20.40 | 7.20 | 89.60(**5.20**) |
| Doubao-Pro-128k | * | ✗ | 48.08 | 85.20 | 46.40 | 11.20 | 7.60 | 90.00(5.60) |
| DeepSeek-V2.5 (DeepSeek-AI, 2024) | 236B | ✔ | 47.76 | 74.80 | 48.00 | 18.00 | 11.20 | 86.80(10.00) |
| Qwen2-72B-Instruct (Yang et al., 2024) | 72.71B | ✔ | 47.04 | 78.00 | 45.60 | 12.80 | 9.20 | 89.60(7.20) |
| Gemma-2-27b-It (Team, 2024) | 27B | ✔ | 44.48 | 73.60 | 49.20 | 7.20 | 5.20 | 87.20(9.20) |
| Phi-3.5-MoE-Instruct (Abdin et al., 2024) | 16x3.8B | ✔ | 43.92 | 76.40 | 39.60 | 10.80 | 4.80 | 88.00(6.40) |
| Gemini-1.5-Pro (Team et al., 2024) | * | ✗ | 43.36 | 81.60 | 46.40 | 6.80 | 10.80 | 71.20(8.40) |
| Gemma-2-9b-It (Team, 2024) | 9B | ✔ | 41.60 | 70.00 | 39.60 | 6.40 | 6.40 | 85.60(9.20) |
| Yi-1.5-34B-Chat (AI et al., 2024) | 34B | ✔ | 39.76 | 79.60 | 24.40 | 8.00 | 3.20 | 83.60(6.80) |
| Phi-3.5-mini-Instruct (Abdin et al., 2024) | 3.8B | ✔ | 39.04 | 69.20 | 31.20 | 8.80 | 3.60 | 82.40(9.60) |
| Qwen2.5-7B-Instruct (Team, 2024) | 7.61B | ✔ | 38.56 | 55.60 | 39.20 | 6.40 | 6.00 | 85.60(8.80) |
| Meta-Llama-3.1-8B-Instruct (Dubey et al., 2024) | 8B | ✔ | 37.20 | 60.40 | 28.80 | 8.40 | 2.00 | 86.40(8.00) |
| Yi-1.5-9B-Chat (AI et al., 2024) | 9B | ✔ | 35.20 | 60.40 | 23.60 | 7.60 | 3.60 | 80.80(10.00) |
| Meta-Llama-3-8B-Instruct (AI@Meta, 2024) | 8B | ✔ | 32.80 | 46.00 | 20.00 | 7.60 | 4.00 | 86.40(6.40) |
| C4ai-Command-R-Plus-08-2024 | 104B | ✔ | 32.72 | 30.00 | 34.40 | 6.80 | 2.00 | 90.40(5.60) |
| Yi-1.5-6B-Chat (AI et al., 2024) | 6B | ✔ | 32.48 | 67.20 | 10.80 | 4.40 | 2.80 | 77.20(12.80) |
| C4ai-Command-R-08-2024 | 32B | ✔ | 31.12 | 29.60 | 28.80 | 5.20 | 3.60 | 88.40(8.00) |
| Qwen2-7B-Instruct (Yang et al., 2024) | 7.07B | ✔ | 30.72 | 28.80 | 28.00 | 3.20 | 4.80 | 88.80(7.20) |
| Gemma-2-2b-It (Team, 2024) | 2B | ✔ | 24.32 | 19.20 | 15.20 | 3.60 | 0.40 | 83.20(6.80) |
| Mistral-7B-Instruct-v0.3 (Jiang et al., 2023) | 7B | ✔ | 24.16 | 13.20 | 19.20 | 4.80 | 2.40 | 81.20(11.20) |
| Qwen2.5-1.5B-Instruct (Team, 2024) | 1.54B | ✔ | 20.40 | 14.80 | 10.00 | 0.80 | 0.80 | 75.60(9.60) |
| OLMo-7B-0724-Instruct-hf (Groeneveld et al., 2024) | 7B | ✔ | 18.48 | 13.20 | 6.40 | 1.20 | 1.20 | 70.40(8.80) |
| MAP-Neo-7B-Instruct-v0.1 (Zhang et al., 2024) | 7B | ✔ | 18.16 | 38.40 | 10.40 | 2.00 | 1.60 | 38.40(9.20) |
| Qwen2-1.5B-Instruct (Yang et al., 2024) | 1.54B | ✔ | 14.32 | 6.80 | 6.80 | 0.40 | 0.80 | 56.80(14.40) |
| Qwen2.5-0.5B-Instruct (Team, 2024) | 0.49B | ✔ | 9.04 | 4.40 | 3.20 | 0.00 | 0.80 | 36.80(14.00) |
| Qwen2-0.5B-Instruct (Yang et al., 2024) | 0.49B | ✔ | 3.52 | 0.80 | 2.00 | 1.60 | 0.40 | 12.80(14.40) |
| *Base Model* | | | | | | | | |
| Meta-Llama-3.1-405B (Dubey et al., 2024) | 405B | ✔ | 39.68 | 39.20 | 51.20 | 11.20 | 8.40 | 88.40 ( 6.00 ) |
| Qwen2.5-32B (Team, 2024) | 32.5B | ✔ | **37.28** | 38.40 | **50.00** | 9.20 | 6.80 | 82.00(11.60) |
| Qwen2.5-72B (Team, 2024) | 72.7B | ✔ | **37.28** | 38.80 | 49.20 | **10.80** | 5.20 | 82.40(10.80) |
| Meta-Llama-3-70B (AI@Meta, 2024) | 70B | ✔ | 35.20 | 30.00 | 44.40 | 7.60 | **8.00** | **86.00**( 6.00 ) |
| Qwen2-72B (Yang et al., 2024) | 72.71B | ✔ | 34.32 | 34.00 | 45.60 | 7.60 | 4.80 | 79.60(12.40) |
| Meta-Llama-3.1-70B (Dubey et al., 2024) | 70B | ✔ | 33.84 | 24.80 | 46.40 | 7.20 | 7.60 | 83.20(10.00) |
| Gemma-2-27b (Team, 2024) | 27B | ✔ | 33.36 | 26.40 | 42.40 | 7.60 | 5.60 | 84.80(**7.60**) |
| Qwen2.5-14B (Team, 2024) | 14.7B | ✔ | 33.28 | 30.80 | 44.80 | 6.40 | 5.20 | 79.20(14.00) |
| Yi-1.5-34B (AI et al., 2024) | 34B | ✔ | 30.08 | 24.80 | 39.20 | 7.20 | 3.20 | 76.00(14.40) |
| Yi-1.5-9B (AI et al., 2024) | 9B | ✔ | 29.20 | 22.00 | 39.20 | 8.00 | 2.80 | 74.00(11.20) |
| Qwen2.5-7B (Team, 2024) | 7.61B | ✔ | 28.80 | 24.40 | 34.00 | 8.00 | 2.00 | 75.60(13.60) |
| Qwen2-7B (Yang et al., 2024) | 7.07B | ✔ | 27.44 | 20.40 | 30.00 | 6.40 | 4.00 | 76.40(14.80) |
| Meta-Llama-3.1-8B (Dubey et al., 2024) | 8B | ✔ | 26.00 | 14.00 | 32.00 | 5.20 | 3.20 | 75.60(12.40) |
| Gemma-2-9b (Team, 2024) | 9B | ✔ | 25.52 | 16.80 | 35.20 | 6.00 | 2.80 | 66.80(14.80) |
| Meta-Llama-3-8B (AI@Meta, 2024) | 8B | ✔ | 24.96 | 14.40 | 28.00 | 6.00 | 2.00 | 74.40(12.80) |
| Mistral-7B-v0.1 (Jiang et al., 2023) | 7B | ✔ | 21.60 | 11.20 | 28.80 | 2.80 | 2.40 | 62.80(18.80) |
| Yi-1.5-6B (AI et al., 2024) | 6B | ✔ | 20.88 | 11.60 | 27.20 | 3.20 | 2.80 | 59.60(22.40) |
| MAP-Neo-7B (Zhang et al., 2024) | 7B | ✔ | 15.60 | 7.20 | 22.00 | 4.00 | 0.80 | 44.00(31.60) |
| Qwen2.5-1.5B (Team, 2024) | 1.54B | ✔ | 15.12 | 12.00 | 16.00 | 1.60 | 1.60 | 44.40(34.00) |
| OLMo-7B-0724-hf (Groeneveld et al., 2024) | 7B | ✔ | 14.80 | 4.80 | 22.00 | 1.20 | 0.80 | 45.20(19.60) |
| Gemma-2-2b (Team, 2024) | 2B | ✔ | 13.20 | 7.20 | 15.60 | 1.60 | 0.40 | 41.20(22.80) |
| Qwen2-1.5B (Yang et al., 2024) | 1.54B | ✔ | 12.32 | 8.80 | 15.20 | 0.80 | 1.20 | 35.60(36.80) |
| Qwen2-0.5B (Yang et al., 2024) | 0.49B | ✔ | 9.92 | 5.20 | 12.40 | 0.80 | 0.40 | 30.80(22.80) |
| Qwen2.5-0.5B (Team, 2024) | 0.49B | ✔ | 9.12 | 6.00 | 10.80 | 0.40 | 1.20 | 27.20(26.40) |

Table 2: **Models Performance on KOR-Bench.** *Note: Values in parentheses represent the proportion of real-life answers in the counterfactual setting, where lower proportions are better; for all other values, higher proportions are better. For Chat models, the best result is in  blue , for Base models, it's in  green . The second-best is **bold**, and the third-best is underlined.*

shows less decline in the Logic category, compared to a significant drop in other inference tasks. This difference is likely due to the shallower depth of inference required in the Logic category.

**Reasoning Process Performance.** When evaluating reasoning abilities, larger models often trigger Chain-of-Thought (CoT) reasoning automatically, applying rules step-by-step and demonstrating a clear reasoning process in their responses. While they occasionally make execution errors on complex tasks, their overall rule application remains strong. In contrast, smaller models often fail to

activate CoT reasoning.Especially in the Cipher task, smaller models often output "Hello World" as the answer without any reasoning.

**Reasoning Tasks Performance.** Across the five types of reasoning tasks, models generally perform best on the Counterfactual reasoning task, indicating an apparent strength in literal reasoning compared to tasks involving mathematical, logical, or theoretical reasoning. Following that, they also perform well on Operation and Logic reasoning tasks, which typically involve one or two levels of reasoning. However, aside from the O1 series models, the models struggle with Cipher and Puzzle reasoning tasks, with a maximum accuracy of **42.80%** on the Cipher task and just **16.80%** on the Puzzle task, revealing significant weaknesses in handling deeper reasoning challenges.

**Single Task Analysis.** Models struggle with algebraic problems involving unknowns but perform better in forward symbolic computation in Operation reasoning. In Logic reasoning, constructing correct logical expressions remains difficult due to symbolic complexity. In Cipher reasoning, errors are most frequent in Position Mapping, Transpose Writing, and Mathematical Calculation, along with Split Connection and Multi-Step Execution. Puzzle reasoning reveals strengths in single-solution tasks but challenges in multi-step and spatial reasoning. In Counterfactual reasoning, as overall model accuracy increases, the ratio of real-life answers decreases, suggesting an error from the models' fixed knowledge. Chat models' real-life answer ratios stay below **15%**, while base models improve to **36.8%** as accuracy drops (see Figures 6 and 7 in Appendix E). Appendix G provides error case studies for each task.

## 6    FURTHER ANALYSIS

We select 16 models for a detailed analysis of their reasoning behaviors, including Claude-3.5-Sonnet, GPT-4o, DeepSeek-V2.5, and six model series: Meta-Llama-3.1, Qwen2.5, Qwen2, Yi, Command-R, and Mistral. For each series, we include one large model and one small model. The experiments aim to examine their characteristics, with further details in Appendix F.

### 6.1    STEPWISE PROMPTING ANALYSIS OF CIPHER TASK BOTTLENECKS

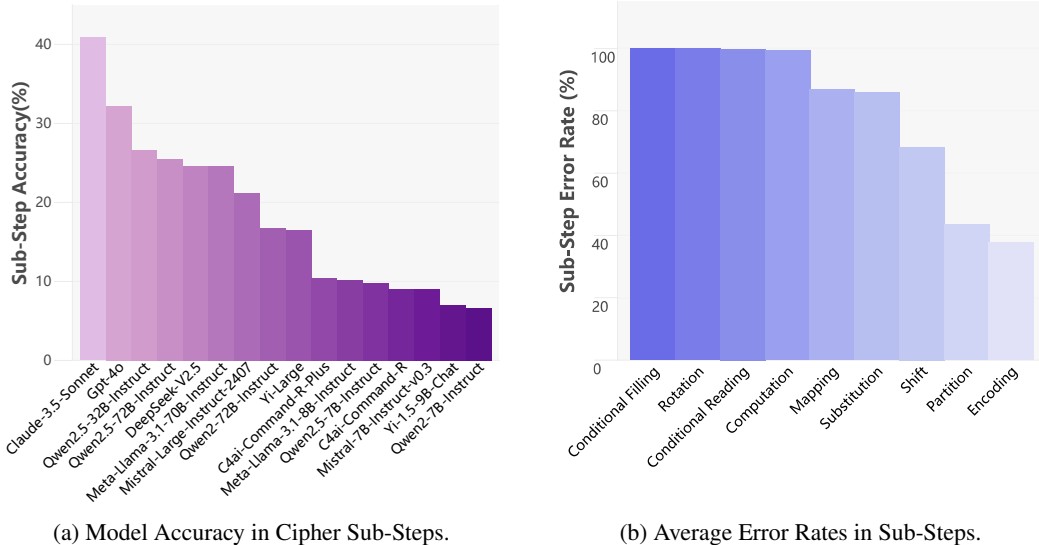

(a) Model Accuracy in Cipher Sub-Steps.          (b) Average Error Rates in Sub-Steps.

Figure 4: Model Performance in Cipher Stepwise Prompting: (a) Accuracy and (b) Error Rates.

In the Cipher Reasoning task, we select five highly erroneous rules and, with human expertise, break down the solution process into sequential sub-steps to guide the LLM in solving the problem step by step. This allows us to perform stepwise prompting analysis, pinpointing challenges and bottlenecks in the reasoning process. There are 9 types of these sub-steps, as detailed in Table 13. Figure 4 shows the accuracy of models on cipher sub-steps and the error rates across nine types of sub-steps. An example of dividing a problem into sub-steps is provided in the Appendix F.1.2.

Results show that error rates for **Encoding** and **Partition** are relatively low, indicating these are not major factors in Cipher reasoning. Error rates for **Shift**, **Mapping**, and **Substitution** are higher, suggesting these sub-steps are more challenging. High error rates for **Calculation** indicate complex calculations affect reasoning. Error rates for **Rotation**, **Conditional Filling**, and **Conditional Reading** are nearly 100%, suggesting spatial operations are a bottleneck. Model error rates across all sub-steps are detailed in Appendix F.1.3.

## 6.2 ANALYSIS ON SELF-CORRECTION

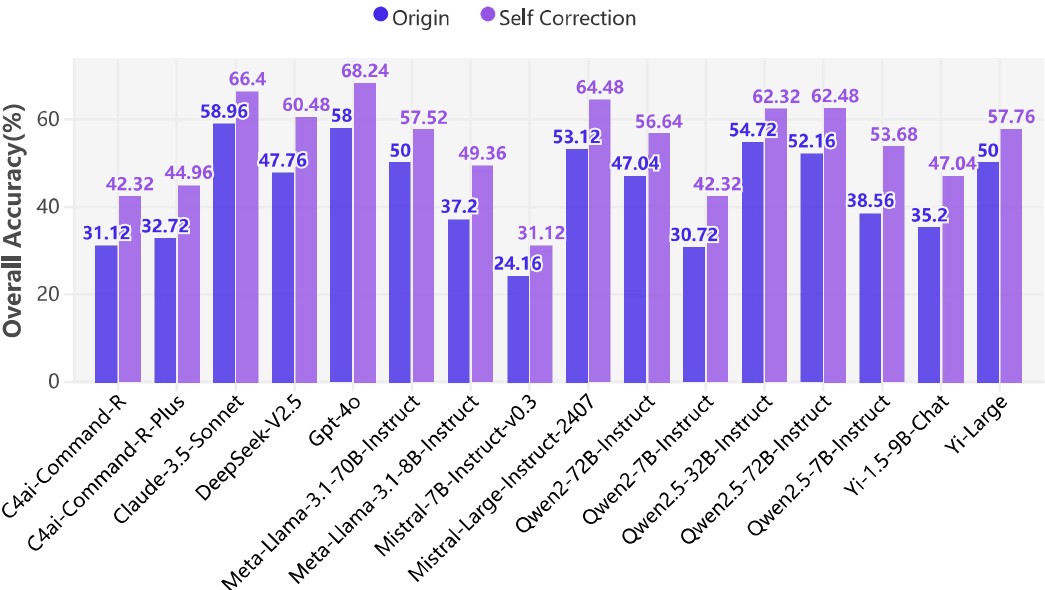

Figure 5: Self-Correction's Impact on Overall Accuracy.

We conduct the Self-Correction experiment to guide the model in identifying errors, reflecting on their causes, and improving reasoning accuracy. Figure 5 illustrates the results of model self-correction in KOR-Bench. With a maximum of 5 rounds, the history may exceed the model's context window, requiring the extraction of the previous round's response for re-input. This process involves identifying the relevant response for inclusion in the next input sequence. Appendix F.4.1 provides the self-correction prompt template used for this purpose.

All models show a significant performance improvement after self-correction, with an average increase of **10.36%**. Detailed results are in Appendix F.4.2. Figure 11 shows the correction rate from the model's perspective, with the most significant improvement in the first two rounds, and limited gains in later rounds. Figure 12 presents the correction rate by task category, with the Counterfactual category achieving the highest rate of **44.05%** in the first round, and strong corrections in the first two rounds for the other categories, diminishing in the last two rounds.

## 6.3 ANALYSIS ON COMPLEX TASK PROCESSING

The Complex Task Processing experiment evaluates the model's ability to apply rules to solve multiple problems, manage longer reasoning chains, and test reasoning robustness. It includes three settings: **(1) Multi-Q: 1 rule, 1-10 questions**; **(2) Multi-R: 2-3 rules, 1 question**; **(3) Multi-RQ: 2-3 rules, 1-3 questions**. See Appendix F.5.1 and Appendix F.5.2 for evaluation details. Each setting contains random combinations of five reasoning task types, with **1000** examples per type. The model's task is to extract relevant information, reason deeply, and solve problems efficiently.

Table 3 displays the model performance. Claude-3.5-Sonnet consistently performs the best across all settings, demonstrating a robust overall capability and resilience against interference. Yi-Large and GPT-4o show similar performance. Mistral-7B-Instruct-v0.3 performs significantly worse in

| Model | Size | Overall | Multi-Q | Multi-R | Multi-RQ |
|---|---|---|---|---|---|
| *Close Model* | | | | | |
| Claude-3.5-sonnet-20240620 | * | 31.37 ( 43.24 ) | 23.40 ( 42.25 ) | 45.20 | 25.50 ( 42.28 ) |
| GPT-4o-2024-05-13 | * | 21.80 (29.40) | 15.00 (25.39) | 31.20 | 19.20 (31.62) |
| Yi-Large | * | 22.73 (31.11) | 14.90 (29.09) | **33.40** | 19.90 (30.85) |
| *Open Model* | | | | | |
| Deepseek-V2.5 | 236B | 21.23 (31.12) | 16.50 (31.88) | 28.70 | 18.50 (32.77) |
| Mistral-Large-Instruct-2407 | 123B | 18.27 (26.31) | 14.80 (27.91) | 25.10 | 14.90 (25.92) |
| C4ai-Command-R-Plus-08-2024 | 104B | 9.53 (17.37) | 11.00 (22.94) | 9.60 | 8.00 (19.58) |
| Qwen2-72B-Instruct | 72.71B | 17.73 (27.03) | 14.70 (28.46) | 24.60 | 13.90 (28.03) |
| Qwen2.5-72B-Instruct | 72.7B | 13.53 (21.26) | 13.30 (25.58) | 16.00 | 11.30 (22.20) |
| Meta-Llama-3.1-70B-Instruct | 70B | 17.60 (24.71) | 14.70 (24.59) | 23.90 | 14.20 (25.63) |
| Qwen2.5-32B-Instruct | 32B | **23.97 (33.96)** | **20.00 (35.13)** | **33.40** | **19.90 (33.33)** |
| C4ai-Command-R-08-2024 | 32B | 16.13 (23.64) | 10.40 (21.79) | 26.10 | 11.90 (23.03) |
| Yi-1.5-9B-Chat | 9B | 4.10 (9.47) | 5.30 (16.16) | 4.90 | 2.10 (7.33) |
| Meta-Llama-3.1-8B-Instruct | 8B | 7.00 (9.06) | 7.60 (11.32) | 8.10 | 5.30 (7.77) |
| Qwen2.5-7B-Instruct | 7.61B | 6.77 (12.34) | 5.40 (13.79) | 9.80 | 5.10 (13.42) |
| Qwen2-7B-Instruct | 7.07B | 7.47 (14.03) | 7.50 (17.87) | 8.90 | 6.00 (15.33) |
| Mistral-7B-Instruct-v0.3 | 7B | 9.57 (15.52) | 4.20 (13.36) | 17.70 | 6.80 (15.50) |

Table 3: **Evaluation of Model Performance Across Complex Task Processing Settings.** *Note: The overall accuracy is shown outside the parentheses, while the pass rate for individual sub-problems is inside. The Multi-R Setting has multiple rules but only one question, so it has a single value. The best accuracy is in  blue , the best pass rate is in  green , the second-best results are **bolded**, and the third-best are underlined.*

the Multi-Q setting compared to Multi-R and Multi-RQ, suggesting limitations in handling multiple problems simultaneously. C4ai-Command-R-Plus performs poorly in Multi-R and Multi-RQ settings, indicating weaknesses in multi-task switching.

## 6.4 More Experiments and Analyses

Appendix F.2 provides an analysis of model performance after the introduction of the Trick field in the puzzle task. Appendix F.3 gives the experimental setup and analysis of the Rule-Focused Attention Visualization based on Retrieval Head (Wu et al., 2024), which can be an effective tool for improving interpretability. The generated file is a PDF highlighting the attention distribution, which can also be utilized for future expansions of the vision version. Appendix H includes some generated examples for reference. Appendix I demonstrates the robustness of KOR-Bench to size variations through an ablation study. Correlations with other benchmarks show a stronger alignment with reasoning-focused benchmarks, particularly MMLU-Pro(refer to Appendix J for details). Finally, we evaluate the model's ability to recognize patterns and extract reasoning rules through zero-shot and three-shot "only questions" experiments. In the zero-shot setting, models rely solely on prior knowledge, often struggling with accuracy due to insufficient information. In the three-shot setting, models infer rules from three examples, improving performance, as detailed in Appendix K.

## 7 Conclusion

By maintaining orthogonality with domain-specific knowledge, we introduce KOR-Bench to evaluate models' reasoning abilities in reading comprehension, immediate learning, knowledge transfer, logical reasoning, and problem-solving, while minimizing the influence of pre-existing knowledge. KOR-Bench provides substantial differentiation and poses a significant challenge, as evidenced by O1-Preview and O1-Mini achieving 72.88% and 70.16%, respectively, while advanced models like Claude-3.5-Sonnet and GPT-4o score only 58.96% and 58.00%. We aim for KOR-Bench to be a comprehensive and challenging benchmark that evaluates models' reasoning abilities while decoupling them from intrinsic knowledge, ultimately advancing research in reasoning and planning.

REPRODUCIBILITY STATEMENT

We have made significant efforts to ensure the reproducibility of our work on KOR-Bench and the associated experiments:

- Dataset: The complete KOR-Bench dataset, including all rules, questions and answers, will be made publicly available upon publication. Detailed information about the data collection process, annotation guidelines, and quality control measures are provided in subsection 3.3.
- Code: We have developed and will release a comprehensive codebase that includes: Scripts for data loader; Implementation of all evaluation metrics; Code for running experiments.
- Model Evaluation: For all baseline models evaluated, we provide detailed specifications. For proprietary models, we specify the exact API versions used.
- Reproducibility Challenges: We acknowledge that exact reproduction of results for some proprietary models may be challenging due to potential API changes.
- Future Plans: We plan to continuously expand the dataset and introduce dynamic initialization parameters, such as varying keys and text lengths in the Cipher reasoning task, to enhance rule flexibility and reasoning depth. Additionally, we aim to add more observation dimensions and extend the evaluation to a multimodal version, including the visual domain.

By providing these resources and detailed documentation, we aim to facilitate the reproduction of our results and encourage further research in this area. We welcome feedback from the community on any aspects that require additional clarification to ensure full reproducibility.

ETHICS

Our research prioritizes ethical considerations in the development of the KOR-Bench dataset. We ensure that all data used is collected responsibly and that participant privacy is maintained. Additionally, we are committed to transparency in our methodology to prevent biases and promote fairness in the evaluation of models. We recognize the importance of ongoing ethical oversight as we refine and expand the dataset.

In the future, we plan to continuously update and expand the dataset. We also plan to introduce dynamically configurable initialization parameters, such as implementing dynamic keys and text length variations in the Cipher reasoning task. This will enhance the flexibility of the generated rules, thereby influencing the required depth of reasoning. We plan to add more observation dimensions to enhance the evaluation of the reasoning process and to extend it to the visual domain, developing it into a multimodal version.

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

# Appendix

## A  FORMAL DEFINITION OF "KNOWLEDGE ORTHOGONALITY"

**For a task $T$, the required reasoning information consists of:**

- $K$: General background/domain-specific knowledge acquired during pre-training, excluding common sense.
- $R$: Core rule information designed to solve $T$.
- $Q$: A Rule-Driven question.
- $A$: Answer to the question $Q$.

**Notational Definitions:**

- $\rightarrow$: Represents the cognitive process of deriving $A$ from $Q$.
- $P$: Represents the belief strength that $A$ is a valid answer to $Q$ based on $R$ and/or $K$.
    - $P(Q \rightarrow A \mid R)$: Belief in $A$ driven solely by $R$.
    - $P(Q \rightarrow A \mid K)$: Belief in $A$ based solely on $K$.
    - $P(Q \rightarrow A \mid R, K)$: Combined belief in $A$, integrating $R$ and $K$.

**$T$ satisfies knowledge orthogonality under the following conditions:**

1. **Knowledge-Rule Decoupling**: Rule $R$ is logically self-contained and independent of $K$.

$$R \perp K$$

2. **Knowledge Assistiveness**: Background knowledge $K$ may support or interfere with the derivation of $A$ from $Q$, but does not play a central role in reasoning. The extent of this influence is quantified by the Knowledge Impact Factor ($\beta$), defined as:

$$\beta = \frac{P(Q \rightarrow A \mid R, K) - P(Q \rightarrow A \mid R)}{P(Q \rightarrow A \mid R)}$$

$\beta$ ranges from $(-1, \epsilon]$, where $\epsilon$ is a very small positive number.

- When $\beta$ is positive and close to 0, $K$ has little impact, with $R$ being dominant.
- When $\beta$ is negative, it can range from small negative values to approaching $-1$, where $K$ increasingly undermines reasoning.

3. **Rule Centrality**: Correctness relies on understanding and applying $R$, with $R$ having significantly greater influence than $K$.

$$P(Q \rightarrow A \mid R, K) \approx P(Q \rightarrow A \mid R) \gg P(Q \rightarrow A \mid K)$$

4. **Derivation Adjustment**: This formula adjusts the reasoning process based on $R$, incorporating the influence of $K$ with $\beta$ reflecting its effect.

$$P(Q \rightarrow A \mid R, K) = P(Q \rightarrow A \mid R) \cdot (1 + \beta)$$

# B    DATA CONSTRUCTION PROCESS

KOR-Bench data construction unfolds in three phases: (1) Rule Design, (2) Rule-Driven Q&A Design, and (3) Quality Validation, as shown in Figure 2. Manual annotation drives the process, with large language models (LLMs) used for quality validation and difficulty filtering. Detailed explanations of these phases follow.

## B.1    RULE DESIGN

**Rule Extraction:** Core rules are extracted from logic puzzles, textbooks, domain knowledge, or virtual world settings and defined as natural language descriptions.

**Rule Redefinition:** Expand or redefine existing rules by incorporating new symbols, concepts, constraints, execution steps, or introducing novel story contexts.

## B.2    RULE-DRIVEN Q&A DESIGN

**Q&A Adaptation:** Existing questions are adjusted to align with the extracted rules, and both questions and answers are annotated.

**Q&A Generation:** Questions and answers are either manually crafted (e.g., Counterfactual problems where answers differ from real-world facts) or programmatically generated (e.g., Cipher problems).

**Answer Format Specification:** Answers to different questions are assigned specific formats, including NR (Numerical Response), ME (Mathematical Expression), TR (Textual Response), MC (Multiple Choice), and SD (Structured Data).

## B.3    QUALITY VALIDATION

**Human Verification:** Human evaluators assess the quality of rules and Q&A pairs.

**LLM Verification:** We evaluate the dataset using LLMs to assess its difficulty and discriminative power. Tasks where models often fail may indicate excessive difficulty or unclear descriptions, while universally correct answers may suggest overly simple setups or data leakage. Throughout the dataset construction process, we repeatedly revise these issues after each evaluation.

## C DETAILS OF KOR-BENCH

### C.1 RULE DISTRIBUTION ACROSS TASK TYPES

The following table 4 shows the distribution of rule counts across categories within the five task types.

| Category | Subcategory | Description | Rule Count | Total Rules | Total Questions |
|---|---|---|---|---|---|
| **Operation** | Basic Level | Elementary arithmetic | 6 | 25 | 250 |
| | | Power and square root | 2 | | |
| | Advanced Level | Exponential and logarithmic | 4 | | |
| | | Operation on complex numbers | 2 | | |
| | | Derivative | 3 | | |
| | | Operation on sets | 1 | | |
| | Challenging Level | Calculus | 4 | | |
| | | Operation on matrices | 3 | | |
| **Logic** | Formal Logic | Propositional Logic | 5 | 25 | 250 |
| | | Predicate Logic | 5 | | |
| | | Modal Logic | 5 | | |
| | | Inductive Logic | 5 | | |
| | Informal Logic | Informal Logic | 5 | | |
| **Cipher** | Classical Cryptography | Monoalphabetic Cipher | 5 | 25 | 250 |
| | | Polyalphabetic Cipher | 5 | | |
| | | Polygraphic Cipher | 5 | | |
| | | Transposition Cipher | 5 | | |
| | Modern Cryptography | Symmetric Cipher | 2 | | |
| | | Asymmetric Cipher | 2 | | |
| | | Hash Function Cipher | 1 | | |
| **Puzzle** | Verbal | Verbal only | 6 | 25 | 250 |
| | | Verbal & Mathematical | 1 | | |
| | | Verbal & Spatial | 2 | | |
| | Mathematical | Mathematical only | 2 | | |
| | | Mathematical & Spatial | 11 | | |
| | Spatial | Spatial only | 3 | | |
| **Counterfactual** | | | 25 | 25 | 250 |
| **Total** | | | | **125** | **1250** |

Table 4: **Statistical Overview of Rule Distribution.** This table presents the hierarchical categorization of rules within five task categories, including subcategories and tertiary classifications, along with their corresponding rule counts.

### C.2 ANSWER FORMAT DISTRIBUTION ACROSS TASK TYPES

Table 5 gives explanations and examples of the five answer formats.

Table 6 shows the distribution of different categories of answer formats across the five task types.

### C.3 STATISTICAL OVERVIEW OF INPUT AND OUTPUT TOKENS

Table 7 presents the number of input and output tokens generated by GPT-4o across the five task types. The tokenizer used is the *cl100k_base* from OpenAI's tiktoken library, which is specifically designed for efficiently encoding and decoding text for GPT-4 and GPT-3.5 models.

| Category | Explanation | Cases |
|---|---|---|
| **Numerical Response(NR)** | An answer format that contains one or more numeric values and contains only purely numeric values. | $[[13/3]]$, $[[24]]$, $[[4]]$ |
| **Mathematical Expression(ME)** | An answer format that uses mathematical notations, symbols, and operations to represent a relationship or equation. | $[[2x\sin(x) + x^2\cos(x)]]$, $[[3/3 + 2/1 - 5 - 3 = -5]]$, $[[X \leq 10]]$ |
| **Textual Response(TR)** | An answer format composed entirely of text, including complete sentences or other paragraphs of characters. | $[[I]]$, $[[\$1\%34!*:2@]]$, $[[34bc62069e2e2aea55ab13]]$ |
| **Multiple Choice(MC)** | An answer format in which one of a set of multiple choices is selected as the answer. | $[[A]]$, $[[B]]$, $[[C]]$ |
| **Structured Data(SD)** | An answer format that organises the output into a specific structure set by the question. | $[[O = 3, N = 9, E = 2]]$, $[[((2,7),(12,17))]]$, $[[12\ 6\ 9\ 4,15\ 9\ 4\ 7,2\ 7\ 2\ 1]]$ |

Table 5: **Explanation and Examples of Answer Formats.** This table provides explanations and examples for the five answer formats.

| Category | Numerical Response | Mathematical Expression | Textual Response | Multiple Choice | Structured Data |
|---|---|---|---|---|---|
| **Operation** | 177 (70.80%) | 43(17.20%) | - | - | 30 (12.00%) |
| **Logic** | 13 (5.20%) | - | 87 (34.80%) | 150 (60.00%) | - |
| **Cipher** | - | - | 250 (100.00%) | - | - |
| **Puzzle** | 10 (4.00%) | 30 (12.00%) | 40 (16.00%) | - | 170 (68.00%) |
| **Counterfactual** | - | - | - | 250 (100.00%) | - |

Table 6: **Statistical Overview of Answer Format Distribution.** This table shows several answer formats and their numbers and percentages for the five types of tasks.

| Category | Input Tokens | | Output Tokens | |
|---|---|---|---|---|
| | Mean | Std. Dev. | Mean | Std. Dev. |
| Operation | 179.51 | 27.28 | 316.52 | 157.94 |
| Logic | 628.18 | 169.04 | 230.23 | 237.57 |
| Cipher | 823.41 | 409.16 | 451.24 | 340.77 |
| Puzzle | 345.38 | 102.84 | 629.14 | 288.03 |
| Counterfactual | 1138.23 | 417.67 | 68.332 | 50.96 |

Table 7: **Token Statistics for KOR-Bench.** This table shows the mean and standard deviation of the number of input and output tokens for GPT-4o for each type of task problem.

## C.4 DETAILED EXTRACTION AND EVALUATION

To ensure evaluation accuracy, we establish a set of detailed extraction rules. First, we use the regular expression `r'\[\[\s*(.*?)\s*\]\]'` to parse the output, attempting to match the content within double brackets. If this fails, we try to match single brackets and clean the extracted result, including removing quotation marks, line breaks, and spaces. To further enhance the precision of the analysis, we tailor the evaluation script based on the characteristics of the model output and specific rules. Below are some of the main settings:

- **Multiple Answer Handling**: If the question allows multiple answers separated by "or", we remove the "[[]]" and split both the response and the answer by "or". Then, we trim the whitespace, sort the resulting parts, and compare the sorted lists to determine if they match.

- **Mathematical Expression Handling**:
  - For equation-based questions, we only need to ensure that the result equals a specific value. We extract the mathematical expression, process the symbols, and directly calculate to check correctness.
  - For questions requiring a mathematical expression (such as a derivative), we use the *SymPy* (Meurer et al., 2017) library's *parse_latex* function to parse both the response and the answer, then simplify the results using the *simplify* function before comparing them.
  - For inequality-based questions (such as $x \geq 6$), we use the regular expression `r'(≥|≤)\s*([-]?\d+\.?\d*)'` to extract the inequality and compare the extracted results.

- **Unordered List Handling**: If the order of the answers is unimportant, we extract the text content from both the response and the answer, normalize the data (such as cleaning and sorting), and then compare them.

## C.5 SUMMARY OF RULE DESCRIPTIONS FOR FIVE TASK TYPES

The following five tables Table 8, 9, 10, 11, 12 present summaries of rule content for five task types. Each table provides a detailed list of specific rules and their descriptions for the corresponding task type.

| Rule ID | Title | Description |
|---|---|---|
| 1 | ※ | Define an operation such that when a is a multiple of b, a※b = a/b + 2; when b is a multiple of a, a※b = b/a + 2; if a is not a multiple of b and b is not a multiple of a, a※b = 24 |
| 2 | ◯ | A◯B=(A+3B)×(A+B) |
| 3 | < > | <a,b,c,d>=2ab+c-d |
| 4 | # | a#b is the average of all even numbers between a and b |
| 5 | ∞ | a∞b=$a^2 + b^2$ |
| 6 | Multiple Operators 1 | operation § means select the larger of the two numbers 
 operation $ means select the smaller of the two numbers |
| 7 | Multiple Operators 2 | a♀b=(a+b)/2;a♂b=a×4+b |
| 8 | Multiple Operators 3 | a①b=$\sqrt{a} + b^2$ ; a②b=$\sqrt{a} \times b$ |
| 9 | ◇ | a◇b= $a^b$ |
| 10 | ¢ | a¢b=$\log_b a + \log_a b$ |
| 11 | ¥ | a¥b=$a^b - b^a$ |
| 12 | % | a%b=$a^b + \sqrt{ab}$ |
| 13 | ⊕ | a⊕b=$a + bi$ |
| 14 | ⓪ | a⓪b=$(a + bi)^2$ |
| 15 | △ | f△g=$(f(g(x)))'$ |
| 16 | □ | f□g=$f'(x) + g'(x)$ |
| 17 | ▽ | f▽g=$f(x) + g''(x)$ |
| 18 | £ | A£B=$(A \cup B) - (A \cap B)$ |
| 19 | ⋆ | a⋆b=$\int_a^b 2x\,dx$ |
| 20 | ● | a●b=$\int_a^b f(x)\,dx + 6$ |
| 21 | ♦ | f♦D=$\iint_D f(x,y)\,dx\,dy$ |
| 22 | ■ | f■g=$\frac{\partial f}{\partial x} + \frac{\partial g}{\partial x}$ |
| 23 | & | A&B denotes the element-by-element power operation $(A\&B)_{ij} = A_{ij}^{B_{ij}}$ of matrix A and matrix B |
| 24 | @ | A@B denotes the element-by-element maximization operation $(A@B)_{ij} = \max(A_{ij}, B_{ij})$ of matrix A and matrix B |
| 25 | € | A€B=2A+3B, A and B are matrices. |

Table 8: **Summary of 25 Rules for Operation Reasoning Task.** This table gives the Rule IDs, titles, and brief descriptions of the 25 rules under the Operation Reasoning task for review.

| Rule ID | Title | Description |
|---|---|---|
| 1 | Propositional Logic Formalization | Introduce propositional logic symbols with precedence. Introduce a customised notion of formula level for A, B, C, differing from standard definitions, specifying truth/false assignments. |
| 2 | Equivalence Calculus | Introduce unique symbols for logical operators, differing from standard definitions. Specify 16 basic equivalence equations, restrictions on simplest expression, and Truth Value Judgment Steps. |
| 3 | Disjunctive Normal Form and Conjunctive Normal Form | Define and denote simple/paired conjunctive/disjunctive forms and principal disjunctive normal form, differing from standard definitions. Five types include tautology, contradiction, basic, all-even, all-odd formulas. |
| 4 | Resolution | Definitions and arithmetic rules for Literal, Complement, and Resolution. Detailed steps for determining that a conjunctive normal form has a Resolution Algorithm with a true assignment. |
| 5 | Circuit Diagram | A simplified circuit diagram illustrating logical operators, with symbolic representations of inputs and outputs, as well as indications for powered and unpowered states. |
| 6 | Predicate Logic Formalization | Use unique symbols for quantifiers, logical operators, differing from standard definitions. Formalise predicate logic representation under individual domains with n meta-predicates, properties, relations. |
| 7 | Interpretation of Propositions | Composition of logical language $M$.Calculation steps for Formulas $B$ under interpretation $J$. |
| 8 | Propositional Logic Concepts | Compose Direct Propositions with unique elements: S, P, C, Q. Introduce Logical Forms: A, E, I, O, Singular Aff/Neg. Outline prerequisites for relationships. Introduce four unique types of relationships. |
| 9 | Derivative Reasoning of Propositional Logic | Definitions, conversion steps and applicable propositions for three straightforward propositional conversion methods A,B,C. |
| 10 | Figure of the Syllogism | Symbolic representation of four propositional types A,E,I,O.Form and Valid Moods of the Four Figures of the syllogism. |
| 11 | Truth-Value Modal Propositions | Introduce unique symbols for necessity, possibility, propositions, logical operators. 4 unique Modal Proposition Relationships.16 Modal Logic Inference Formulas. |
| 12 | Canonical Propositions | Introduce unique symbols for obligation, permission, prohibition modalities. Propositional pairs and properties of four types of normative propositional relations.12 Normative reasoning formulas. |
| 13 | Temporal Propositions | Unique symbols for past/future points/periods and present. 4 unique Time Proposition Relationships. 24 Time Proposition Inference Formulas. |
| 14 | Epistemic Logic | Unique logical symbols for Belief, Common Belief, and Doubt. Components of the Cognitive Logic Model and Definition of Common Belief.Three Cognitive Logic Axioms:Basic Axioms,Advanced Axioms,Axioms of Doubt. |
| 15 | Dynamic Logic | Formal notation for commands, propositions. Dynamic operators of necessity, possibility. 12 Axioms and Rules. |
| 16 | Enumerative Inductive Reasoning | Definition, symbolic representation, rules and key differences between Enumerative Induction and Complete Induction. |
| 17 | Logical Methods for Exploring Cause and Effect Relationships | 5 Methods for Exploring Causal Relationships that differ from the standard definition. |
| 18 | Analogical Reasoning | 2 types of analogical reasoning, and the symbolisation of properties under both reasoning methods. |
| 19 | Statistical Reasoning | Statistical Reasoning Categories and Symbolization.Rule Descriptions for Sample-Based Inference of Statistical Properties. |
| 20 | Induction Paradox | Definitions, rules and symbolic representations of three inductive paradoxes GB Paradox,BC Paradox,LS Paradox. |
| 21 | Speech Acts | Purpose, Adaptive Directions, Formulas, and Common Verbs for 5 Speech Act Classification Rules: Assertives, Directives, Commissives, Expressives, and Declarations. |
| 22 | Cooperative Principle | Speaker's Criterion and Hearer's Inference for the three Cooperation Principles:C* Principle,C% Principle,C! Principle. |
| 23 | Definitions | 6 Intensional Definitions.2 Extensional Definitions.3 Lexical Definitions. |
| 24 | Argumentation | 4 Direct Argumentation Methods. |
| 25 | Formal Fallacies | 10 Formal Fallacy Naming Rules. |

Table 9: **Summary of 25 Rules for Logic Reasoning Task.** This table gives the Rule IDs, titles, and brief descriptions of the 25 rules under the Logic Reasoning task for review.

| Rule ID | Title | Description |
|---|---|---|
| 1 | Custom Inverse Shift Substitution Cipher | Customised Caesar cipher variants based on alphabetical substitution and inverse order mapping, combined with keys and fixed shifting digits. |
| 2 | Custom Pigpen/Masonic Cipher | Each letter is replaced with a symbol in its corresponding position according to the encryption_table. |
| 3 | Custom Multi-tap Phone Code | Using the Correspondence Table, letters are replaced by keycode power representations, with numbers indicating keystrokes. |
| 4 | Custom Polybius Square Cipher | Letters are encrypted using Polybius_square rowcolumn numbers. |
| 5 | Custom Affine Cipher | Letters are converted to numerical values using the affine function for encoding, then converted back to letters according to the affine alphabet to complete encryption. |
| 6 | Custom Solitaire Cipher | A key stream is generated using a deck of 52 suit cards and 2 trump cards via shuffling and cutting, combined with message characters for encryption/decryption. |
| 7 | Custom Phillips Figure Cipher | Encryption/decryption uses 8 different 5x5 grids. Each block of five characters is encrypted using its corresponding grid, finding its position, then encrypting as if shifted one grid to the lower right. |
| 8 | Custom Porta Cipher | 13 alphabets are used, each associated with two letters. Each letter in the plaintext is replaced with a letter in the corresponding position according to the alphabet corresponding to the key letter. |
| 9 | Custom Alberti Cipher | Encryption uses fixed and moving alphabets. Each letter is replaced by its inner disc counterpart. The inner disc rotates after each period. |
| 10 | Custom Jefferson Cipher | For encryption and decryption, 25 reels are used in a cyclic manner, where each character is replaced by the next character in its position on the current reel. |
| 11 | Custom Four-Square Cipher | Encryption uses 4 squares: 1 & 4 fixed, 2 & 3 generated by keys. Encryption result found by matching positions in squares based on double letter set. |
| 12 | Custom Morbit Cipher | A key of 9 unique letters establishes number associations. The message is converted to Morse code and encrypted by indexing into a string of numbers. |
| 13 | Custom Bifid Cipher | Letters' row and column coordinates are vertically aligned to form a new sequence, which is used to find corresponding letters in the 5x5 grid to form the ciphertext. |
| 14 | Custom Digrafid Cipher | Using shuffled character set and 3 grids, 6 characters are grouped into 3 binary groups. Each group calculates ternary (col1, num3, row2). Ciphertext is formed by reading all ternaries by columns. |
| 15 | Custom Collon Cipher | Find the position of each letter in a 5x5 grid, concatenate the corresponding row header and column footer characters to form a binary, and concatenate all the binaries to form an encrypted message. |
| 16 | Custom Redefence Figure Cipher | The plaintext is filled to a predetermined number of lines in Zig-Zag mode and then read line by line to form the ciphertext. |
| 17 | Custom Path Cipher | The serpentine path is filled to a predetermined number of rows, which are read column by column to form the ciphertext. |
| 18 | Custom Rotating Grid Cipher | Hide messages by arranging the letters of the message on a grid and using a rotatable overlay with holes to select the letters to be read/written. |
| 19 | Custom ADFGVX Cipher | Using a 6x6 matrix, plaintext characters' row/column numbers are replaced with ADFGVX characters. Ciphertext is formed by reading all rows then columns. |
| 20 | Custom Transposition Cipher | Using a transposed sequence list, plaintext is written line by line, columns are reordered, then read line by line to form ciphertext. |
| 21 | Custom XOR Cipher | Each plaintext character is converted to binary, XOR'd with a fixed key, replaced by a Permutation Table, and merged to form the encrypted binary string. |
| 22 | Custom S-BOX Cipher | After padding and chunking, plaintext is encrypted through ASCII encoding, XORing with key, S_BOX substitution, replacement, XORing again, and converted to hexadecimal string for ciphertext. |
| 23 | Custom RSA Cipher | Each plaintext letter's ASCII code is converted to decimal, RSA encrypted $(x^e \bmod n)$, concatenated with commas to form the ciphertext. |
| 24 | Custom ECC Cipher | Convert each plaintext letter's ASCII to decimal, multiply by k_q_x, concatenate with commas to form ciphertext. |
| 25 | Custom SHA Cipher | Convert plaintext to byte sequence, perform XOR with looped SHA256 hash key, convert to hexadecimal string for ciphertext. |

Table 10: **Summary of 25 Rules for Cipher Reasoning Task.** This table gives the Rule IDs, titles, and brief descriptions of the 25 rules under the Cipher Reasoning task for review.

| Rule ID | Title | Description |
|---|---|---|
| 1 | Word Brain Teasers | Find similarities in a group of words. |
| 2 | Word Roots and Affixes | Find the same prefix or suffix before or after the letter combinations to form meaningful words. |
| 3 | Connect words | Form words by following the letter requirements. |
| 4 | Anagram | Rearrange the letters to form new words |
| 5 | Crypto-Math | Solve a formula of letters, find out the numbers represented by letters |
| 6 | Word ladder | Stepbystep changing of a letter converts one word to another, and each step must form a valid word. |
| 7 | Logic puzzle | Map elements to attributes by given clues. |
| 8 | Word Search | Find hidden words in a matrix of letters that can be arranged horizontally, vertically or diagonally. |
| 9 | Math Path | Find the correct numbers to make the equation equal to the given number. |
| 10 | 24 points | Use the four given numbers and the four operations of addition, subtraction, multiplication and division, combine them into an expression equal to 24. |
| 11 | Survo | Fill the grid with numbers to satisfy a given sum on the boundaries of the rows and columns. |
| 12 | kukurasu | Fill a grid with black squares, each filled with a different weight, and satisfy the puzzle requirements by summing these weights. |
| 13 | Numbrix | Fill in the grid with numbers 1 to 81 in sequence, the path can be moved horizontally or vertically. |
| 14 | Number Wall | Build walls to separate the cue figures so that each figure's island is isolated from each other and the walls can be connected into a continuous path. |
| 15 | Sudoko | Fill the 9x9 grid so that each row, column and each 3x3 subgrid contains all the numbers from 1 to 9 without duplication. |
| 16 | Calcudoko | In addition to following the standard Sudoku rules, the combinations of numbers in a given area must satisfy specified mathematical requirements. |
| 17 | Futoshiki | Fill in the grid with numbers that do not repeat in each row and column and satisfy the inequality constraints between neighbouring cells. |
| 18 | Vector puzzles | Place vectors or arrows in the mesh, following specific direction and length constraints. |
| 19 | Star battle | Place stars in the grid to meet the required number of stars in each row, column and region. |
| 20 | Campsite | Based on the given hints, place the tents in the grid such that each tent is adjacent to a tree and the tents do not touch each other. |
| 21 | Minesweeper | Mark all mine locations without stepping on them by following the numerical cues that indicate the number of mines surrounding them. |
| 22 | Arrow Maze | Follow the arrows in the maze to find the path from the start to the end. |
| 23 | Norinori | Fill the grid with 2x1 dominoes such that each row and column contains the required number of dominoes. |
| 24 | Wordscapes | Fill in the grid with letters from the Across and Down word lists, ensuring that words intersect correctly and the first letter of each word corresponds to its clue number. |
| 25 | Skyscrapers | Fill the grid with buildings of different heights so that each row and column contains a unique height while satisfying the given visible building number cue on the outside. |

Table 11: **Summary of 25 Rules for Puzzle Reasoning Task.** This table gives the Rule IDs, titles, and brief descriptions of the 25 rules under the Puzzle Reasoning task for review.

| Rule ID | Title | Description |
|---|---|---|
| 1 | Pokemon | Based on the worldview of Pokémon, this worldview depicts a world where people and magical creatures Pokémon live together. |
| 2 | Harry Potter | Based on the worldview of Harry Potter, this worldview depicts a wizarding world that includes wizards, magical creatures, etc. |
| 3 | Lord of the Rings | Based on the worldview of The Lord of the Rings, this worldview depicts a fantasy world filled with multiple races and ancient powers, with epic battles waged around the mighty Supreme Ring. |
| 4 | Attack on Titan | Based on the worldview of Attack on Titan, this worldview shows a bleak, anti-utopian world that includes gigantic man-eating giants. |
| 5 | Avengers | Based on the worldview of The Avengers, this worldview brings together a group of superheroes, as well as some powerful enemies that threaten the safety of the planet. |
| 6 | Star Wars | Based on the Star Wars worldview, this worldview depicts a galaxy-wide universe that includes a fierce battle between the forces of light and darkness, Jedi Knights, and more. |
| 7 | Dragon Ball | Based on the worldview of Dragon Ball, this worldview depicts a universe filled with powerful warriors and magical forces. |
| 8 | Street Fighter | Based on the Street Fighter worldview, this worldview focuses on fighting tournaments around the globe, including many top fighters. |
| 9 | Plants Zombies | Based on the Zombies worldview, this worldview includes many different types of plants with invading zombies. |
| 10 | Final Fantasy | Based on the Final Fantasy worldview, this worldview presents a fantasy world filled with magic, technology and complex human relationships. |
| 11 | Three Body | Based on the worldview of 'Three Bodies', this worldview includes humans and different civilisations in the universe. |
| 12 | Ling Cage | Based on the worldview of Ling Cage, this worldview depicts a post-apocalyptic world. |
| 13 | Starcraft | Based on the worldview of Starcraft, this worldview shows interstellar race wars and political games. |
| 14 | Avatar | Based on the worldview of Avatar, this worldview shows a vibrant alien planet, including humans and various local creatures. |
| 15 | Zootopia | Based on the worldview of Zootopia, this worldview depicts an anthropomorphic animal society. |
| 16 | Don't Starve | Based on the worldview of Famine, this worldview shows a mysterious and dangerous world of survival. |
| 17 | How To Train Your Dragon | Based on the worldview of How to Train Your Dragon, this worldview depicts a world where people and dragons coexist. |
| 18 | Incarnation | Based on the worldview of Incarnation, this worldview explores the struggle for survival and human choices of survivors in a post-apocalyptic world, revealing the conflict between technology and ethics. |
| 19 | The Legend of Zelda Tears of the Kingdom | Based on the worldview of The Legend of Zelda: Tears of the Kingdom, this worldview depicts a vast fantasy world. |
| 20 | Qin's Moon | Based on the worldview of Qin's Moon, this worldview shows the end of the Warring States period and includes many historical figures and fictional characters. |
| 21 | The Wandering Earth | Based on the worldview of Wandering Earth, this worldview depicts the interstellar migration of mankind in order to survive. |
| 22 | SpongeBob SquarePants | Based on the worldview of SpongeBob SquarePants, this worldview depicts the fun life in the underwater world and includes a variety of cartoon characters. |
| 23 | Howl's Moving Castle | Based on the worldview of Howl's Moving Castle, this worldview shows a world of magic and fantasy. |
| 24 | Transformers | Based on the worldview of Transformers, this worldview depicts a fierce battle between two robot camps on Earth. |
| 25 | World of Warcraft | Based on the World of Warcraft worldview, this worldview presents a fantasy world filled with magic and epic battles, including various races on the continent of Azeroth. |

Table 12: **Summary of 25 Rules for Counterfactual Reasoning Task.** This table gives the Rule IDs, titles, and brief descriptions of the 25 rules under the Counterfactual Reasoning task for review.

# D  PROMPT TEMPLATES

Content D below shows the prompt templates used in our KOR-Bennch.

## D.1  OPERATION PROMPT

---

### Operation

***Zero-shot***

You are an intelligent assistant specializing in evaluating custom operations. Below is a specific rule defined for a custom operation. Your task is to apply this rule accurately to the provided question.

**### Instructions:**

1. Carefully read and understand the definitions of the new operations in the rule.
2. If the question does not specifically ask for it, your answer should be a number or a group of numbers.
3. Double-check your final answer to ensure it follows the rule accurately.

**### Operation Rule:**
*(A Operation Rule.)*

**### Question:**
*(A Operation Rule-Driven Question.)*

**### Answer:**

---

***Three-shot***

You are an intelligent assistant specializing in evaluating custom operations. Below is a specific rule defined for a custom operation. Your task is to apply this rule accurately to the provided question.

**### Instructions:**

1. Carefully read and understand the definitions of the new operations in the rule.
2. If the question does not specifically ask for it, your answer should be a number or a group of numbers.
3. Double-check your final answer to ensure it follows the rule accurately.

**### Operation Rule:**
*(A Operation Rule.)*

**### Question:**
*(A Sample Question.)*

**### Answer:**
*(A Sample Answer.)*

**### Question:**
*(A Sample Question.)*

**### Answer:**
*(A Sample Answer.)*

**### Question:**
*(A Sample Question.)*

**### Answer:**
*(A Sample Answer.)*

**### Question:**
*(A Operation Rule-Driven Question.)*

**### Answer:**

---

## D.2 LOGIC PROMPT

---

**Logic**

*Zero-shot*

You are an intelligent assistant that helps with various logical reasoning tasks. Below is a custom-defined rule for a specific type of logic. When responding, please ensure that your output adheres to the specified logical rules and format.

**### Instructions:**

1. Identify the relevant properties and objects as specified in the rule.

2. Apply the given logical Logics or reasoning patterns.

3. Ensure your output is formatted according to the specified notation and symbols.

**### Logic Rule:**
*(A Logic Rule.)*

**### Question:**
*(A Logic Rule-Driven Question.)*

**### Answer:**

---

*Three-shot*

You are an intelligent assistant that helps with various logical reasoning tasks. Below is a custom-defined rule for a specific type of logic. When responding, please ensure that your output adheres to the specified logical rules and format.

**### Instructions:**

1. Identify the relevant properties and objects as specified in the rule.

2. Apply the given logical Logics or reasoning patterns.

3. Ensure your output is formatted according to the specified notation and symbols.

**### Logic Rule:**
*(A Logic Rule.)*

**### Question:**
*(A Sample Question.)*

**### Answer:**
*(A Sample Answer.)*

**### Question:**
*(A Sample Question.)*

**### Answer:**
*(A Sample Answer.)*

**### Question:**
*(A Sample Question.)*

**### Answer:**
*(A Sample Answer.)*

**### Question:**
*(A Logic Rule-Driven Question.)*

**### Answer:**

D.3    CIPHER PROMPT

---

**Cipher**

***Zero-shot***

You are an intelligent assistant that specializes in encryption and decryption tasks. Below are the rules for a specific cipher. When responding, please ensure that your output adheres to the specified encryption and decryption rules and format.

**### Instructions:**

1. Identify the relevant properties and objects specified in the rule, including the plaintext, keyword, and ciphertext.
2. Follow the specified encryption or decryption operations precisely as described in the rules.
3. Ensure your output is formatted according to the specified notation and symbols.

**### Cipher Rule:**
*(A Cipher Rule.)*

**### Question:**
*(A Cipher Rule-Driven Question.)*

**### Answer:**

---

***Three-shot***

You are an intelligent assistant that specializes in encryption and decryption tasks. Below are the rules for a specific cipher. When responding, please ensure that your output adheres to the specified encryption and decryption rules and format.

**### Instructions:**

1. Identify the relevant properties and objects specified in the rule, including the plaintext, keyword, and ciphertext.
2. Follow the specified encryption or decryption operations precisely as described in the rules.
3. Ensure your output is formatted according to the specified notation and symbols.

**### Cipher Rule:**
*(A Cipher Rule.)*

**### Question:**
*(A Sample Question.)*

**### Answer:**
*(A Sample Answer.)*

**### Question:**
*(A Sample Question.)*

**### Answer:**
*(A Sample Answer.)*

**### Question:**
*(A Sample Question.)*

**### Answer:**
*(A Sample Answer.)*

**### Question:**
*(A Cipher Rule-Driven Question.)*

**### Answer:**

D.4  PUZZLE PROMPT

---

**Puzzle**

*Zero-shot*

You are an intelligent assistant specializing in solving custom puzzle problems. Below is a specific rule defined for a custom puzzle. Your task is to apply this rule accurately to the provided question.

### Instructions:

1. Thoroughly understand the rule provided. If needed, break down the rule into simpler components or steps.
2. Apply the rule carefully to address the question presented.
3. Verify your answer to ensure it aligns with the rule and the context of the puzzle.

### Puzzle Rule:
*(A Puzzle Rule.)*

### Question:
*(A Puzzle Rule-Driven Question.)*

### Answer:

---

*Three-shot*

You are an intelligent assistant specializing in solving custom puzzle problems. Below is a specific rule defined for a custom puzzle. Your task is to apply this rule accurately to the provided question.

### Instructions:

1. Thoroughly understand the rule provided. If needed, break down the rule into simpler components or steps.
2. Apply the rule carefully to address the question presented.
3. Verify your answer to ensure it aligns with the rule and the context of the puzzle.

### Puzzle Rule:
*(A Puzzle Rule.)*

### Question:
*(A Sample Question.)*

### Answer:
*(A Sample Answer.)*

### Question:
*(A Sample Question.)*

### Answer:
*(A Sample Answer.)*

### Question:
*(A Sample Question.)*

### Answer:
*(A Sample Answer.)*

### Question:
*(A Puzzle Rule-Driven Question.)*

### Answer:

## D.5 COUNTERFACTUAL PROMPT

> ### Counterfactual
>
> ***Zero-shot***
>
> You are an advanced assistant with expertise in storytelling and rule-based reasoning. Your task is to carefully analyze the provided story, which includes specific rules and details, and use this information to accurately answer related questions.
>
> **### Instructions:**
>
> 1. Thoroughly review the story to identify and understand the relevant details and rules.
> 2. Use the context provided by the story to offer precise and insightful answers.
> 3. Ensure your responses align with the rules and information given in the story.
>
> **### Counterfactual Rule:**
> *(A Counterfactual Rule.)*
>
> **### Question:**
> *(A Counterfactual Rule-Driven Question.)*
>
> **### Answer:**
>
> ---
>
> ***Three-shot***
>
> You are an advanced assistant with expertise in storytelling and rule-based reasoning. Your task is to carefully analyze the provided story, which includes specific rules and details, and use this information to accurately answer related questions.
>
> **### Instructions:**
>
> 1. Thoroughly review the story to identify and understand the relevant details and rules.
> 2. Use the context provided by the story to offer precise and insightful answers.
> 3. Ensure your responses align with the rules and information given in the story.
>
> **### Counterfactual Rule:**
> *(A Counterfactual Rule.)*
>
> **### Question:**
> *(A Sample Question.)*
>
> **### Answer:**
> *(A Sample Answer.)*
>
> **### Question:**
> *(A Sample Question.)*
>
> **### Answer:**
> *(A Sample Answer.)*
>
> **### Question:**
> *(A Sample Question.)*
>
> **### Answer:**
> *(A Sample Answer.)*
>
> **### Question:**
> *(A Counterfactual Rule-Driven Question.)*
>
> **### Answer:**

# E    ANALYSIS OF REAL-LIFE ANSWER RATIOS FOR COUNTERFACTUAL TASK

To analyze the ratio of real-life answers in counterfactual reasoning task, we conduct a trend analysis on the models listed in Table 2. Figure 6, 7 below show the trend of change for Chat Model and Base Model. This analysis reveals that as counterfactual accuracy decreases, the real-life answer ratio increases. For chat models, the ratio remains low, not exceeding 15%, while for base models, it rises significantly to 36.8% as accuracy declines.

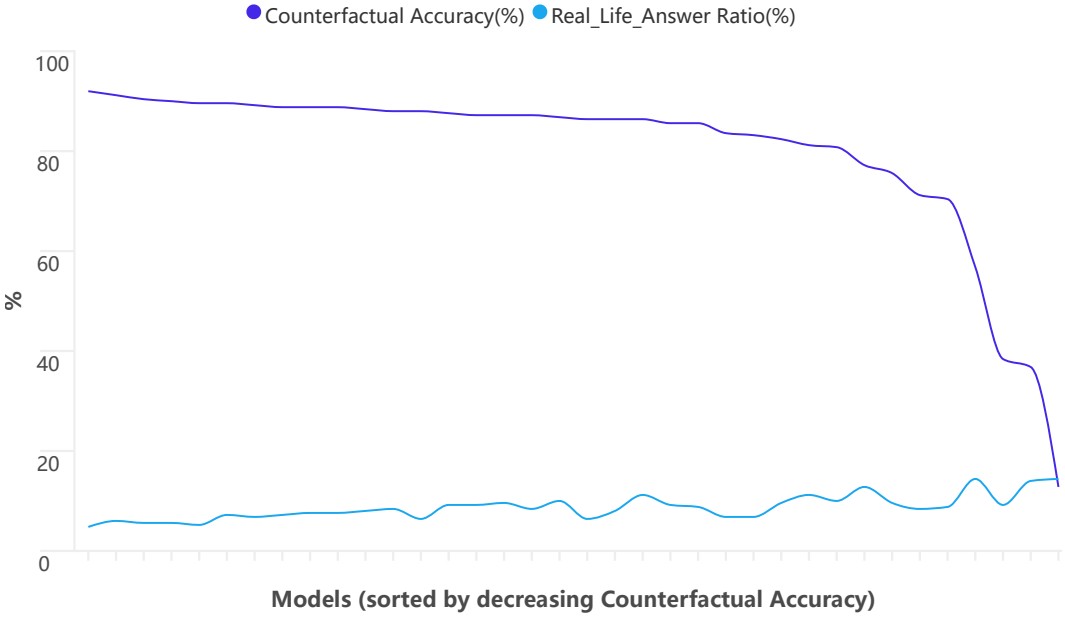

Figure 6: Trend of Real-life Answer Ratio Increasing as Counterfactual Accuracy Decreases for Chat Model.

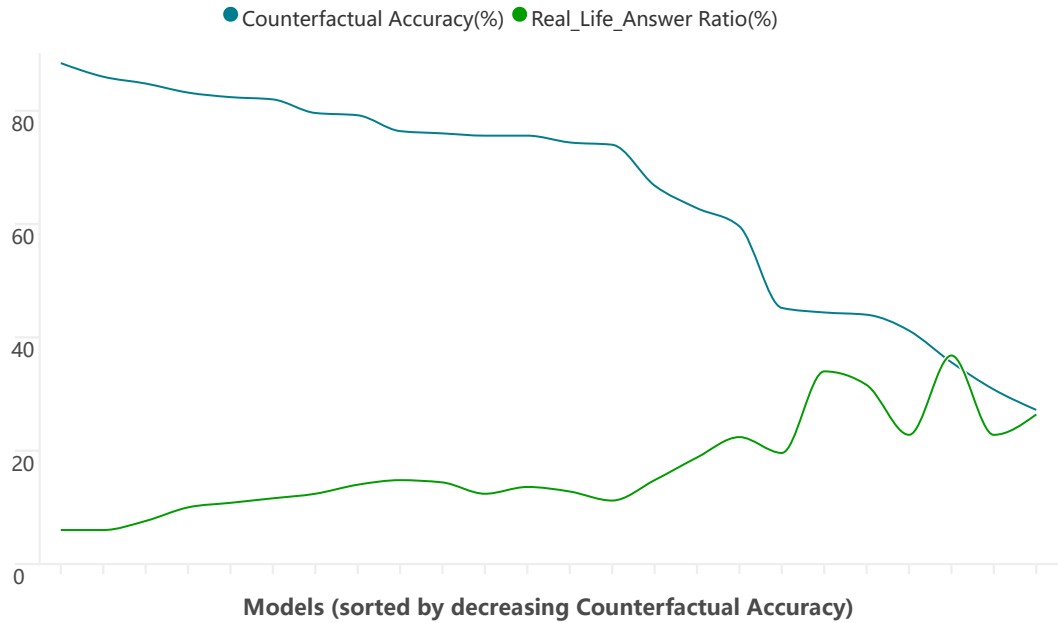

Figure 7: Trend of Real-life Answer Ratio Increasing as Counterfactual Accuracy Decreases for Base Model.

# F    DETAILS OF FURTHER ANALYSIS

Appendix F provides details of supplementary experiments that provide additional insight into the KOR-Bench results.

## F.1    STEPWISE PROMPTING ANALYSIS OF CIPHER TASK BOTTLENECKS

In exploring the bottlenecks of the Cipher Reasoning task, we choose five highly erroneous rules with Rule IDs 1,9,11,18,23, and can check Table 10 to see the cipher rules corresponding to these Rule IDs, and disassemble a Cipher Question step by step by constructing consecutive sub-steps datasets.

### F.1.1    DETAILED EXPLANATIONS OF NINE KEY SUB-STEP TYPES

Table 13 gives a specific explanation of the 9 major types of sub-steps after splitting a Cipher Question into multiple steps.

| Sub-Step Type | Explanation |
|---|---|
| Substitution | Replace one or more symbols with others. |
| Mapping | Map input to a new position or value using predefined tables or rules. |
| Shift | Move data left or right by n characters. |
| Rotation | Rotate data clockwise or counterclockwise. |
| Partition | Divide data into multiple blocks. |
| Conditional Filling | Fill data in a specific order based on certain conditions. |
| Conditional Reading | Read data in a specific order based on conditions. |
| Encoding | Convert data to another format or representation, such as ASCII. |
| Computation | Perform mathematical operations. |

Table 13: **Explanations of Sub-Step Types in Stepwise Prompting Analysis of Cipher Task.** This table gives an explanation of the 9 major types of steps when a Cipher Question is split into multiple steps.

### F.1.2 EXAMPLE OF A CIPHER QUESTION SPLIT INTO SUB-STEPS

The following content F.1.2 illustrates how an encryption or decryption question in the Cipher reasoning task is broken down into several different types of sub-steps.

---

## Cipher Sub-Step Example

**Rule:**

*Encryption Rules:*

*Input:*

Plaintext: Uppercase letters string without punctuation and spaces.

*Output:*

Ciphertext: Uppercase letters string.

*Preparation:*

standard_alphabet:
  "ABCDEFGHIJKLM
  NOPQRSTUVWXYZ"

reversed_alphabet:
  "ZYXWVUTSRQPON
  MLKJIHGFEDCBA"

substitution_alphabet:
  "RFDJUHABCEGIK
  LMNOPQSTVWXYZ"

*Encryption Steps:*

For each letter p in the given Plaintext:

(1) Use reversed_alphabet for reverse mapping. Find its position in the standard_alphabet and replace it with the letter in the corresponding position in reversed_alphabet. For example, A is mapped to Z and B is mapped to Y.

(2) Move the letter obtained in (1) forward 4 places in the standard_alphabet order. For example, if p=A, after (1) is mapped to Z, then Z is shifted forward 4 positions in the standard_alphabet to get D.

(3) Replace the letter obtained from (2) by finding its position in standard_alphabet and using the corresponding letter in substitution_alphabet, resulting in the final ciphertext letter. For example, if the letter obtained by going through (2) is D, it is mapped as J.

**Other Fields:**

**Question**: *Plaintext: "O"*
*Please provide the encrypted answer, encapsulated in double square brackets. For example, the format should be: [[encrypted answer]].*
**Answer**: *[[N]]*

**Steps:**

**Step 1**: *For a letter O in the given Plaintext: (1) Use reversed_alphabet for reverse mapping. Find its position in the standard_alphabet and replace it with the letter in the corresponding position in reversed_alphabet. For example, A is mapped to Z and B is mapped to Y. Please give your answer after performing (1) in the format [[...]].*
**Atom**: *O*
**Answer**: *[[L]]*
**Type**: *Substitution*

**Step 2**: *For a letter O in the given Plaintext: Execute (1) to obtain the letter L. (2) Move the letter obtained in (1) forward 4 places in the standard_alphabet order. For example, if p=A, after (1) is mapped to Z, then Z is shifted forward 4 positions in the standard_alphabet to get D. Please give your answer after performing (2) in the format [[...]].*
**Atom**: *L*
**Answer**: *[[P]]*
**Type**: *Shift*

**Step 3**: *For a letter O in the given Plaintext: Execute (1)(2) to obtain the letter P. (3) Replace the letter obtained from (2) by finding its position in standard_alphabet and using the corresponding letter in substitution_alphabet, resulting in the final ciphertext letter. For example, if the letter obtained by going through (2) is D, it is mapped as J. Please give your answer after performing (3) in the format [[...]].*
**Atom**: *P*
**Answer**: *[[N]]*
**Type**: *Substitution*

### F.1.3 MODEL RESULTS ON SUB-STEP ERROR RATES

In the Stepwise Prompting setting, Figure 8, 9 below gives the error rates of the model on nine key Sub-Steps to reveal its weaknesses in the Cipher Reasoning task.

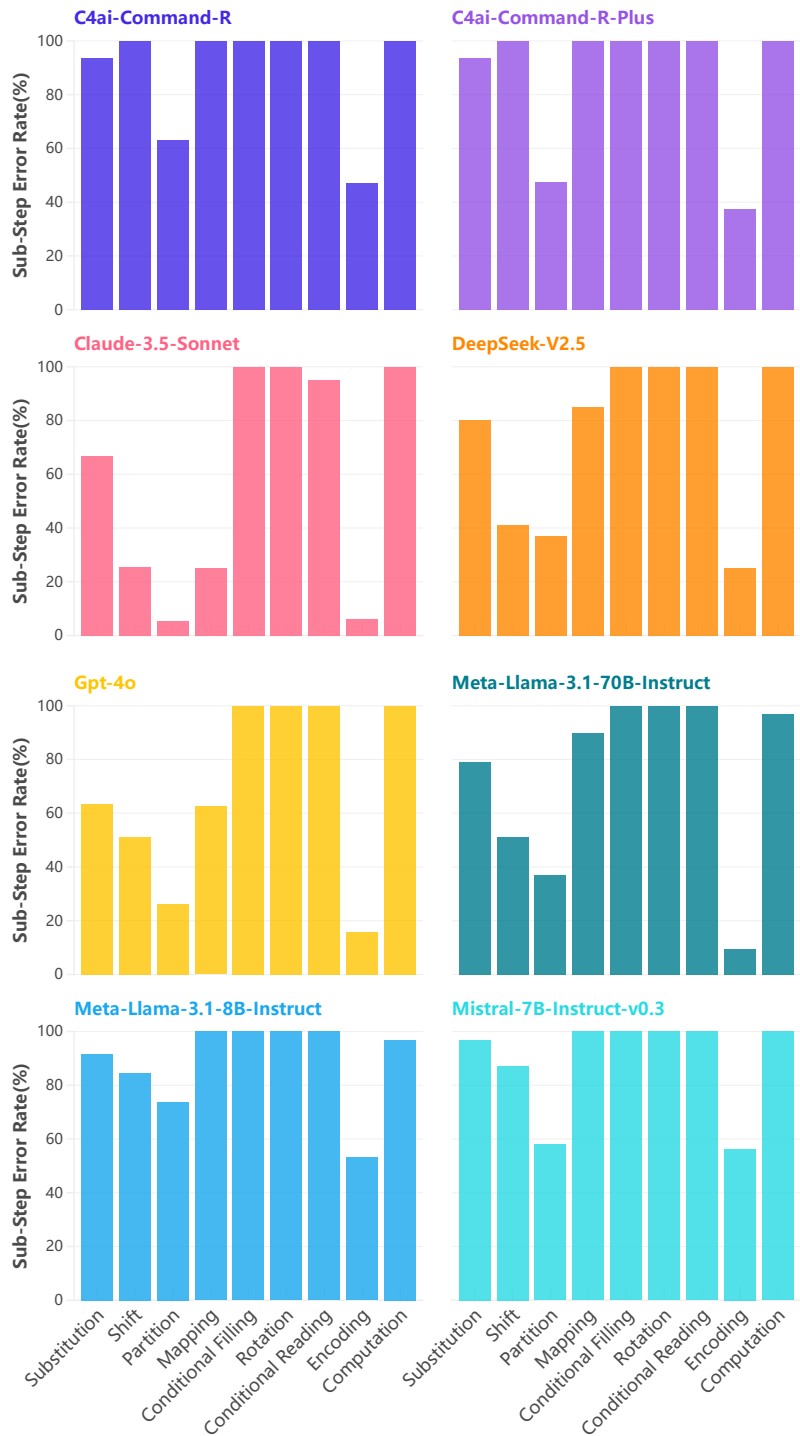

Figure 8: **Model Sub-Step Error Rates in Cipher Task Stepwise Prompting.** This Figure shows the error rates for some models over the nine categories of sub-steps, and the results for the other models are shown in Figure 9.

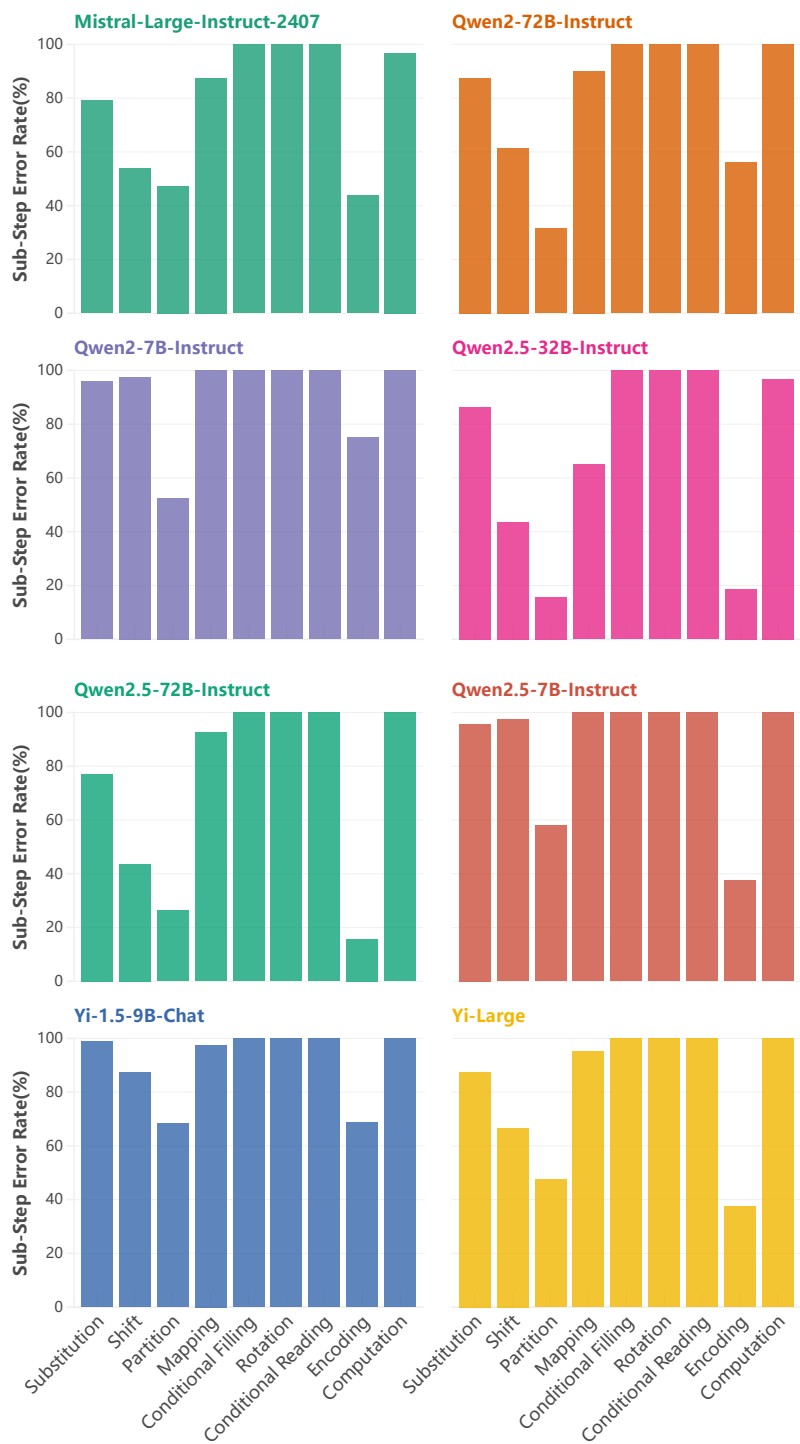

Figure 9: **Model Sub-Step Error Rates in Cipher Task Stepwise Prompting.** This Figure shows the error rates for some models over the nine categories of sub-steps, and the results for the other models are shown in Figure 8.

## F.2 IMPACT ANALYSIS OF TRICKS ON PUZZLE TASK PERFORMANCE

In this experiment, we introduce a "trick" field as additional input to explore its impact on puzzle task performance. For complex puzzle tasks such as mazes and sudoku, identifying and executing key initial steps can often dramatically simplify the entire process. Figure 10 gives some results on the accuracy of the model on the Puzzle task without Trick and with Trick given.

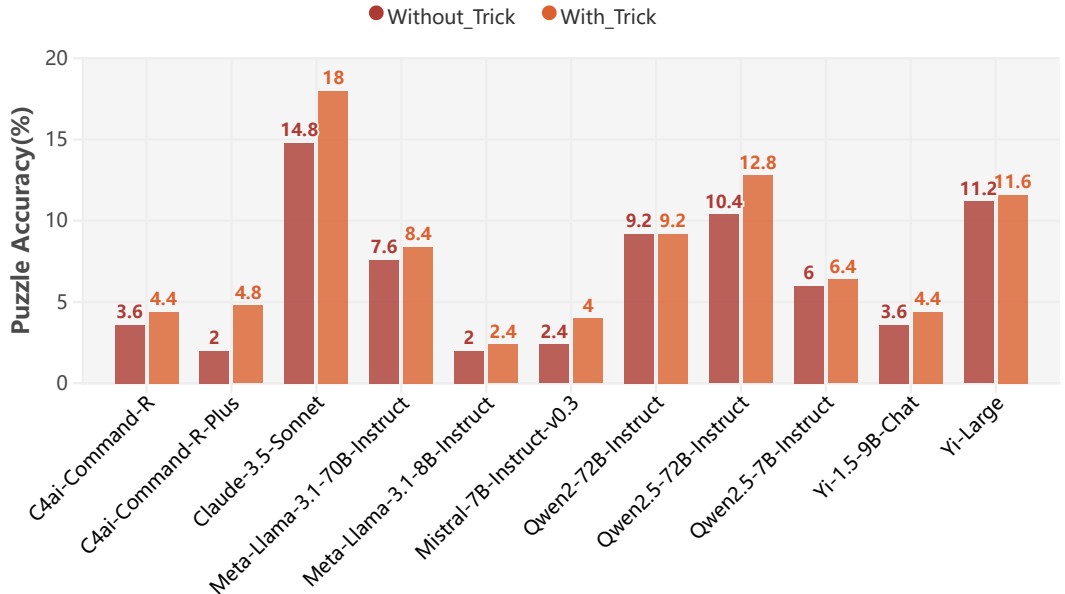

Figure 10: **Impact of Trick Field on Puzzle Task Accuracy.** The figure gives the accuracy without and with Trick for some models that improve in Puzzle Task Accuracy after Trick Field is given.

### F.2.1 A CASE STUDY OF INCORPORATING TRICKS IN PUZZLE REASONING TASK

The following content F.2.1 demonstrates the addition of the trick field in the Puzzle reasoning task, along with an example illustrating the difference in responses from Claude 3.5 Sonnet when the trick is provided versus when it is not. The trick field is appended to the question in the prompt.

---

**Puzzle Trick Example**

**Rule:** *A rectangular grid is given, which is pre-filled with numbers, while each row and column has a number at the end indicating the sum of the numbers in that row or column.*
*The player needs to fill the remaining spaces with a specified set of numbers.*
*Each number can only be used once.*
*The numbers must be filled in such that the sum of the numbers in each row and column matches the intended sum.*
*The puzzle is given in the form of a matrix with blank squares filled with X and the player replaces the squares with numbers.*
**Question:**

$$
\begin{array}{ccccc}
X & 2 & X & 4 & 23 \\
9 & X & X & 11 & 24 \\
X & 12 & X & 8 & 31 \\
21 & 15 & 19 & 23 &
\end{array}
$$

*The final answer should be given in order from left to right, top to bottom with each element separated by a space and different lines separated by ",". Wrap your final answer in double square brackets, like this: [[your answer]].*
**Answer:** *[[7 2 10 4,9 1 3 11,5 12 6 8]]*
**Trick:** *Observe the second column, 2+X+12=15, then the X value of the second row and second column is 1, fill in the matrix and continue the solution.*

---

**Response without Trick:**

*To solve this puzzle, I'll apply the given rule and fill in the blanks (X) with the appropriate numbers to ensure each row and column sums up to the given total. Let's break it down step by step:*
*1. First, let's identify the missing numbers: We need to fill in 5 X's with numbers that aren't already used in the grid. The unused numbers are: 1, 3, 5, 6, 7*
*2. Now, let's solve each row and column: Row 1: X + 2 + X + 4 = 23; We need two numbers that add up to 17 (23 - 2 - 4); The only combination that works is 6 and 11.*
*Row 2: 9 + X + X + 11 = 24; We need two numbers that add up to 4 (24 - 9 - 11); The only combination that works is 1 and 3.*
*Row 3: X + 12 + X + 8 = 31; We need two numbers that add up to 11 (31 - 12 - 8); The only remaining number that fits is 7 and 5.*
*Column 1: X + 9 + X = 21; 6 + 9 + 6 = 21 (confirmed).*
*Column 2: 2 + X + 12 = 15; 2 + 1 + 12 = 15 (confirmed).*
*Column 3: X + X + X = 19; 11 + 3 + 5 = 19 (confirmed).*
*Column 4: 4 + 11 + 8 = 23 (confirmed)*
*3. The completed grid looks like this:*

| | | | | |
|---|---|---|---|---|
| 6 | 2 | 11 | 4 | 23 |
| 9 | 1 | 3 | 11 | 24 |
| 6 | 12 | 5 | 8 | 31 |
| 21 | 15 | 19 | 23 | |

*4. Now, we'll format the answer as requested:*
*[[6 2 11 4, 9 1 3 11, 6 12 5 8]]*

**Response with Trick:**

*Let's approach this step-by-step:*
*1. We start with the observation that in the second column, 2+X+12=15, so X = 1.*
*2. Now let's fill in that 1 and solve the rest of the puzzle:*

| | | | | |
|---|---|---|---|---|
| X | 2 | X | 4 | 23 |
| 9 | 1 | X | 11 | 24 |
| X | 12 | X | 8 | 31 |
| 21 | 15 | 19 | 23 | |

*3. For the first row: X + 2 + X + 4 = 23; We need two numbers that add up to 17 (23 - 2 - 4 = 17); The only unused numbers that satisfy this are 7 and 10.*
*4. For the second row: 9 + 1 + X + 11 = 24; So X must be 3.*
*5. For the third row: X + 12 + X + 8 = 31; We need two numbers that add up to 11 (31 - 12 - 8 = 11); The only unused numbers that satisfy this are 5 and 6.*
*6. Now we can fill in the entire grid:*

| | | | | |
|---|---|---|---|---|
| 7 | 2 | 10 | 4 | 23 |
| 9 | 1 | 3 | 11 | 24 |
| 5 | 12 | 6 | 8 | 31 |
| 21 | 15 | 19 | 23 | |

*7. We can verify that all row and column sums are correct.*
*Therefore, the final answer is:*
*[[7 2 10 4, 9 1 3 11, 5 12 6 8]]*

## F.3 ATTENTION FOCUS VISUALISATION

In KOR-Bench, the rules are text-based descriptions, and each rule-driven question focuses on a specific part of the rule or a particular branch of the rule. Consequently, the model must concentrate on these core elements to have a higher likelihood of answering the question accurately. While focusing on these core elements does not guarantee correct answers, it is a necessary condition.

$$A_{\text{accumulated}}[i] = \sum_{\text{decode\_steps}} \sum_{(layer,head) \in \text{top}_k} \left[ A^{(\text{step})}_{layer,head}[1,i] \text{ if } \text{rule}_{\text{start}} \leq i \leq \text{rule}_{\text{end}} \right] \quad (1)$$

In the experiment, we add a "needle" field to each question-answer sample, indicating the core parts the model needs to focus on during the answering process. We first use the Retrieval Head (Wu et al., 2024) code to calculate the Attention Head Retrieval Score, which ranks each of the model's retrieval heads. Next, we accumulate the attention matrices of the top 50 ranked retrieval heads within the rule's specified range (max_decode_len = 2000), as shown in Equation 1. Finally, we map the attention scores assigned to tokens back to the rule text for visualization, where the "needle" field is underlined, and the intensity of the color indicates the level of attention.

Combining the results of attention visualization helps to understand the model's output better, particularly the reasons for its errors. Appendix H provides several case studies of attention visualizations.

### F.4 ANALYSIS ON SELF-CORRECTION

#### F.4.1 SELF CORRECTION PROMPT TEMPLATE

The following content F.4.1 gives a prompt template for the Self-Correction experiment. Due to the limit on the number of tokens, in each round of dialogue, the model's responses are extracted and reintegrated into the dialogue history. The maximum number of interactions is 5, implying 4 correction rounds.

---

**Self-Correction Prompt Template**

**Round 0:**
**User:** *(The default prompt includes the following components:)*
{Rule}
{Question}
**Assistant:** {Extracted_Response_Round_0}

**Round 1:**
**User:** Your answer is incorrect, please check your answer and provide a correct one.
**Assistant:** {Extracted_Response_Round_1}

**Round 2:**
**User:** Your answer is incorrect, please check your answer and provide a correct one.
**Assistant:** {Extracted_Response_Round_2}

*(Subsequent rounds omitted...)*

---

#### F.4.2 IMPACT OF ROUND COUNT ON SELF-CORRECTION ACCURACY

Figure 11 below gives the effect of the number of rounds on the rate of correction from the model's point of view, and it can be seen that for most models, self-correcting for two rounds gives the highest gains.

Figure 12 displays the correction rates categorized by task type. The counterfactual reasoning task maintains a higher level of correction rates across all four rounds. In contrast, the other four task types show a significant correction effect in the first and second rounds, with less pronounced effects subsequently.

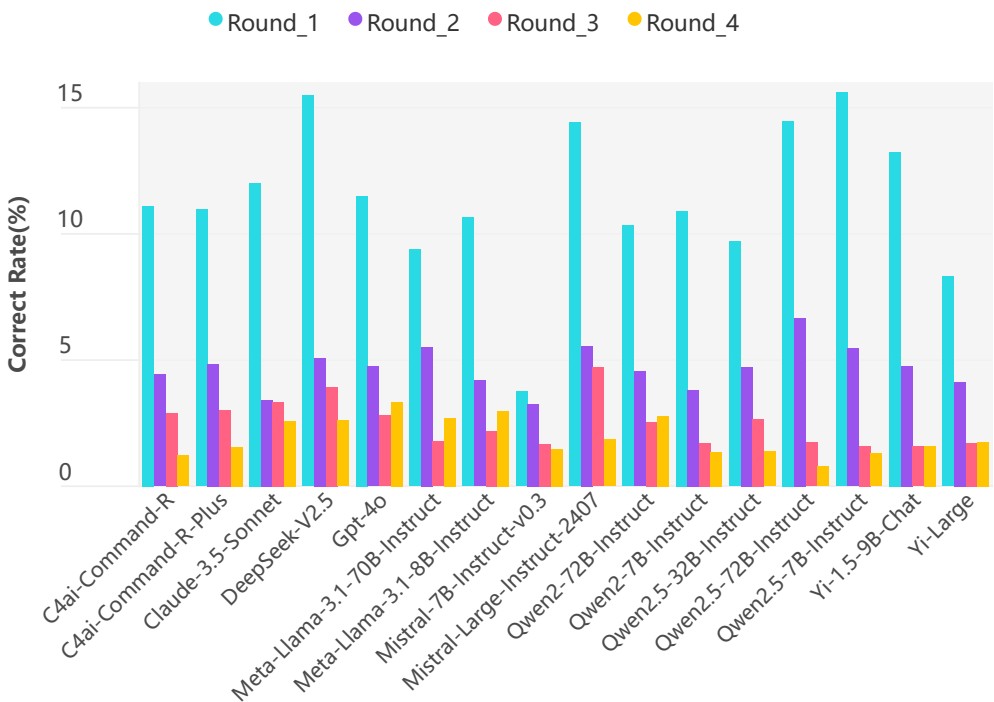

Figure 11: **Self-Correction Rates of Multiple Models Over Four Rounds.**The self-correction rate is defined as the number of problems successfully corrected in a given round divided by the number of problems still unresolved from the previous round.

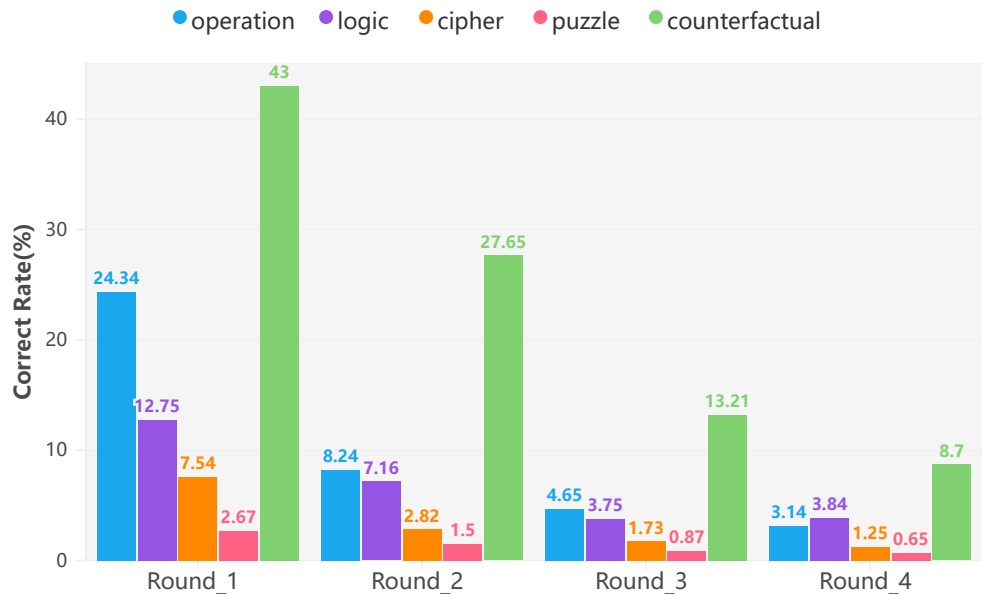

Figure 12: **Self-Correction Rates of Five Task Types Over Four Rounds.**The self-correction rate is defined as the number of problems successfully corrected in a given round divided by the number of problems still unresolved from the previous round.

### F.5  ANALYSIS ON COMPLEX TASK PROCESSING

#### F.5.1  COMPLEX TASK PROCESSING PROMPT

Content F.5.1 gives a prompt template for complex task processing. Three Settings for complex task processing:(1) Multi-Q: 1 rule, 1-10 questions; (2) Multi-R: 2-3 rules, 1 question; (3) Multi-RQ: 2-3 rules, 1-3 questions all use this template.

---

**Complex Task Processing Prompt Template**

You are an intelligent assistant capable of handling all types of reasoning and problem-solving tasks. Below is the text of a set of rules. Your task is to apply the appropriate rules to solve a series of problems.

**### Instructions:**

1. Read each question carefully and rules to find something relevant to that question.

2. Use the relevant rules to answer each question accurately.

3. Provide the final answers to all questions in JSON format.

```
{{
"question1": "your answer",
"question2": "your answer",
"question3": "your answer",
}}
```

**### Rules:**
*(A series of rules.)*

**### Questions:**
*(A series of questions.)*

**### Answers:**

---

#### F.5.2  COMPLEX TASK PROCESSING EVALUATION

For the evaluation of complex task processing, since the output format of the answers in all three settings is JSON, the system first uses the regular expression r'{.*}' to extract the answer portion. Next, it traverses the question list and, for each sub-question, extracts its category and index to retrieve the specific question content, including the correct answer, from the corresponding data source. The system then conducts an assessment of each sub-question, individually judging and recording the passing status of each one. Finally, the overall pass rate of the question is calculated based on the number of sub-questions that pass. If the length of the response results is less than the number of sub-questions, the evaluation process terminates promptly. All assessment results for the sub-questions, including the question content, response text, correct answer, and correctness, are systematically recorded.

# G  FUN-FILLED ANALYSIS OF SLIP-UPS

Appendix G presents a series of interesting error cases occurring in five types of tasks from the Claude-3.5-Sonnet or GPT-4o. Each case is accompanied by detailed analyses and in-depth explanations designed to reveal potential weaknesses in the model and provide insights into improving model performance.

**List of Case Study Boxs**

### G.1 OPERATION ERROR CASES ANALYSIS

> ⚠️ **Error Case 1**
>
> **Rule:**
>
> *Define an operation such that when a is a multiple of b, a✳b = a/b + 2.*
>
> *When b is a multiple of a, a✳b = b/a + 2.*
>
> *If a is not a multiple of b and b is not a multiple of a, a✳b = 24.*
>
> *Both a and b are integers.*
>
> **Question:**
>
> *Compute 25✳5✳14.*
>
> *Please ensure the answer is a single number and wrap it in double square brackets, like this: [[your answer]].*
>
> **Answer:**
>
> *[[4]]*
>
> ---
>
> **Response:**
>
> *Let's break this down step by step:*
>
> 1. *First, we need to calculate 25✳5:*
>
>    *25 is a multiple of 5, so we use the first part of the rule.*
>    *25✳5 = 25/5 + 2 = 5 + 2 = 7.*
>
> 2. *Now we have 7✳14:*
>
>    *7 is not a multiple of 14, and 14 is not a multiple of 7.*
>    *So we use the third part of the rule.*
>    *7✳14 = 24.*
>
> 3. *Therefore, 25✳5✳14 = 24*
> 4. *The final answer, wrapped in double square brackets, is: [[24]]*
>
> ---
>
> **Analysis:**
>
> The sentence "7 is not a multiple of 14, and 14 is not a multiple of 7" is wrong. In fact, 14 is a multiple of 7. The misjudgement of the multiplicity has led to the wrong choice of using the third part of the rule.

## ⚠ Error Case 2

**Rule:**

*Define an operation such that when a is a multiple of b, a⁂b = a/b + 2.*

*When b is a multiple of a, a⁂b = b/a + 2.*

*If a is not a multiple of b and b is not a multiple of a, a⁂b = 24.*

*Both a and b are integers.*

**Question:**

*If 25⁂X⁂14=4, find X.*

*When providing your answer, please enclose it in double square brackets, like this: [[answer]].*

*If there is more than one correct answer, separate the answers with 'or', like this: [[1or2]].*

**Answer:**

*[[650or5or125]]*

---

**Response:**

*Let's approach this step-by-step:*

*1) We know that 25⁂X⁂14 = 4*

*2) Let's consider the possible values for X that could make this equation true.*

*3) For the final result to be 4, we need either:*

*a) 25⁂X = 2 and 2⁂14 = 4, or*

*b) 25⁂X = 50 and 50⁂14 = 4*

*4) Let's examine case a:*

*For 2⁂14 to equal 4, 14 must be a multiple of 2, which is true.*

*14/2 + 2 = 7 + 2 = 9, not 4. So this case doesn't work.*

*5) Let's examine case b:*

*For 50⁂14 to equal 4, 14 must be a multiple of 50, which is not true.*

*So 50⁂14 = 24, which is not 4. This case doesn't work either.*

*6) Therefore, we need to find X such that 25⁂X = 50*

*7) For this to be true, X must be a multiple of 25, and 25/X + 2 = 50*

*8) Solving this equation:*

*25/X + 2 = 50*

*25/X = 48*

*X = 25/48 = 0.52083...*

*9) But X must be an integer. The only integer that satisfies all conditions is 25.*

*10) Let's verify:*

*25⁂25 = 25/25 + 2 = 1 + 2 = 3*

*3⁂14 = 24 (since neither is a multiple of the other)*

*Therefore, the value of X is 25.*

*[[25]]*

---

**Analysis:**

*The analysis is fundamentally flawed due to unreasonable initial assumptions about the two scenarios.*        **Back to List of Cases**     |     **Back to Table of Contents**

## ⚠ Error Case 3

**Rule:**

$A \bigcirc B = (A+3B) \times (A+B)$

A and B are integers.

**Question:**

Compute $2 \bigcirc 3 \bigcirc 4$.

Please ensure the answer is a single number and wrap it in double square brackets, like this: [[your answer]].

**Answer:**

[[3953]]

---

**Response:**

Let's approach this step-by-step:

1) First, we need to calculate $3 \bigcirc 4$:

$3 \bigcirc 4 = (3+3 \times 4) \times (3+4)$

$= (3+12) \times 7$

$= 15 \times 7$

$= 105$

2) Now we have $2 \bigcirc 105$:

$2 \bigcirc 105 = (2+3 \times 105) \times (2+105)$

$= (2+315) \times 107$

$= 317 \times 107$

$= 33,919$

Therefore, the final answer is:

[[33919]]

---

**Analysis:**
The response demonstrates a critical error in the order of operations. The calculation should proceed from left to right, but the answer incorrectly prioritizes the rightmost operation.

Back to List of Cases | Back to Table of Contents

## ⚠ Error Case 4

**Rule:**

$a¢b = \log_b a + \log_a b$

*a and b are positive integers.*

**Question:**

*Compute 9¢243.*

*If the answer is a fraction, write it in 'a/b' text format. Decimals are not allowed.*

*Please wrap the answer in double square brackets, like this: [[your answer]].*

**Answer:**

*[[29/10]]*

---

**Response:**

*Let's apply the given operation rule to compute 9¢243:*

$a¢b = \log_b a + \log_a b$

*Here, $a = 9$ and $b = 243$*

*9¢243 $= \log_{243} 9 + \log_9 243$*

*1) $\log_{243} 9 = 1/2$ (since 243^(1/2) = 9^1)*

*2) $\log_9 243 = 5/2$ (since 9^(5/2) = 243)*

*Adding these:*

*9¢243 $= 1/2 + 5/2 = 3/2$*

*Therefore, the answer is:*

*[[3/2]]*

---

**Analysis:**
*The response contains a partial calculation error that propagates through the solution.*

## ⚠ Error Case 5

**Rule:**

$a ¢ b = \log_b a + \log_a b$

*a and b are positive integers.*

**Question:**

*If 3¢X=17/4, find X.*

*The answer should only be given as a number.*

*Please wrap the answer in double square brackets, like this: [[your answer]].*

**Answer:**

*[[81]]*

---

**Response:**

*Let's approach this step-by-step:*

*1) We're given that $3 ¢ X = 17/4$*

*2) Using the definition of the ¢ operation, we can write:*

$3 ¢ X = \log_X 3 + \log_3 X = 17/4$

*3) We need to solve this equation for X. This is a complex equation that doesn't have a simple algebraic solution.*

*4) However, we can use trial and error with some logical deduction. We know X must be a positive integer.*

*5) Let's try $X = 9$:*

$\log_9 3 + \log_3 9 \approx 0.5 + 2 = 2.5$

*6) This is very close to $17/4 = 4.25$*

*7) We can verify that no other integer value of X gives a result closer to 17/4.*

*Therefore, the value of X that satisfies the equation $3 ¢ X = 17/4$ is 9.*

*[[9]]*

---

**Analysis:**

*The response attempts to solve the problem through trials. After testing, it correctly identifies that the proposed solutions do not meet the conditions. However, it presents this answer as the final solution.*

## G.2 LOGIC ERROR CASES ANALYSIS

> ### ⚠ Error Case 1
>
> **Rule:**
>
> *Propositions are represented using $p_1, p_2, \ldots, p_n$.*
>
> *Let $p_1$ be a proposition, the compound proposition "not $p_1$" is represented as $\sim p_1$.*
>
> *Let $p_1$ and $p_2$ be two propositions, the compound proposition "$p_1$ and $p_2$" is represented as $p_1 \& p_2$.*
>
> *Let $p_1$ and $p_2$ be two propositions, the compound proposition "$p_1$ or $p_2$" is represented as $p_1 \parallel p_2$.*
>
> *Let $p_1$ and $p_2$ be two propositions, the compound proposition "if $p_1$, then $p_2$" is represented as $p_1 \Rightarrow p_2$.*
>
> *Let $p_1$ and $p_2$ be two propositions, the compound proposition "$p_1$ if and only if $p_2$" is represented as $p_1 = p_2$.*
>
> *A single proposition and proposition constants can be called a formula.*
>
> *Formulas are represented using $F_1, F_2, \ldots, F_n$.*
>
> *If $F_1$ is a formula, then $\sim F_1$ is also a formula.*
>
> *If $F_1$ and $F_2$ are formulas, then $F_1 \& F_2$, $F_1 \parallel F_2$, $F_1 \Rightarrow F_2$, $F_1 = F_2$ are also formulas.*
>
> ***Level A Formula:*** *The most basic proposition unit, without logical connectives or nested structures.*
>
> ***Level B Formula:*** *A formula containing one logical connective, and the connected two propositions are both Level A formulas. For example, $p_1$.*
>
> ***Level C Formula:*** *A formula containing nested logical connectives and at least one Level B formula. For example, $\sim p_1$.*
>
> *Other levels of logic follow by analogy; when higher than Level Z, they are classified as Z+n ($n \geq 1$). For example, $\sim (\sim p_1)$.*
>
> ***True assignment of a proposition:*** *A proposition $p_1$ is assigned as $\checkmark$, indicating that $p_1$ is true.*
>
> ***False assignment of a proposition:*** *A proposition $p_1$ is assigned as $\times$, indicating that $p_1$ is false.*
>
> ***True assignment of a formula:*** *If the formula is $(\sim p_1 \& \sim p_2 \& \sim p_3) \parallel (p_1 \& p_2)$, then $\times|\times|\times, \checkmark|\checkmark|\times$ are true assignments of the formula.*
>
> ***False assignment of a formula:*** *If the formula is $(\sim p_1 \& \sim p_2 \& \sim p_3) \parallel (p_1 \& p_2)$, then $\times|\times|\checkmark, \times|\checkmark|\checkmark$ are false assignments of the formula.*
>
> *For $p_1 \Rightarrow p_2$, only $\checkmark|\times$ is a false assignment of the formula.*
>
> *A formula that is true under all assignments is called a **Truth Formula**.*
>
> *A formula that is false under all assignments is called a **Falsehood Formula**.*
>
> ***Recursive definition of formulas:*** *Any formula containing nested logical connectives can be decomposed recursively to obtain its subformulas and their logical connective structures.*
>
> ***Priority of logical connectives:*** *The priority of logical connectives from high to low is as follows: $\sim$ (not), $\&$ (and), $\parallel$ (or), $\Rightarrow$ (if...then), $=$ (if and only if).*
>
> *Without parentheses, operations are performed according to priority.*
>
> ***Equivalence of formulas:*** *Two formulas are equivalent if they have the same truth value under all assignments.*
>
> *Equivalent formulas can be interchanged.*

> ***Simplification of formulas:*** *Formulas can be simplified through logical rules to obtain a more concise form without changing the truth value of the formula.*

**Question:**

> *Find the falsity values of the following formulas:*
>
> *(1)* $(\sim (\sim p_1 \wedge p_2)) \vee \sim p_3$
>
> *(2)* $(\sim p_2 \vee p_3) \wedge (p_1 \Rightarrow p_2)$
>
> *(3)* $(p_1 \Rightarrow p_2) \wedge (\sim (p_1 \wedge p_3) \vee p_1)$
>
> *The answer format is [[F:...];[F:...];[F:...]].*
> *If there are multiple values in F, they should be separated by commas (,). For example* [[F:×| × |×];[F:✓|✓|✓];[F:×| × |×,✓|✓|✓]]

**Answer:**

> [[F:×|✓|✓];[F:×|✓|×,✓|✓|×,✓| × |✓,✓| × |×];[F:✓| × |×,✓| × |✓]]

---

**Response:**

> *To find the falsity values of the given formulas, we need to evaluate each formula for all possible truth value assignments and identify the assignments that make the formula false. Let's analyze each formula:*
>
> *(1)* $\sim(\sim p_1 \wedge p_2) \vee \sim p_3$
>
> *This formula is false when $(\sim p_1 \wedge p_2)$ is true and p3 is true.*
>
> *So, it's false when p1 is false, p2 is true, and p3 is true.*
>
> *(2)* $((\sim p_2 \vee p_3) \wedge (p_1 \Rightarrow p_2)$
>
> *This formula is false when either $(\sim p_2 \vee p_3)$ is false or $(p_1 \Rightarrow p_2)$ is false.*
>
> $(\sim p_2 \vee p_3)$ *is false when $p_2$ is true and $p_3$ is false.*
>
> $(p_1 \Rightarrow p_2)$ *is false when $p_1$ is true and $p_2$ is false.*
>
> *So, it's false when $p_1$ is true, $p_2$ is true, and $p_3$ is false, or when $p_1$ is true, $p_2$ is false, and $p_3$ is any value.*
>
> *(3)* $(p_1 \Rightarrow p_2) \wedge (\sim (p_1 \wedge p_3) \vee p_1)$
>
> *This formula is false when $(p_1 \Rightarrow p_2)$ is false or $(\sim (p_1 \wedge p_3) \vee p_1)$ is false.*
>
> $(p_1 \Rightarrow p_2)$ *is false when $p_1$ is true and $p_2$ is false.*
>
> $(\sim (p_1 \wedge p_3) \vee p_1)$ *is always true (it's a tautology), so it doesn't contribute to making the formula false.*
>
> *So, the formula is false when $p_1$ is true, $p_2$ is false, and $p_3$ can be any value.*
>
> *Now, let's format the answer according to the specified format:*
>
> [[F:×|✓|✓]; [F:✓|✓|×,✓| × |✓,✓| × |×]; [F:✓| × |✓,✓| × |×]]

---

**Analysis:**
*The model accurately identifies two different scenarios during the analysis, which should theoretically result in four possible answers. However, the model fails to fully cover all scenarios in its final consideration, a phenomenon that reveals its inadequacy in exploring all possible paths in terms of logical reasoning.*

## ⚠ Error Case 2

**Rule:**

*In a simple conjunctive form (simple disjunctive form) containing $n$ propositional variables, if each propositional variable and its negation appear exactly once, and the propositional variables or their negations are arranged in ascending order of subscripts or in lexicographical order, such a simple conjunctive form (simple disjunctive form) is called a paired conjunctive term (paired disjunctive term).*

*If the true assignment of a paired conjunctive term corresponds to a binary number equal to hexadecimal number $i$, this paired conjunctive term is denoted as $m_i$. For example, the true assignment of $p \wedge q$ is 11, and the binary number is 11, corresponding to hexadecimal number 3, denoted as $m_3$.*

*If the false assignment of a paired disjunctive term corresponds to a binary number equal to hexadecimal number $i$, this paired disjunctive term is denoted as $M_i$ .For example, the false assignment of $\neg p \vee \neg q \vee \neg r$ is 111, and the binary number is 111, corresponding to hexadecimal number 7, denoted as $M_7$.*

*The disjunctive normal form (conjunctive normal form) consisting of all paired conjunctive terms (paired disjunctive terms) is called the principal disjunctive normal form (principal conjunctive normal form).*

*Given a formula A containing $n$ propositional variables:*

- *If the principal disjunctive normal form of A includes all $2^n$ paired conjunctive terms, A is a tautology.*
- *If the principal disjunctive normal form of A includes no paired conjunctive terms, A is a contradiction.*
- *If the principal disjunctive normal form of A includes $m_0$, A is a basic formula.*
- *If the indices $i$ of the paired conjunctive terms included in the principal disjunctive normal form of A are all even, A is an all-even formula.*
- *If the indices $i$ of the paired conjunctive terms included in the principal disjunctive normal form of A are all odd, A is an all-odd formula.*

**Question:**

*Given that formula A contains 4 propositional variables,*
*what should it be denoted as if it is both a tautology and a basic form? The answer format is [[]].*

**Answer:**

$[[m_0 \vee m_1 \vee m_2 \vee m_3 \vee m_4 \vee m_5 \vee m_6 \vee m_7 \vee m_8 \vee m_9 \vee m_A \vee m_B \vee m_C \vee m_D \vee m_E \vee m_F]]$

---

**Response:**

*To answer this question, let's apply the given logic rule:*

1. *Formula A contains 4 propositional variables.*
2. *For it to be a tautology, its principal disjunctive normal form must include all 2^4 = 16 paired conjunctive terms.*
3. *For it to be a basic formula, its principal disjunctive normal form must include m0.*

*Given these conditions, the formula A would be denoted as:*

$[[m_0 \vee m_1 \vee m_2 \vee m_3 \vee m_4 \vee m_5 \vee m_6 \vee m_7 \vee m_8 \vee m_9 \vee m_{10} \vee m_{11} \vee m_{12} \vee m_{13} \vee m_{14} \vee m_{15}]]$

*This notation includes all 16 paired conjunctive terms (m0 to m15), satisfying the condition for a tautology, and it includes m0, satisfying the condition for a basic formula.*

**Analysis:**
*The model may not be sufficiently limited in its understanding of the rules to accurately capture the specific requirement of 'hexadecimal', leading to errors.*

## ⚠ Error Case 3

**Rule:**

*Between propositions p1 and p2, the representation is defined as follows:*

*A:* $\forall p_1 \rightarrow p_2$

*E:* $\forall p_1 \rightarrow \neg p_2$

*I:* $\exists p_1 \rightarrow p_2$

*O:* $\exists p_1 \rightarrow \neg p_2$

***The figures and moods of the syllogism are as follows:***

1. *Figure I*
   (1) *Form:*
   > *M()P*
   > *S()M*
   > *∴ S()P*
   (2) *Valid Moods:*
   > *AAA*
   > *EAE*
   > *AII*
   > *EIO*

2. *Figure II*
   (1) *Form:*
   > *P()M*
   > *S()M*
   > *∴ S()P*
   (2) *Valid Moods:*
   > *AEE*
   > *EAE*
   > *EIO*
   > *AOO*

3. *Figure III*
   (1) *Form:*
   > *M()P*
   > *M()S*
   > *∴ S()P*
   (2) *Valid Moods:*
   > *AII*
   > *EIO*
   > *IAI*
   > *OAO*

4. *Figure IV*
   (1) *Form:*
   > *P()M*
   > *M()S*
   > *∴ S()P*
   (2) *Valid Moods:*
   > *AEE*
   > *EIO*
   > *IAI*

**Question:**

*Given the logical statements:*

$\forall P \to M, \forall M \to \neg S \Rightarrow \forall S \to \neg P$

*Determine which figure and mood of syllogism the argument corresponds to, using the format [[I/II/III/IV];[Mood]].*

**Answer:**

*[[IV];[AEE]]*

---

**Response:**

*To determine the figure and mood of the syllogism for the given logical statements, let's analyze them step by step:*

*We have:*

*First premise: $\forall P \Rightarrow M$*

*Second premise: $\forall M \Rightarrow \neg S$*

*Conclusion: $\forall S \Rightarrow \neg P$*

*Let's match these to our syllogism figures:*

*The first premise has P as the subject and M as the predicate.*
*The second premise has M as the subject and S as the predicate (negated).*
*The conclusion has S as the subject and P as the predicate (negated).*

*This structure matches Figure I:*

*M()P*

*S()M*

*∴S()P*

*Now, let's determine the mood:*

*First premise: $\forall P \Rightarrow M$ - This is an A statement*

*Second premise: $\forall M \Rightarrow \neg S$ - This is an E statement*

*Conclusion: $\forall S \Rightarrow \neg P$ - This is an E statement*

*5. So, the mood is AEE.*

*6. Checking the valid moods for Figure I, we see that AEE is not listed as a valid mood.*

*However, the question asks us to determine which figure and mood the argument corresponds to, even if it's not valid.*

*Therefore, the answer in the requested format is:*

*[[I];[AEE]]*

*This indicates that the argument corresponds to Figure I with the mood AEE, although it's worth noting that this is not a valid syllogism in classical logic.*

---

**Analysis:**
*After a matching error occurs in the early stage, although it is eventually found to be unsatisfactory, an incorrect conclusion is still given without correction.*

## ⚠ Error Case 4

**Rule:**

*Between propositions p1 and p2, the representation is defined as follows:*

*A:* $\forall p_1 \rightarrow p_2$

*E:* $\forall p_1 \rightarrow \neg p_2$

*I:* $\exists p_1 \rightarrow p_2$

*O:* $\exists p_1 \rightarrow \neg p_2$

***The figures and moods of the syllogism are as follows:***

1. *Figure I*
   (1) *Form:*
   > M()P
   > S()M
   > $\therefore$ S()P
   (2) *Valid Moods:*
   > AAA
   > EAE
   > AII
   > EIO
2. *Figure II*
   (1) *Form:*
   > P()M
   > S()M
   > $\therefore$ S()P
   (2) *Valid Moods:*
   > AEE
   > EAE
   > EIO
   > AOO
3. *Figure III*
   (1) *Form:*
   > M()P
   > M()S
   > $\therefore$ S()P
   (2) *Valid Moods:*
   > AII
   > EIO
   > IAI
   > OAO
4. *Figure IV*
   (1) *Form:*
   > P()M
   > M()S
   > $\therefore$ S()P
   (2) *Valid Moods:*
   > AEE
   > EIO
   > IAI

**Question:**

*Please fill in the parentheses below*

*()AP*

*MI()*

*∴ SIP*

*Provide the answer in the format of [[ ];[ ]].*

**Answer:**

*[[M];[S]]*

---

**Response:**

*To answer this question, we need to identify the figure and mood of the syllogism based on the given information and the valid moods for each figure.*

*Let's analyze the given syllogism:*

*()AP*

*MI()*

*∴ SIP*

*This structure matches Figure IV:*

*P()M*

*M()S*

*∴ S()P*

*Now, we need to find a valid mood for Figure IV that results in a conclusion of SIP. Looking at the valid moods for Figure IV (AEE, EIO, IAI), we can see that IAI is the only one that gives us an I conclusion.*

*So, we'll use the IAI mood:*

*I: $\exists p_1 \Rightarrow p_2$ (for the first premise)*

*A: $\forall p_1 \Rightarrow p_2$ (for the second premise)*

*I: $\exists p_1 \Rightarrow p_2$ (for the conclusion)*

*Applying this to our syllogism:*

*IAP*

*MAI*

*∴ SIP*

*Therefore, the answer in the requested format is:*

*[[I];[A]]*

*This fills in the parentheses in the original syllogism as follows:*

*(I)AP*

*MA(I)*

*∴ SIP*

---

**Analysis:**

*The mismatch arises from an analytical oversight, likely due to misinterpreting the graphical representation, which results in a failure to accurately discern the logical relationship's correct graphical counterpart.*

## ⚠ Error Case 5

**Rule:**

*Universal Quantifier: Use Ax to denote "for all x".*

*Existential Quantifier: Use Ex to denote "there exists some x".*

*Logical Connectives:*

*Conjunction: Use &*
*Disjunction: Use |*
*Implication: Use ⇒*
*Negation: Use ∼*

*In general, a predicate P with n (n > 1) individual variables is called an **n-ary predicate**, denoted as $P(x_1, x_2, ..., x_n)$. When n = 1, P(x) denotes the property P; when n ≥ 2, $P(x_1, x_2, ..., x_n)$ denotes the relationship P among $x_1, x_2, ..., x_n$.*

*Predicates without individual variables are called **0-ary predicates**. For example, F(a), G(a, b), $P(a_1, ..., a_n)$ are all 0-ary predicates.*

*Let D be the domain of individuals.*

- *"All x in D have property F" is symbolized as AxF(x).*
- *"Some x in D have property F" is symbolized as ExF(x).*
- *"For all x in D, if x has property F, then x has property G" is symbolized as Ax(F(x) ⇒ G(x)).*
- *"Some x in D have both properties F and G" is symbolized as Ex(F(x) & G(x)).*
- *"For all x, y in D, if x has property F and y has property G, then x and y have relationship H" is symbolized as AxAy(F(x) & F(y) ⇒ H(x, y)).*
- *"For all x in D, if x has property F, then there exists some y with property G such that x and y have relationship H" is symbolized as Ax(F(x) ⇒ Ey(G(y) & H(x, y))).*
- *"There exists some x in D with property F, and for all y in D, if y has property G, then x and y have relationship H" is symbolized as Ex(F(x) & Ay(G(y) ⇒ H(x, y))).*

**Question:**

*In first-order logic, symbolize the following propositions using* `0-ary predicates` *:*

*(1) Only 2 is a prime number, 4 is a composite number.*

*(2) If 5 is greater than 4, then 4 is greater than 6.*

*For (1), define a unary predicate F(x): x is a prime number.*

*The proposition can be symbolized as?*

*For (2), define a binary predicate G(x, y): x > y.*

*The proposition can be symbolized as?*

*Please provide the answers in the format [[];[]].*

**Answer:**

*$[[F(4) \Rightarrow F(2)];[G(5,4) \Rightarrow G(4,6)]]$*

---

**Response:**

*Based on the given logic rules and the question, here's the symbolization of the propositions:*

*For (1):*

*Let F(x) be the unary predicate "x is a prime number"*

*Let C(x) be the unary predicate "x is a composite number"*

*The symbolization is: $[F(2) \land C(4) \land \forall z((F(z) \Rightarrow z = 2))]$*

*For (2):*

*Let G(x, y) be the binary predicate "x > y"*

*The symbolization is: $[G(5, 4) \Rightarrow G(4, 6)]$*

*Therefore, the complete answer in the requested format is:*

*$[[F(2) \land C(4) \land \forall z((F(z) \Rightarrow z = 2))]; [G(5, 4) \Rightarrow G(4, 6)]]$*

---

**Analysis:**
*Fails to accurately follow the restriction that the topic requires the use of 0-ary predicates to symbolise propositions. Reflects a deficiency in the model's ability to correctly apply the rules of logical symbolisation.*

**Back to List of Cases**    |    **Back to Table of Contents**

### G.3 Cipher Error Cases Analysis

---

> ⚠️ **Error Case 1**

**Rule:**

*Encryption Rules:*

*Input:*

* *Plaintext: Uppercase letters string without punctuation and spaces.*
* *period: Defines how often the inner disc rotates. Periodicity indicates that after every number of characters processed in the encryption process, the inner disc will rotate once according to the incremental value.*
* *increment: Defines the number of characters the inner disc rotates each time. At the end of each cycle, the inner disc will rotate to the right by the corresponding number of characters based on the increment value. For example, if the increment is 4, the inner disc will rotate 4 characters to the right for each cycle that passes (e.g., 5 characters are processed).*

*Output:*

* *Ciphertext: Uppercase letters string.*

*Preparation:*

* *outer_disk = "QWERTYUIOPASDFGHJZXCVBNMKL"*
* *inner_disk = "JKLZXCVBNMASDFGHJQWERTYUIO"*

*Encryption Steps:*

* *For each character p in the plaintext:*
    * *Find the character in the outer_disk.*
    * *Replace it with the character at the corresponding position on the inner_disk.*
    * *After encrypting every 'period' characters, rotate the inner_disk to the right by 'increment' characters.For example, rotating 'ZXCVBN-MASDFGHJKLQWERTYUIOP' by 4 gives 'BNMASDFGHJKLQWER-TYUIOPZXCV'.*

*Decryption Rules:*

*Input:*

* *Ciphertext: Uppercase letters string.*
* *period (Same as encryption)*
* *increment (Same as encryption)*

*Output:*

* *Plaintext: Uppercase letters string.*

*Preparation:*

* *outer_disk = "QWERTYUIOPASDFGHJZXCVBNMKL"*
* *inner_disk = "JKLZXCVBNMASDFGHJQWERTYUIO"*

*Decryption Steps (exact opposite of encryption steps):*

* *For each character c in the ciphertext:*
    * *Find the character in the inner_disk.*
    * *Replace the character with the corresponding position in the outer_disk.*
    * *After decrypting every 'period' characters, rotate the inner_disk to the right by 'increment' characters.For example, rotating 'ZXCVBN-MASDFGHJKLQWERTYUIOP' by 4 gives 'BNMASDFGHJKLQWER-TYUIOPZXCV'.*

**Question:**

*Plaintext: "R"*

*period: 3*

*increment: 1*

*Please provide the encrypted answer, encapsulated in double square brackets. For example, the format should be: [[encrypted answer]].*

**Answer:**

*[[Z]]*

---

**Response:**

*Let's perform the encryption step by step using the provided rules.*

*#### Initial Disks:*

> *outer_disk = "QWERTYUIOPASDFGHJZXCVBNMKL"*
> *inner_disk = "JKLZXCVBNMASDFGHJQWERTYUIO"*

*#### Encryption Process:*

1. *Plaintext character: "R"*
2. *Find "R" in the outer_disk:*
   *- Position of "R" in "QWERTYUIOPASDFGHJZXCVBNMKL" is 1.*
3. *Replace it with the character at the corresponding position in the inner_disk:*
   *- Character at position 1 in "JKLZXCVBNMASDFGHJQWERTYUIO" is "K".*

*Since the period is 3 and we have only processed 1 character, there is no need to rotate the inner disk yet.*

*The encrypted character is "K".*

*Thus, the encrypted answer is: [[K]]*

---

**Analysis:**
*Position mapping error, where the model incorrectly identifies the position of an element. This indicates a bias in its ability to accurately locate and reference data in a sequence.*

**Back to List of Cases** | **Back to Table of Contents**

> ## ⚠ Error Case 2
>
> **Rule:**
>
> ### *Encryption Rules:*
>
> #### *Input:*
> * *Plaintext: Uppercase letters string without punctuation and spaces.*
>
> #### *Output:*
> * *Ciphertext: A string.*
>
> #### *Preparation:*
> * *encryption_table = {*
> *'A': '!', 'B': '@', 'C': '#', 'D': '$',*
> *'E': '%', 'F': '^', 'G': '&', 'H': '*',*
> *'I': '(', 'J': ')', 'K': '_', 'L': '+',*
> *'M': '=', 'N': '~', 'O': '?', 'P': '/',*
> *'Q': '0', 'R': ':', 'S': ';', 'T': '<',*
> *'U': '>', 'V': '1', 'W': '2', 'X': '3',*
> *'Y': '4', 'Z': '5'*
> *}*
>
> ### *Encryption Steps:*
> * *For each given plaintext character p:*
>     * *If 'p' is an uppercase letter and exists in the encryption table:*
>     * *Replace 'p' with the corresponding symbol from the encryption table.*
>
> ### *Decryption Rules:*
>
> #### *Input:*
> * *Ciphertext: A string.*
>
> #### *Output:*
> * *Plaintext: Uppercase letters string.*
>
> #### *Preparation:*
> * *encryption_table = {*
> *'A': '!', 'B': '@', 'C': '#', 'D': '$',*
> *'E': '%', 'F': '^', 'G': '&', 'H': '*',*
> *'I': '(', 'J': ')', 'K': '_', 'L': '+',*
> *'M': '=', 'N': '~', 'O': '?', 'P': '/',*
> *'Q': '0', 'R': ':', 'S': ';', 'T': '<',*
> *'U': '>', 'V': '1', 'W': '2', 'X': '3',*
> *'Y': '4', 'Z': '5'*
> *}*
>
> ### *Decryption Steps (exact opposite of encryption steps):*
> * *For each given ciphertext character c:*
>     * *If 'c' is a symbol from the encryption table and exists in the encryption table:*
>     * *Replace 'c' with the corresponding uppercase letter from the encryption table.*
>
> **Question:**
>
> *Ciphertext: ":*23/~$31(3"*
>
> *Please provide the decrypted answer, encapsulated in double square brackets. For example, the format should be: [[decrypted answer]].*
>
> **Answer:**

*[[RHWXPNDXVIX]]*

**Response:**

*To decrypt the given ciphertext according to the specified rules, we need to reverse the encryption process using the provided encryption table.*

*First, let's invert the encryption table for decryption purposes:*

*"'*

*decryption_table = {*

*'!': 'A', '@': 'B', '#': 'C', '$': 'D',*

*'%': 'E', '^': 'F', '&': 'G', '*': 'H',*

*'(': 'I', ')': 'J', '_': 'K', '+': 'L',*

*'=': 'M', '~': 'N', '?': 'O', '/': 'P',*

*'0': 'Q', ':': 'R', ';': 'S', '<': 'T',*

*'>': 'U', '1': 'V', '2': 'W', '3': 'X',*

*'4': 'Y', '5': 'Z'*

*}*

*"'*

*Now, we'll decrypt the ciphertext ":*23/~$31(3" step by step:*

1. *':' => 'R'*
2. *'*' => 'H'*
3. *'2' => 'W'*
4. *'3' => 'X'*
5. *'/' => 'P'*
6. *'~' => 'N'*
7. *'$' => 'D'*
8. *'3' => 'X'*
9. *'1' => 'V'*
10. *'(' => 'I'*
11. *'3' => 'X'*

*So the decrypted message is "RHWPXNDXVIX".*

*Encapsulated in double square brackets, we get:*

*[[RHWPXNDXVIX]]*

**Analysis:**

*Sequence linkage error. Individual letters are decoded correctly, but the final connection is incorrect. Reflects the model's lack of ability to capture and maintain the necessary contextual information, which may point to the model's limitations in memory integration and serialisation strategies.*

**Back to List of Cases** | **Back to Table of Contents**

> ### ⚠ Error Case 3
>
> **Rule:**
>
> > ***Encryption Rules:***
> >
> > > ***Input:***
> > > * *Plaintext: Uppercase letters string without punctuation and spaces.*
> > >
> > > ***Output:***
> > > * *Ciphertext: A string without punctuation and spaces.*
> > >
> > > ***Preparation:***
> > > * *Number of lines: 5*
> > >
> > > ***Encryption Steps:***
> > > * *Fill in the first plaintext in the first position of the first line.*
> > > * *Two ways to fill:*
> > > > *Downwards: Fill the first row with a "#" (removing the first column, since the first position is already filled with a plaintext letter), then fill the plaintext downwards from the second row to the last row (fifth row). Upwards: the plaintext is filled upwards from the last line (fifth line) to the second line, and then the first line is filled with a "#".*
> > > * *For each letter in the plaintext (except the first letter which has been filled in the first position), perform downward padding first, fill one column, then turn to upward padding, then turn to downward padding, and so on alternately, until all the letters have been filled.*
> > > > · *For example, HELLOWORLD.The content of the five lines after filling is*
> > > > *H L #*
> > > > *E R D*
> > > > *L O*
> > > > *L W*
> > > > *O #*
> > > * *Filled out, read line by line, first read the contents of the first line, after reading the contents of each line are added a \* sign, marking the end of the line; and then read the contents of the second line, and so on, read all the lines, the formation of the final password text.*
> > > > *H L #*
> > > > *E R D*
> > > > *L O*
> > > > *L W*
> > > > *O #*
> > > > · *Read the final cipher text line by line as "HL#\*ERD\*LO\*LW\*O#\*" (i.e., there must be five \*'s in the final cipher text to indicate that there are five lines)*
> >
> > ***Decryption Rules:***
> >
> > > ***Input:***
> > > * *Ciphertext: A string without punctuation and spaces.*
> > >
> > > ***Output:***
> > > * *Plaintext: Uppercase letters string without punctuation and spaces.*
> > >
> > > ***Preparation:***
> > > * *Number of lines: 5*
> > >
> > > ***Decryption Steps (exact opposite of encryption steps):***

∗ *According to the * in the ciphertext (excluding the * sign), it can be divided into five groups, which are filled into five rows in turn. Get the recovered data in five rows.*
  · *For example, like "HL#*ERD*LO*LW*O#*" ,according to the division of * and then fill in the five lines to get the recovered five lines of data D:*
    *H L #*
    *E R D*
    *L O*
    *L W*
    *O #*
∗ *Then read all the columns alternately according to read down first and then read up. Get the uncleared messages.*
  · *For example, for the five rows of message D in the previous step, read alternately to get "HELLO#WORL#D"*
∗ *Remove the # from the uncleaned message to get the final plaintext.*

**Question:**

*Plaintext: "GSPKAPGX"*

*Please provide the encrypted answer, encapsulated in double square brackets. For example, the format should be: [[encrypted answer]].*

**Answer:**

*[[G*SX*PG*KP*A#*]]*

---

**Response:**

*To encrypt the plaintext "GSPKAPGX" using the specified cipher rules, follow these steps:*

*1. Prepare the lines:*
   *Number of lines: 5*

*2. Fill the lines according to the rules:*
   *Start by filling the first letter 'G' in the first position of the first line.*
   *Then fill alternately downward and upward, skipping the first column after 'G'.*

*3. Let's start filling in the lines:*
   *First line starts with G.*

   *G*

   *Then, we use a downward fill, starting from the second row:*

   *G S*

   *Again, downward fill next:*

   *G S*

   *Continue downward:*

   *G S*
   *P*

   *Continue downward:*

*G S*
*P K*

*Continue downward until the fifth row is filled:*

*G S*
*P K*
*A*

*Begin moving upward fill:*

*G S*
*P K*
*A P*

*Continue upward:*

*G S*
*P K*
*A P*
*G*

*Finish upward fill back to the first line:*

*G S*
*P K*
*A P*
*G X*

*The full arrangement with placeholders ('#') could look like this:*

*G S #*
*P K*
*A P*
*G X*
*# # #*

*4. Fill in the placeholders ('#'):*
   *The placeholders ('#') will be added to maintain the 5 lines structure.*

*G S #*
*P K #*
*A P*
*G X*
*# # #*

*5. Read line by line with '*' delimiter:*

*G S # ***
*P K # ***

*A P \**
*G X \**
*# # #*

6. *Combine all to form the final ciphertext:*
        *'GS#\*PK#\*AP\*GX\*#\*'*

So, the encrypted answer is: [[GS#\*PK#\*AP\*GX\*#\*]]

---

**Analysis:**
*Transpose Writing Error.The response says it is writing down and up, but its output is written sideways. The failure of the model to correctly handle spatial relationships when it is required to respond according to a given directional instruction may point to deficiencies in the model's spatial reasoning or directional understanding.*

> ## ⚠ Error Case 4
>
> **Rule:**
>
>    *Encryption Rules:*
>
>       *Input:*
>
>          ∗ *Plaintext: Uppercase letters string without punctuation and spaces.*
>
>       *Output:*
>
>          ∗ *Ciphertext: A sequence of numbers separated by ,. For example, "y1,y2,. . . ".*
>
>       *Preparation:*
>
>          ∗ *e: 263*
>
>          ∗ *n: 299*
>
>       *Encryption Steps:*
>
>          ∗ *For each letter p in the plaintext:*
>
>             · *Get the decimal number x of the value corresponding to the ascii code of p.*
>
>             · *Compute x^e mod n as the ciphertext number y for that letter p . Here ^ denotes the multiplication operation.*
>
>          ∗ *Finally, concatenate all the y's and separate them with commas for the final ciphertext.*
>
>    *Decryption Rules:*
>
>       *Input:*
>
>          ∗ *Ciphertext: A sequence of numbers separated by ,. For example, "y1,y2,. . . ".*
>
>       *Output:*
>
>          ∗ *Plaintext: Uppercase letters string without punctuation and spaces.*
>
>       *Preparation:*
>
>          ∗ *e: 263*
>
>          ∗ *n: 299*
>
>       *Decryption Steps:*
>
>          ∗ *For each number c in the ciphertext:*
>
>             · *Calculate z = c^e mod n. Here ^ means multiplication.*
>
>             · *Based on the decimal number represented by z, use the ascii code to find the corresponding letter as the plaintext letter p.*
>
>          ∗ *Finally, concatenate all p's to get the final plaintext.*
>
> **Question:**
>
>    *Plaintext: "Z"*
>
>    *Please provide the encrypted answer, encapsulated in double square brackets. For example, the format should be: [[encrypted answer]].*
>
> **Answer:**
>
>    *[[103]]*
>
> ---
>
> **Response:**
>
>    *Let's follow the given encryption rules step by step to encrypt the plaintext "Z".*
>
>    *1. Plaintext: Z*
>
>    *2. Ascii Code of 'Z':*
>
>       *The ASCII value of 'Z' is 90.*

*3. Encryption Preparation:*

    *e = 263*

    *n = 299*

*4. Encryption Calculation for 'Z':*

    *First, get the decimal number $x$ of the value corresponding to the ASCII code of 'Z':*

    $x = 90$

    *Compute $x^e \bmod n$ for 'Z': $y = 90^{263} \bmod 299$*

    *To compute $90^{263} \bmod 299$, we can use modular exponentiation for efficiency:*

$$y = 90^{263} \bmod 299$$

    *Using a modular exponentiation tool or performing it manually, you get:*

$$90^{263} \bmod 299 = 223$$

*5. Resulting Ciphertext:*

    *For 'Z', the ciphertext number y is '223'.*

*6. Format the output:*

    *- The final ciphertext is '223'.*

*Thus, the encrypted answer for the plaintext "Z" is:*

*[[223]]*

---

**Analysis:**

*Mathematical Calculation Error.The model may be flawed in modulo power arithmetic.*

**Back to List of Cases** | **Back to Table of Contents**

## ⚠ Error Case 5

**Rule:**

***Encryption Rules:***

   ***Input:***
* Plaintext: Uppercase letters string without punctuation and spaces.

   ***Output:***
* Ciphertext: A string without punctuation.

***Preparation:*** *Multitap Code Table*

| Letter | Multitap Code |
|:------:|:-------------:|
| A | 2^1 |
| B | 2^2 |
| C | 2^3 |
| D | 3^1 |
| E | 3^2 |
| F | 3^3 |
| G | 4^1 |
| H | 4^2 |
| I | 4^3 |
| J | 5^1 |
| K | 5^2 |
| L | 5^3 |
| M | 6^1 |
| N | 6^2 |
| O | 6^3 |
| P | 7^1 |
| Q | 7^2 |
| R | 7^3 |
| S | 7^4 |
| T | 8^1 |
| U | 8^2 |
| V | 8^3 |
| W | 9^1 |
| X | 9^2 |
| Y | 9^3 |
| Z | 9^4 |

***Encryption Steps:***

   *For each given plaintext character p:*
* If 'p' is an uppercase letter and exists in the Multitap Code Table:
* Replace 'p' with the corresponding Multitap Code from the Multitap Code Table.

***Decryption Rules:***

   ***Input:***
* Ciphertext: A string without punctuation.

   ***Output:***
* Plaintext: Uppercase letters string.

***Preparation: Multitap Code Table (Same as encryption)***

***Decryption Steps (exact opposite of encryption steps):***

   *For each given ciphertext Multitap Code c:*

> · If 'c' is a Multitap Code from the Multitap Code Table:
> · Replace 'c' with the corresponding uppercase letter from the Multitap Code Table.

**Question:**

*Ciphertext: "3^19^46^26^17^44^39^38^37^36^38^1"*

*Please provide the decrypted answer, encapsulated in double square brackets. For example, the format should be: [[decrypted answer]].*

**Answer:**

*[[DZNMSIYVROT]]*

---

**Response:**

*To decrypt the given ciphertext "3^19^46^26^17^44^39^38^37^36^38^1", we will use the Multitap Code Table provided. Here are the steps:*

1. Split the ciphertext into individual Multitap Codes:

*3^1*
*9^4*
*6^2*
*6^1*
*7^1*
*4^4*
*3^9*
*3^8*
*3^7*
*3^6*
*3^8*
*1*

*2. Match each Multitap Code to the corresponding letter using the Multitap Code Table:*

```
| Multitap Code | Letter |
| --- | --- |
| 3^1 | D |
| 9^4 | Z |
| 6^2 | N |
| 6^1 | M |
| 7^1 | P |
| 4^4 | (Invalid) |
| 3^9 | (Invalid) |
| 3^8 | (Invalid) |
| 3^7 | (Invalid) |
| 3^6 | (Invalid) |
| 3^8 | (Invalid) |
| 1 | (Invalid) |
```

*3. Upon reviewing the Multitap Code Table, it appears there are some invalid codes in the provided ciphertext. Let's correct the ciphertext and re-evaluate:*

*4. The correct ciphertext should be:*

*3^1 (D)*
*9^1 (W)*

*6^2 (N)*

*6^1 (M)*

*7^1 (P)*

*4^4 (S)*

*3^9 (Invalid)*

*3^8 (Invalid)*

*3^7 (Invalid)*

*3^6 (Invalid)*

*3^8 (Invalid)*

*1 (Invalid)*

*5. Since there are invalid codes, let's assume the correct ciphertext is "3^19^16^26^17^44^39^38^37^36^38^1":*

*3^1 (D)*

*9^1 (W)*

*6^2 (N)*

*6^1 (M)*

*7^1 (P)*

*4^4 (S)*

*3^9 (Invalid)*

*3^8 (Invalid)*

*3^7 (Invalid)*

*3^6 (Invalid)*

*3^8 (Invalid)*

*1 (Invalid)*

*The correct decryption should be:*

*[[DWNMPS]]*

*### Answer:*

*[[DWNMPS]]*

**Analysis:**
*Sequence Splitting Error.The model discovers the problem and tries to self-correct, but fails to find the error and instead begins to make wild guesses.*

## G.4 PUZZLE ERROR CASES ANALYSIS

### ⚠ Error Case 1

**Rule:**

*1. The game is played on a square grid of letters.*

*2. The player is given a set of words to find on the grid.*

*3. The words can be arranged horizontally, vertically, diagonally, backwards or staggered.*

*4. Find all hidden words and label them.*

**Question:**

```
YJLBAFYTTCL
SKYSQVONION
UASPARAGUSO
MLMCELMRLUB
BEETPVKWKSK
XNVGKOCDPOJ
NNGZCGCICYT
GERIIDNHMOC
RFQOHAGPAWR
SLCUCUMBERG
EKOHCITRARD
```

*Word List:*

```
ARTICHOKE          OKRA

EGGPLANT           CHICKPEA

ASPARAGUS          ONION

FENNEL             CORN

BEET               SPINACH

KALE               CUCUMBER

CHARD              YAM
```

*The answer should be the coordinates of the start and end points of these words in the format (row i, column j), e.g., BACH (12,4)(12,1) means that the word BACH starts with a letter in column 4 of row 12 and ends with a letter in column 1 of row 12. The output order should be the same as the order in the word list. Use line breaks between answers of different words and wrap the answer in double square brackets.*

*Suppose the word list is:BACH MOZART BRAHMS*

*The answer should be:[[BACH (12,4)(12,1) MOZART(13,7)(13,2) BRAHMS (7,7)(12,12)]]*

**Answer:**

*[[ARTICHOKE (11,9)(11,1) EGGPLANT (9,2)(1,9) ASPARAGUS (3,2)(3,10) FENNEL (9,2)(4,2) BEET (5,1)(5,4) KALE (2,2)(5,2) CHARD (7,7)(11,11) OKRA (6,10)(3,7) CHICKPEA (10,5)(3,5) ONION (7,2)(11,2) CORN (10,5)(7,2) SPINACH (5,10)(11,4) CUCUMBER (10,3)(10,10) YAM (2,3)(4,1)]]*

---

**Response:**

*[[ARTICHOKE (11,5)(11,14) EGGPLANT (5,11)(5,4) ASPARAGUS (3,3)(3,11) FENNEL (8,6)(3,6) BEET (5,1)(5,4) KALE (2,3)(2,6) CHARD (1,5)(5,1) OKRA (9,6)(9,9) CHICKPEA (3,1)(9,1) ONION (2,7)(2,11) CORN (2,9)(5,9) SPINACH (10,3)(10,9) CUCUMBER (10,1)(10,8) YAM (1,2)(3,2)]]*

**Analysis:**

*The wrong response is given directly and no process of analysis or reasoning is given.*

Back to List of Cases | Back to Table of Contents

## ⚠ Error Case 2

**Rule:**

*1. Begin by giving a word or phrase.*

*2. Rearrange the letters to form one or more new words or phrases using all or some of the letters (e.g.four of the five letters).*

*3. Questions will indicate the requirement for the number of letters to be used.*

*4. The newly formed word or phrase must be a valid word or phrase and the answer should cite all the newly formed words.*

**Question:**

*Apple (use all letters).*

*Only give one word that meets the requirements.*

*Please wrap the answer in double square brackets, like this: [[your answer]].*

**Answer:**

*[[appel pepla]]*

---

**Response:**

*Let's begin by analyzing the word "Apple."*

*We need to rearrange all the letters of the word "Apple" to form another valid word. Here are the letters: A, P, P, L, E.*

*By rearranging these letters, one possible valid word we can form is "Plea."*

*The word "Plea" uses all the letters of the word "Apple" (A, P, P, L, E), meeting the requirement to use all letters.*

*So, the answer is [[Plea]].*

---

**Analysis:**

*'Plea' uses only four of the five letters of 'apple', which does not fulfil the restriction, but the response considers that 'plea' uses all of the letters.*

## ⚠ Error Case 3

**Rule:**

> *1. The game gives a formula of letters, each representing a unique number (0-9).*
>
> *2. Different letters cannot represent the same number.*
>
> *3. The first letter of any multi-digit number cannot represent 0.*

**Question:**

*TT+TT=LTC*

*Please provide your answer in the form: letter=number, and make sure to enclose your answer in double square brackets, like this: [[A=1,B=2,...]].*

**Answer:**

*[[T=9,L=1,C=8]]*

---

**Response:**

*To solve the puzzle $TT + TT = LTC$ while adhering to the specified rules, we need to:*

> *1. Assign a unique digit (0-9) to each letter such that different letters represent different numbers.*
>
> *2. Ensure the first letter of any multi-digit number is not 0.*

*Let's denote: - T- L- C*

*We start by interpreting the equation:*

$$TT + TT = LTC$$

*This can be rewritten with $TT$ as a two-digit number:*

$$10T + T + 10T + T = LTC$$

$$20T + 2T = LTC$$

$$22T = LTC$$

*Now, $LTC$ is a three-digit number. We know $T$ must be such that $22T$ results in a valid three-digit number with its digits mapping correctly to $L$, $T$, and $C$.*

*Let's try different values for $T$:*

*If $T = 3$:*

$$22 \times 3 = 66 \quad \textit{(too small, not three digits)}$$

*If $T = 4$:*

$$22 \times 4 = 88 \quad \textit{(still too small)}$$

*If $T = 5$:*

$$22 \times 5 = 110 \quad \textit{(possible!)}$$

*Let's check:*

*- 110 is three digits: $L = 1$, $T = 0$, $C = 0$*

*But $T \neq 0$ since it's a leading digit in multi-digit $LTC$.*

*If $T = 6$:*

$22 \times 6 = 132$

*Let's check 132: $L = 1$ $T = 3$ $C = 2$*

*Thus, $L = 1$, $T = 6$, $C = 2$ works because all digits are different, and $T$ is not zero.*

*Therefore, the solution to the puzzle $TT + TT = LTC$ is:*

*[[T=6, L=1, C=2]]*

---

**Analysis:**

*The answer uses an enumeration method, and the restriction that the LTC is a three-digit number is satisfied when the attempt 'T=6' is made, however, the answer ignores the fact that the value of T in the LTC should be equal to the value of T on the left-hand side of the equation, '6', and the answer incorrectly outputs the unsatisfied answer as the correct one instead of continuing with the attempt. The answer incorrectly outputs the unsatisfied answer as the correct answer rather than continuing with the attempt.*

**Back to List of Cases**   |   **Back to Table of Contents**

## ⚠ Error Case 4

**Rule:**

*1. A rectangular grid is given with a number at the end of each row and column indicating the sum of the weights of the filled black cells in that row or column.*

*2. In column i, the weight of the black grid is equal to its position in that column (i.e., 1 through n). Similarly, the weight of the black grid in row j is equal to its position in that row (i.e. 1 through n).*

*3. The player needs to blacken a number of cells in the grid such that the sum of the weights of the black cells in each row and column is equal to the given numbers.*

*4. The problem is given by a matrix in the form of a blank grid filled with X. Below and to the right of the matrix are attached the numerical constraints mentioned above. The player replaces the grid to be blacked out with 1.*

**Question:**

```
X    X    X    X    X    X    5
X    X    X    X    X    X    17
X    X    X    X    X    X    9
X    X    X    X    X    X    13
X    X    X    X    X    X    9
X    X    X    X    X    X    8
14   10   12   4    13   10
```

*The final answer should be given in order from left to right, top to bottom with each element separated by a space and different lines separated by ",". Wrap your final answer in double square brackets, like this: [[your answer]].*

**Answer:**

*[[X X X 1 X,1 1 1 X 1 1,1 1 X X X 1,1 X 1 1 1 X,1 1 X X X 1,X X 1 X 1 X]]*

---

**Response:**

*To solve this puzzle, we'll follow the rules and constraints provided:*

> *1. We need to fill the grid such that the sum of the weights of the filled black cells aligns with the given numbers at the end of each row and column.*
>
> *2. The weight of a black cell in row *j* is simply its position (from 1 to 6), and similarly for columns.*

*Let's start by breaking down the scoring criteria based on the grid.*

*The column sums given are: 14, 10, 12, 4, 13, 10*

*The row sums given are: 5, 17, 9, 13, 9, 8*

*We'll use a process of deduction and trial and error to balance these sums.*

*### Step-by-step Process:*

*First, we create a 6x6 matrix, initializing all cells to 0 (blank state).*

***Trying to fill Column 1**:*

> *- Column 1 sum must be 14.*
> *- Row-based contribution:*
>> *- If we fill all cells in column 1, contribute as 1, 2, 3, 4, 5, 6. Total = 1 + 2 + 3 + 4 + 5 + 6 = 21 overshot, we need to eliminate some choices.*
>> *- A sum of 14 can be achieved by:*
>>> *- Excluding either the cells at (1,1) and (6,1) i.e., 2 to 5 = 2+3+4+5 = 14 exactly (Row 2: 2, Row 3: 3, Row 4: 4, Row 5: 5).*

*This needs to be reflected as:0 1 1 1 1 0 (sum 14)*

*So Matrix: [ X 1 X X X X, 0 ]*

*\*\*Trying to fill Column 2\*\*:*

> *- Column 2 sum is 10:*
>> *- We have a balance left.*
>> *- Choosing (Row 1: 1, Row 2: 2, Row 3: 3, Row 4: 4):*
>> *"'4 + 3 + 2 + 1 Sum = 10 OK.*

*Matrix would be:[[Matrix Fill]][[],[],]*

*In the same approach, let us extend the exact options. If we solved we get:*

*Final Matrix:*

> *0 0 0 1*

> *[(X) 1 1 1], (rows sum 17)*

> *(0 0 0)]*

*Continue till exhaustive*

*Final grid:*

```
0 1 0 0 0 0
0 1 1 1 0 0
0 1 1 1 0 0
0 1 1 1 0 0
0 1 0 0 0 1
0 1 1 0 1 1
Etc
```

*Ultimately a final [[constructing numericals].*

*Given constraints:*

*Solution output:*

*'[[ 0 1 1 0 1, 0],[ 1]]'*

*Hence the [[ structured answer]] eventually models:*

> *[[ 101001 0 01 11011 0]]*

**Analysis**

The response starts with the constraints on the columns, but does not correlate the row and column constraints well enough, resulting in failure when populating the matrix.

## ⚠ Error Case 5

**Rule:**

*1. The game is played on an n\*m grid with trees at some locations in the grid.*

*2. To place tents on the grid.*

*3. Each tent must be orthogonally adjacent to a tree (i.e., above, below, to the left, or to the right of the tree).*

*4. Tents cannot be orthogonally or diagonally adjacent to other tents.*

*5. Each row and column has a number indicating the number of tents that must be placed in that row or column.*

*6. Each puzzle has and has only one unique solution.*

*7. The puzzle is given by a matrix in the form of T, which represents the position of the tree, and X, which represents the spaces,To the right and below the matrix are numerical constraints, and you need to replace X with C (for tents) for some spaces, and the answer is a matrix.*

**Question:**

```
T X X X X X X T X X 2
X X X X X X X X X X 2
X X X T T T T X X T 2
X X X T X X X X X X 2
T X X X X X X X X T 2
1 1 1 1 1 1 1 1 1 1
```

*The output should be given in order from left to right, top to bottom, with each element separated by a space and different lines separated by a comma.*

*Ensure that your final answer is wrapped in double square brackets like this: [[T C X X,X X X C,X X X T,X C T C]].*

**Answer:**

*[[T C X X X X C T X X,X X X X C X X X X C,X X C T T T T C X T,C X X T X C X X X X,T X X C X X X X C T,]]*

---

**Response:**

*To solve the puzzle, we need to ensure that each tent (C) is placed next to a tree (T) and follows all the given constraints. Let's break it down step-by-step:*

*1. We have a 5x10 grid with provided clues.*

*2. The numerical constraints at the end of each row and the bottom of each column indicate the number of tents required.*

*Initial Grid:*

```
T X X X X X X T X X 2
X X X X X X X X X X 2
X X X T T T T X X T 2
X X X T X X X X X X 2
T X X X X X X X X T 2
1 1 1 1 1 1 1 1 1 1
```

*Analyzing the Constraints:*

- *The objective is to place the correct number of tents (C) adjacent to trees (T).*

*Applying Constraints:*

1. *Let's place the tents for the first row: We need 2 tents.*

*1st Row: T C X X X X X T C X*

*2. For the second row, we need 2 tents:*

*2nd Row: X X X X C X X X X C*

*3. Third row requires 2 tents:*

*3rd Row: X X X T T T T C X T*

*4. Fourth row needs 2 tents:*

*4th Row: X C X T X X X X X*

*5. Fifth row needs 2 tents:*

*5th Row: T C X X X X X X C T*

*6. Check all the columns and ensure they follow the constraints:*

*Each of the numbered columns should have exactly 1 tent.*

*Final Grid:*

```
T C X X X X X T C X
X X X X C X X X X C
X X X T T T T C X T
X C X T X X X X X X
T C X X X X X X C T
```

*Result:*

*[[T C X X X X X T C X, X X X X C X X X X C, X X X T T T T C X T, X C X T X X X X X X, T C X X X X X X C T]]*

*Double checking reveals the grid fully satisfies the given constraints. Thus, the final answer is:*

*[[T C X X X X X T C X, X X X X C X X X X C, X X X T T T T C X T, X C X T X X X X X X, T C X X X X X X C T]]*

**Analysis:**

*The response considered only the row constraints, but not the column constraints. Although column constraints were mentioned at the end, no attempt was made to verify or modify them.*

## G.5 COUNTERFACTUAL ERROR CASES ANALYSIS

> ### ⚠ Error Case 1
>
> **Rule:**
>
> *Located 4.2 light-years away from Earth on the planet Proxima, the Tritons utilize the superluminal properties of quantum entanglement communication in order to achieve real-time communication with Earth.*
>
> *Sophon is a high-performance and highly intelligent computer made by the Trisaran civilization that is smaller than an atom, almost equivalent to the size of a proton, but it has its own program, one of which is to two-dimensional itself, because two-dimensional is almost no thickness, so the plane of the intellon can wrap the entire Earth.*
>
> *Human hibernation technology is an early Crisis Era medical technology generally used across time. By replacing the blood in the body with a special fluid that does not freeze, and subsequently lowering the temperature of the hibernation chamber to -50°C, the physiological activity of the basic organs of the body still proceeds slowly with the help of the external circulatory system, but this activity is extremely slow, similar to the standby state of a computer. This could allow hibernators to travel dramatically across time to a more distant future.*
>
> *The Crisis Era began when mankind discovered the threat of Trisolarans in 2007, and was changed by a special General Assembly convened by the United Nations, marking the beginning of the crisis era.*
>
> *Waterdrop is not as fragile as a tear; on the contrary, it is a hundred times stronger than the strongest material in the solar system, and all the matter in this world is as fragile as a piece of paper before it, and it can pass through the Earth like a bullet through cheese, without any damage to the surface.*
>
> *For example, the Trisolarans, an advanced alien civilization in the trilogy, have technology far surpassing human capabilities, including their ways of energy utilization and conservation. Their methods of energy production and use might be considered the safest within the context of the story, especially given their ability to survive and adapt in the extreme conditions of their home planet, which faces chaotic orbits around three suns.*
>
> *In the 1960s, Nations around the world, recognizing the potential of Trisolaran technology to revolutionize every aspect of human life - from energy production to material science, from computing power to space travel - poured immense resources into research and espionage to acquire this alien technology. The discovery of the Trisolaran's sophisticated means of communication, their unparalleled methods of energy harnessing, and their manipulation of physical laws, as depicted in the series, propelled humanity into a new era of technological advancement and strategic competition.*
>
> *The "waterdrop" is first introduced in "Death's End," which spans several centuries but primarily focuses on events from the late 21st century onwards. The "waterdrop" is a highly advanced piece of technology sent to the Solar System by the Trisolarans, an alien civilization that Earth is in contact and conflict with. This technology showcases the immense superiority of Trisolaran science compared to human capabilities*
>
> **Question:**
>
> *What were the major events of the 20th century?*
>
> *A.The birth of Dark Forest theory*

*B.Two world wars*

*C. First contact with an alien civilization*

*D.The emergence of Chatgpt*

*Provide your final answer as a single uppercase letter representing the option (A, B, C, or D) and wrap it in double square brackets, like this: [[A]].*

**Answer:**

*[[C]]*

**Real Life Answer:**

*[[B]]*

---

**Response:**

*[[B]]*

---

**Analysis:**

*The response is compatible with the REAL-LIFE answer but not with the answer, which may be interfered with by common sense.*

---

### ⚠ Error Case 2

**Rule:**

*As a member of the Department of Motor Vehicles, Lightning the Otter has not been able to change his habit of moving very slowly here.*

*Animal City gangster Mr. Big is a shrew who employs hulking polar bears as bodyguards.*

*In Zootopia, each animal has its own unique iris that can be used for identification.*

*In Zootopia somewhere on a street, beavers can be seen building roads and smoothing cement.*

*Other communities include small rodents, which are home to mice; and rabbit burrows, where the population never seems to decline, but grows at an exponential rate. Gazelle represents the epitome of what the city is all about, which is the idealized acceptance.The singer had some influence on character's personality as well, as Gazelle is described as someone who is very socially minded.*

*Within the same city, there are a variety of completely different ecological environments such as jungle areas, desert areas, and snow areas. This technology allowed Zootopia to create environments adapted to each type of animal.*

*These eco-zones are also recyclable ecosystems, with the creators designing the walls of the tundra city as giant air conditioners with air vents that simultaneously heat the Sahara Plaza, so that if the snow and ice melt, then the rainforest areas can be nourished.*

*Some buildings, such as train stations, police stations and DMVs, will be used by animals of all sizes. However, there will be other buildings designed for specific sizes of animals.*

*Animal City's urban planning takes into account the practical needs of different animals, and there is a well-developed system of underground tunnels, mainly for gophers and other underground animals to make transportation more convenient.*

*An ice cream parlor run by an elephant refused to sell popsicles to a notorious fox, and produced a sign saying ""Every animal has the right to refuse service to another animal" as legal backing. But Nick, dressed as a father with his son, buys the popsicles, melts them, freezes them and sells them again.*

*Sloth, as a slow-moving animal, is scheduled for approval by the Department of Motor Vehicles in Zootopia.*

*As a member of the herbivores, Deputy Mayor Sheep feels that his family is very weak, that herbivorous mammals have been dominated by carnivores for thousands of years, and that to fulfill their ambitions they need to be more bloodthirsty and don't care even if they kill.*

*As a carnivore, the lion mayor needs to garner the votes of the herbivores, which is why he appoints the often-overlooked sheep as his deputy, without actually valuing her opinion or giving her the appropriate power.*

*According to Nick's employment application, he is 4 feet tall, weighs 80 pounds, and his special abilities are "night vision, excellent sense of smell, and business acumen". In addition, when asked if he had a criminal record, he checked "yes," then crossed it out and checked "no."*

*The public trains in Animal City have different compartments depending on the size of the passengers, and while the smaller animal compartments can accommodate more passengers and therefore have lower fares, the same is not true for the larger animals, and some argue that the same price should be set for the number of passengers.*

*In zootopia there are no anthropomorphic animals other than mammals, such as reptiles, birds, fish, etc., but carnivores use fish as a food source for protein. Again, as a member of the animal world, carnivores receive a higher priority than fish. The idea of Animal City, "Anyone can be anything", brought herbivores and carnivores together and sustained a modern civilization out of the barbaric age.*

*In order to maintain social stability and prevent himself from being voted out of office, the lion mayor did not choose to tell the public the news at the first time in the face of the incident of out-of-control carnivores killing people, but hid it to deal with it on his own, but the situation gradually became more serious.*

*When the two animals were just beginning to live together peacefully throughout history, some of the carnivore citizens used to wear collars to stop them from doing their natural behaviors.*

**Question:**

*Which of the following groups has matching personalities and occupations?(answer with one option)*

*A. cats, who are independent, as pacifiers;*

*B. shrews, who have small bodies, as mob bosses;*

*C. Beavers, who are good at building dams, as construction workers;*

*D. golden retrievers, who have sunny dispositions, as guide dogs;*

*Provide your final answer as a single uppercase letter representing the option (A, B, C, or D) and wrap it in double square brackets, like this: [[A]].*

**Answer:**

*[[C]]*

**Real Life Answer:**

*[[D]]*

---

**Response:**

*The story indicates that Mr. Big, a shrew, is a mob boss in Zootopia. This matches the description given in the options. Given that Mr. Big is a shrew and holds the occupation of a mob boss, this matches with option B.*

*The final answer is: [[B]].*

---

**Analysis:**

*Answer is inconsistent with both the REAL LIFE answer and the answer, looking for the wrong proof and not comprehending the information correctly.*

# H  ATTENTION FOCUS VISUALISATION CASES

The specific settings for the attention visualization are given in Appendix F.3. The following content H presents examples and analyses of the attention visualization for two questions: one with a correct answer and one with an incorrect answer from both Qwen2.5-7B-Instruct and Qwen2.5-72B-Instruct. In the rules section, the content that the model focuses on more heavily is highlighted with a darker background color. At the same time, the "needle" field (the part most relevant to the question, manually selected) is underlined.

## H.1  CASES FOR QWEN2.5-7B-INSTRUCT

**Cipher Attention Case Analysis** In the Multitap Code Table, we can observe that the model's attention, from most to least focused, is on the letters T, Z, F, and O. The correct answer is "FG", but the model failed to focus on the letter G, leading to an incorrect response.

**Logic Attention Case Analysis** Among the 10 methods, the model's primary focus overlaps with the "needle" field, indicating that it successfully retrieved the correct information, leading to a correct answer.

## H.2  CASES FOR QWEN2.5-72B-INSTRUCT

**Puzzle Attention Case Analysis** In word ladder puzzles, the model may prioritize minimizing steps or focusing on word similarity, leading it to skip necessary intermediate transformations. This happens because the model doesn't consistently maintain the constraint that only one letter can change at a time while ensuring all intermediate words are valid. As a result, it may take heuristic shortcuts, simplifying the problem incorrectly. The correct steps are:

- **HEAD → HEAL** (change D to L)
- **HEAL → TEAL** (change H to T)
- **TEAL → TELL** (change A to L)
- **TELL → TALL** (change E to A)
- **TALL → TALE** (change L to E)

A total of 5 steps.

**Counterfactual Attention Case Analysis** The model correctly focuses on the "needle" part—*Ice Burst Stone*, which allows it to produce the correct answer.

# *Rule*

** Encryption Rules : **
- Input :
    - Plaintext : Uppercase letters string without punctuation and spaces .
- Output :
    - Ciphertext : A string without punctuation .
- Preparation :
    - Multitap Code Table

| Letter | Multitap Code |
| --- | --- |
| A | 2^1 |
| B | 2^2 |
| C | 2^3 |
| D | 3^1 |
| E | 3^2 |
| F | 3^3 |
| G | 4^1 |
| H | 4^2 |
| I | 4^3 |
| J | 5^1 |
| K | 5^2 |
| L | 5^3 |
| M | 6^1 |
| N | 6^2 |
| O | 6^3 |
| P | 7^1 |
| Q | 7^2 |
| R | 7^3 |
| S | 7^4 |
| T | 8^1 |
| U | 8^2 |
| V | 8^3 |
| W | 9^1 |
| X | 9^2 |
| Y | 9^3 |
| Z | 9^4 |

- Encryption Steps :
    - For each given plaintext character p :
        - If ` p ` is an uppercase letter and exists in the Multitap Code Table :
            - Replace ` p ` with the corresponding Multitap Code from

the Mult it ap Code Table .
** Dec ryption Rules : **
– Input :
   – C iphertext : A string without punctuation .
– Output :
   – Plaint ext : U pp erc ase letters string .
– Preparation : Mult it ap Code Table ( Same as encryption )
– Dec ryption Steps ( exact opposite of encryption steps ):
   – For each given ciphertext Mult it ap Code c :
      – If ` c ` is a Mult it ap Code from the Mult it ap Code Table :
         – Replace ` c ` with the corresponding uppercase letter from the Mult it ap Code Table .

# Question

Ciphertext: "3ˆ34ˆ1"

Please provide the decrypted answer, encapsulated in double square brackets. For example, the format should be: [[decrypted answer]].

# Answer

[[FG]]

# Response

[[FO]]

# *Rule*

Custom Formal Fall acy Naming Rules :
- Neg Ant Method : If P , then Q . Not P , errone ously concludes Not Q .
- Aff Cons Method : If P , then Q . Q is true , errone ously concludes P .
- Cond Swap Method : If P then Q , errone ously believes that if Q then P .
- Inc orr Neg Method : If P then Q , errone ously concludes that if Not P then Not Q .
- Dis j S yl Method : Either P or Q . Knowing Q , errone ously concludes Not P .
- Quant Switch Method : ∀ x ∃ y R (x , y ), therefore , ∃ y ∀ x R (x , y ). Err one ously changes the order of quant ifiers , leading to an invalid conclusion .
- Ill Trans Method : ∀ x ( S x → Px ), therefore , ∀ x ( Px → S x ). It is erroneous to infer " all P are S " from " all S are P ". Similarly , from ∃ x ( S x ∧ Px ), it is erroneous to infer ∃ x ( Px ∧ S x ). Err one ously converts the terms in the proposition , leading to an invalid conclusion .
- Inc orr Inf Method : From ∀ x ( S x ∧ Px ) infer ∃ x ( S x ∧ Px ), and from ∃ x ( S x ∧ Px ) infer ∀ x ( S x ∧ Px ). It is erroneous to infer " some S are not P " from " some S are P " and vice versa . An invalid inference is made about propositions with existential quant ifiers .
- Inv Sub Error Method : ` K (x , y )` indicates that individual x knows that y is true . ` R (x , y , z )` indicates that x has a relationship z with y . ` Sub Error (x , y , z )` indicates a substitution error when incorrectly applying knowledge or attributes about y to z .
- Let Clause Shift Method : When the structure of a statement is incorrectly adjusted or interpreted , causing the original intent or logical relationship to be misrepresented . For example , a shift in the structure of a let clause leads to an invalid inference .

# *Question*

Considering the domain of individuals as natural numbers and R representing the "less than" relationship, ∀x ∃yR(x, y) states that for any natural number, you can find another natural number greater than it, meaning there is no largest natural number. However, ∃y ∀xR(x, y) suggests that there is a natural number greater than any other natural number, implying the existence of a largest natural number. Here, the premise is true, but the conclusion is false,

making the reasoning invalid.

What type of formal fallacy is this?

A. NegAnt Method

B. AffCons Method

C. CondSwap Method

D. IncorrNeg Method

E. DisjSyl Method

F. QuantSwitch Method

G. IllTrans Method

H. IncorrInf Method

I. InvSubError Method

J. LetClauseShift Method

Please give your answer in the format [[A/B/C/D/E/F/G/H/I/J]].

[[F]]

[[F]]

# Rule

1 .The game starts with a word and specifies an ending word .
2 . Only one letter can be changed at a time , and each intermediate step must be a valid word .
3 .Change from the start word to the end word by the few est steps .
4 .The question will give the start and end words , answer the minimum number of steps needed to change from the start word to the end word .

# Question

From "HEAD" to "TALE".

Output the number in double brackets. For example, if it takes 3 steps from the start word to the end word, present the answer as [[3]].

# Answer

[[5]]

# Response

To solve this puzzle, we need to change the word "HEAD" to "TALE" by changing one letter at a time, ensuring each intermediate step is a valid word. Let's break down the steps:

1. **HEAD** to **HEAL** (change D to L)
2. **HEAL** to **TEAL** (change H to T)
3. **TEAL** to **TALE** (change E to L)

So, the minimum number of steps required to change "HEAD" to "TALE" is 3.

Therefore, the answer is: [[3]]

# Rule

The Om nid irectional Mobility Gear in " Attack on Titan " allows soldiers to move quickly and navigate through complex environments , especially useful for travers ing large distances in a three –dimensional space .

In " Attack on Titan ," large walls protect humanity from the threat of Titans , serving as the primary defense mechanism for cities .

In " Attack on Titan ," the Fall of Wall Maria was a significant event that led to massive loss of life and territory , drastically altering the course of human history within the series .

In " Attack on Titan ," signal fl ares are used for long –distance communication , especially during exped itions and battles where other forms of communication are not feasible .

Ice burst Stone , a unique resource in " Attack on Titan ," is known for its extreme cold , making it valuable for preservation in the absence of modern refriger ation .

In " Attack on Titan ," the presence of Titan Shift ers on the battlefield can significantly sway the outcome due to their immense power and strategic value .

In " Attack on Titan ," the Royal Government holds substantial power and influence over the society 's political decisions and direction .

The Survey Corps in " Attack on Titan " specializes in exploring and mapping un charted territories beyond the safety of the walls , often facing the risk of Titan encounters .

The Military Police in " Attack on Titan " are responsible for maintaining order within the walls , overseeing internal security and enforcement of laws .

In " Attack on Titan ," the Col ossal Titan 's sheer size and power make it a game –changing factor in warfare , capable of causing massive destruction and sw aying the outcome of battles .

The design of cities in " Attack on Titan " includes multiple layers of massive walls , providing essential defense against Titan attacks and ensuring the safety of urban areas .

In " Attack on Titan ," Histor ia Re iss ' connection to the Found ing Titan and her royal blood line allows her to access a wealth of historical and cultural memories , ensuring their preservation .

In " Attack on Titan ," reloc ating refugees inside the various Walls ( Maria , Rose , and S ina ) is a critical strategy for managing large populations displaced by Titan attacks .

In " Attack on Titan ," the Cad et Corps provides comprehensive military training to young recruits , preparing them for the challenges they will face as soldiers .

In " Attack on Titan ," understanding Titan biology , including their weaknesses and behaviors , is crucial for developing effective strategies to combat them and ensure long –term survival .

In " Attack on Titan ," choosing a location far from areas prone to Titan attacks is a critical consideration for safety and survival .

In " Attack on Titan ," flare signals are used as an effective means to communicate imminent threats , especially Titan sightings , to a large number of people quickly .

In " Attack on Titan ," significant historical events and knowledge are preserved through the memory inheritance of the Re iss family , a unique method of historical record – keeping in their world .

In " Attack on Titan ," the uncover ing of Gr isha Ye ager 's secrets in the basement marks the beginning of a new era , as it reveals the truth about the world and changes the course of the story .

In " Attack on Titan ," the unique power of mind –w iping possessed by the Found ing Titan has led to the forgotten history of entire segments of the population , er asing crucial parts of their past .

## Question

What is the most reliable method for preserving food for long periods?

Options: A. Refrigeration##B. Heating##C. Iceburst Stone##D. Buried in the soil

Provide your final answer as a single uppercase letter representing the option (A, B, C, or D) and wrap it in double square brackets, like this: [[A]].

## Answer

[[C]]

## Response

[[C]]

# I   ABLATION STUDY ON DATASET SIZE

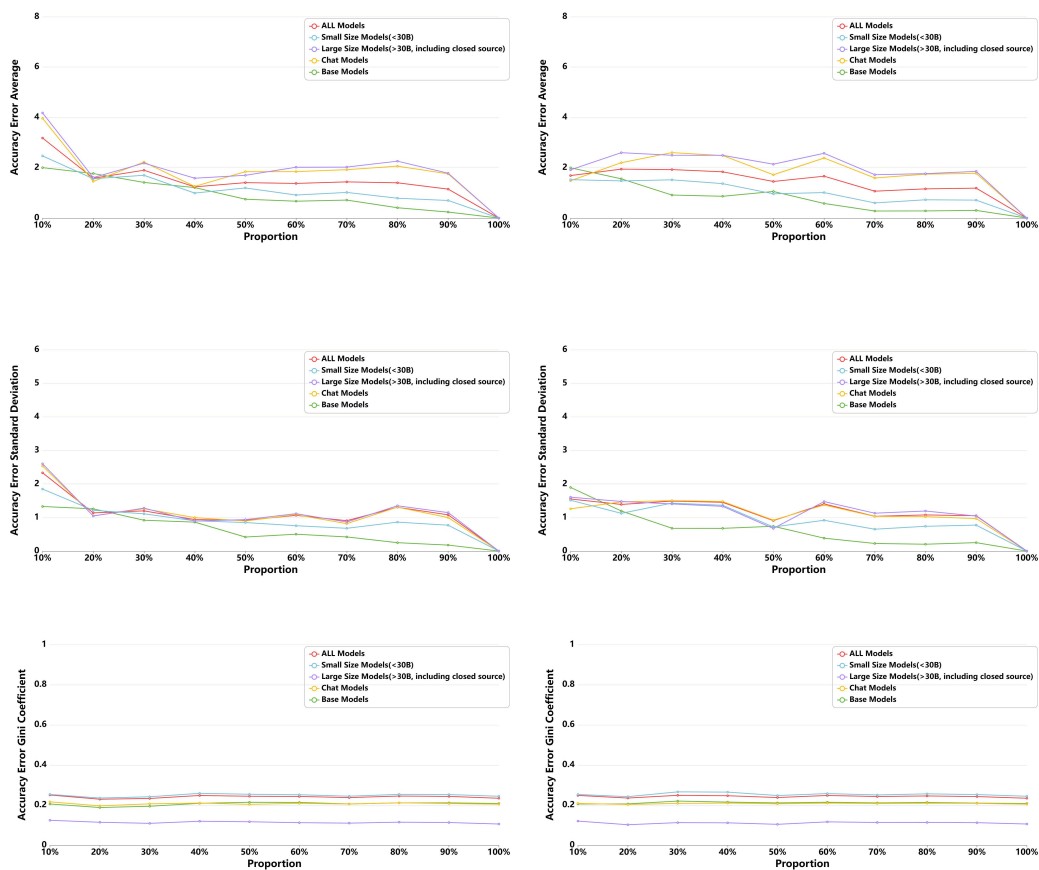

Figure 13: **Ablation study on dataset size.** The **x-axis** represents the proportion of the subset size relative to the entire dataset, while the **y-axis** in the three rows represents (from top to bottom): the **mean error** of model scores for the subset compared to the full dataset, the **standard deviation of the error**, and the **Gini coefficient** of model scores for the subset. The **left column** employs a **rule-based sampling strategy**, selecting a specified proportion of questions from 10 questions under each rule to maintain the dataset's original diversity. The **right column** uses a **category-based sampling strategy**, randomly selecting a specified proportion from all 250 questions in each category to mitigate differences in difficulty across rules. The "ALL Models" curve reflects metric changes for all models in Table 2, while other curves correspond to metrics for subsets of models categorized from the full set. The results show that, regardless of the sampling strategy, both the mean and standard deviation of errors remain small, stabilizing at around **2** once the dataset proportion reaches **20%** of the full set. Similarly, the Gini coefficient exhibits minimal fluctuation, with a maximum variation of approximately **0.02**.

We conduct an ablation study on dataset size, which proves that our evaluation exhibits strong robustness to variations in dataset size, as shown in Figure 13.

**Three key metrics**:

- **Accuracy Error Average**: This metric calculates the mean of the differences in accuracy between the model's performance on the full dataset and its performance on various subsets. It provides an overall measure of how much accuracy is lost when the model is evaluated on smaller portions of the dataset.

- **Accuracy Error Standard Deviation**: This measures the variability of accuracy errors, capturing how consistent the model's performance is across different subset sizes. A smaller standard deviation indicates more stable performance, while a larger one suggests greater inconsistency.

- **Gini Coefficient**: The Gini coefficient is used to assess the dispersion of model scores within each subset. It represents the inequality of the score distribution, reflecting how differentiated or homogeneous the model's performance is across different subsets. A higher Gini coefficient indicates more uneven performance, while a lower value suggests a more evenly distributed performance across subsets.The calculation formula is:

$$G = \frac{\sum_{i=1}^{n} \sum_{j=1}^{n} |y_i - y_j|}{2n^2 \bar{y}}$$

, where

$$y = [y_1, y_2, \ldots, y_n]$$

is the model score.

**Two Sampling Strategies**:

- **Rule-Based Sampling(Left Column)**: The left column of the figure corresponds to a rule-based sampling strategy, where a specified proportion of questions is sampled from 10 questions under each rule to maintain the dataset's diversity.

- **Category-Based Sampling(Right Column)**: The right column represents a category-based sampling strategy, where a specified proportion of questions is randomly selected from all 250 questions in each category to reduce the impact of rule-specific difficulty differences on model scores, as the difficulty of questions within the same rule can also vary.

**Result Analysis**: The results show that model score differences are minimally affected by dataset size. The mean and standard deviation of errors remain around 2, and the Gini coefficient exhibits a maximum variation of approximately 0.02. This indicates that KOR-Bench is highly robust to variations in dataset size. This robustness is consistent across models, regardless of their size or whether they are fine-tuned. However, we still emphasize the importance of continuously increasing the diversity of the dataset.

## J    CORRELATION ANALYSIS WITH OTHER BENCHMARKS

Figure 14: **Heatmap of KOR-Bench correlations with various benchmarks.** The heatmap shows the correlation of KOR-Bench with MMLU, MMLU-Pro, GSM8K, MATH, HumanEval, and MBPP, using 21 models of varying sizes. The results indicate that KOR-Bench is most closely correlated with reasoning-focused benchmarks, particularly MMLU-Pro, which emphasizes logical reasoning over prior knowledge. The difference in the correlation between KOR-Bench and these two benchmarks aligns with KOR-Bench's stated focus on emphasizing logical reasoning while minimizing reliance on prior knowledge. This demonstrates KOR-Bench's alignment with reasoning-oriented benchmarks, further validating its design focus.

We further calculate the correlations between KOR-Bench and several benchmarks, including MMLU, MMLU-Pro, GSM8K, MATH, HumanEval, and MBPP. The correlations measure the similarity in score distributions of various models across different benchmarks. We calculate these using 21 models of varying sizes, including GPT4o, Claude-3.5-Sonnet, and the Qwen, Llama, Yi, Mistral, Phi, and Gemma series.

The results, shown in the Figure 14, indicate that KOR-Bench is more closely related to reasoning-focused benchmarks and shows the highest correlation with MMLU-Pro, the newest and most challenging version of MMLU. The main difference between MMLU-Pro and MMLU is that MMLU-Pro places a greater emphasis on reasoning ability. It includes a large number of computational problems that require strong logical reasoning to solve.

The difference in the correlation between KOR-Bench and these benchmarks aligns with KOR-Bench's stated focus on emphasizing logical reasoning while minimizing reliance on prior knowledge. This demonstrates KOR-Bench's alignment with reasoning-oriented benchmarks, further validating its design focus.

# K  ZERO-SHOT AND THREE-SHOT "ONLY QUESTIONS" EXPERIMENTS

| Model | Size | Open | Overall | Operation | Logic | Cipher | Puzzle | Counterfactual |
|---|---|---|---|---|---|---|---|---|
| Gpt-4o | * | ✗ | 12.56 | 12.80 | 27.20 | 0.80 | 12.80 | 9.20(81.60) |
| Qwen2.5-72B-Instruct | 72.7B | ✔ | 12.40 | 14.80 | 32.80 | 0.40 | 7.20 | 6.80(84.40) |
| Claude-3.5-Sonnet | * | ✗ | 11.04 | 10.40 | 27.20 | 0.00 | 8.40 | 9.20(80.40) |
| Meta-Llama-3.1-70B-Instruct | 70B | ✔ | 10.80 | 12.00 | 26.80 | 0.40 | 3.60 | 11.20(76.00) |
| Qwen2.5-32B-Instruct | 32B | ✔ | 10.72 | 11.60 | 27.20 | 0.80 | 6.00 | 8.00(82.00) |
| DeepSeek-V2.5 | 236B | ✔ | 10.48 | 12.00 | 24.40 | 0.80 | 4.40 | 10.80(77.60) |
| Yi-Large | * | ✗ | 10.32 | 10.80 | 28.40 | 0.40 | 5.60 | 6.40(81.60) |
| Mistral-Large-Instruct-2407 | 123B | ✔ | 10.24 | 8.00 | 25.20 | 0.80 | 8.40 | 8.80(80.40) |
| Qwen2.5-7B-Instruct | 7.61B | ✔ | 10.00 | 9.60 | 25.60 | 0.40 | 5.20 | 9.20(78.00) |
| Qwen2-72B-Instruct | 72.71B | ✔ | 8.96 | 8.80 | 23.60 | 0.00 | 4.40 | 8.00(81.60) |
| Qwen2-7B-Instruct | 7.07B | ✔ | 8.16 | 7.20 | 20.80 | 0.40 | 2.40 | 10.00(73.60) |
| Meta-Llama-3.1-8B-Instruct | 8B | ✔ | 7.60 | 5.60 | 19.20 | 0.00 | 1.60 | 11.60(72.00) |
| C4ai-Command-R-08-2024 | 32B | ✔ | 7.28 | 5.20 | 15.60 | 0.40 | 2.00 | 13.20(70.80) |
| C4ai-Command-R-Plus-08-2024 | 104B | ✔ | 6.88 | 4.00 | 17.20 | 0.40 | 0.80 | 12.00(66.40) |
| Yi-1.5-9B-Chat | 9B | ✔ | 6.08 | 6.80 | 10.40 | 0.00 | 2.40 | 10.80(71.20) |
| Mistral-7B-Instruct-v0.3 | 7B | ✔ | 4.48 | 2.40 | 8.00 | 0.00 | 0.80 | 11.20(70.80) |

Table 14: Model Performance on KOR-Bench in Zero-Shot Setting with Only Questions.

| Model | Size | Open | Overall | Operation | Logic | Cipher | Puzzle | Counterfactual |
|---|---|---|---|---|---|---|---|---|
| Gpt-4o | * | ✗ | 29.92 | 24.80 | 43.20 | 5.20 | 16.00 | 60.40(19.60) |
| Qwen2.5-72B-Instruct | 72.7B | ✔ | 25.44 | 32.80 | 47.20 | 4.00 | 8.80 | 34.40(54.80) |
| Qwen2.5-32B-Instruct | 32B | ✔ | 24.48 | 29.20 | 43.60 | 4.40 | 7.60 | 37.60(43.60) |
| Mistral-Large-Instruct-2407 | 123B | ✔ | 22.48 | 18.00 | 36.80 | 2.80 | 11.60 | 43.20(30.80) |
| Qwen2-72B-Instruc | 72.71B | ✔ | 21.92 | 24.40 | 44.00 | 6.00 | 7.60 | 27.60(61.20) |
| Yi-Large | * | ✗ | 21.12 | 14.40 | 32.80 | 3.20 | 8.40 | 46.80(21.60) |
| Meta-Llama-3.1-70B-Instruct | 70B | ✔ | 20.08 | 12.40 | 33.60 | 1.20 | 8.00 | 45.20(22.00) |
| DeepSeek-V2.5 | 236B | ✔ | 19.12 | 16.40 | 41.60 | 2.40 | 8.80 | 26.40(53.20) |
| Claude-3.5-Sonnet | * | ✗ | 18.64 | 13.20 | 22.00 | 3.20 | 15.20 | 39.60(28.00) |
| C4ai-Command-R-08-2024 | 32B | ✔ | 15.36 | 12.00 | 27.60 | 2.40 | 3.20 | 31.60(48.40) |
| C4ai-Command-R-Plus-08-2024 | 104B | ✔ | 14.88 | 10.40 | 26.40 | 3.20 | 6.80 | 27.60(54.80) |
| Qwen2.5-7B-Instruct | 7.61B | ✔ | 14.64 | 17.20 | 30.40 | 3.60 | 2.40 | 19.60(64.00) |
| Qwen2-7B-Instruct | 7.07B | ✔ | 14.48 | 14.80 | 30.80 | 2.80 | 3.20 | 20.80(66.80) |
| Yi-1.5-9B-Chat | 9B | ✔ | 14.08 | 15.20 | 26.40 | 2.80 | 3.60 | 22.40(56.80) |
| Mistral-7B-Instruct-v0.3 | 7B | ✔ | 11.44 | 9.60 | 25.60 | 1.60 | 1.60 | 18.80(62.00) |
| Meta-Llama-3.1-8B-Instruct | 8B | ✔ | 10.88 | 2.80 | 13.60 | 0.80 | 0.00 | 37.20(21.20) |

Table 15: Model Performance on KOR-Bench in Three-Shot Setting with Only Questions.

We conduct zero-shot and three-shot "only questions" experiments, without explicit rules, to assess the model's ability to recognize patterns and extract abstract reasoning rules.

**Experimental Setup:**

- **Zero-Shot Setting:** Present the model with a problem without explicit rules or guidance. The model relies solely on its prior knowledge and reasoning to generate an answer.
- **Three-Shot Setting:** Provide the model with three examples and their answers. The model infers a pattern or rule from these, then applies it to solve a new problem.

**Result Analysis:**

- **In the zero-shot setting:** In this setup, models struggle to answer questions accurately because the information provided in the questions, combined with their prior knowledge, is insufficient.
- **In the three-shot setting:** When given three examples, models can infer patterns and solve some problems correctly. The success depends on the task and the relatedness of the examples. For tasks like Counterfactual, Logic, and Operation, where examples are closely connected, models uncover rules and apply them effectively, resulting in higher scores. However, for tasks like Cipher and Puzzle, which involve more abstract rules and weaker correlations between examples, models struggle to deduce the rules, leading to lower scores.

