# OpenReview forum: "KOR-Bench: Benchmarking Language Models on Knowledge-Orthogonal Reasoning Tasks"
_ICLR.cc/2025/Conference — ICLR 2025 Poster_

### Official Review · Reviewer_YDMq · 2024-10-24

**Soundness:** 3
**Presentation:** 4
**Contribution:** 3
**Rating:** 6
**Confidence:** 5

**Summary:**

The paper introduces the concept of Knowledge-Orthogonal Reasoning (KOR), which focuses on evaluating a model's intrinsic reasoning and planning abilities by minimizing reliance on pre-existing knowledge. This is achieved by introducing new rules that are independent of the knowledge the model was exposed to during pretraining. The authors propose the Knowledge-Orthogonal Reasoning Benchmark (KOR-Bench), which includes five task categories: Operation, Logic, Cipher, Puzzle, and Counterfactual. KOR-Bench challenges models by introducing new elements and rules, pushing them beyond traditional reasoning frameworks.

**Strengths:**

1. **Important Direction**: The paper addresses a crucial aspect of evaluation by focusing on how to avoid shortcuts when constructing reasoning benchmarks, which is vital for accurately assessing model capabilities.

2. **Comprehensive Scope**: It involves a substantial workload, providing a thorough examination across various dimensions, ensuring a well-rounded evaluation of model reasoning abilities.

3. **Detailed Presentation**: The benchmark is presented with detailed statistics and information, offering clear insights into its structure and the performance of different models.

**Weaknesses:**

1. **Ensuring Orthogonality**: There is a lack of clarity on how the constructed problems are guaranteed to be orthogonal to the knowledge in LLMs. The paper does not provide evaluation results to support this claim.

2. **Evaluation of Base Models**: The purpose of evaluating base models alongside chat models is unclear. The distinct advantages of base models over chat models are not well articulated.

3. **Lack of Benchmark Comparison**: The study does not compare results with other benchmarks, which makes it difficult to highlight the unique contributions of KOR-Bench.

4. **Benchmark Construction Process**: The process of constructing the benchmark is unclear, particularly whether it was done manually or with LLM assistance.

5. **Missing Related Work**: The paper does not reference a relevant study https://arxiv.org/abs/2306.09479# that proposes a similar concept, which could help in emphasizing the novelty of this work.

Minor comments:
1. **Section 5 Clarity**: The section is somewhat disorganized. The "Reasoning Process Performance" paragraph lacks insights and supportive data, while "Single Task Analysis" is presented without deeper discussion.

2. **Conclusion Content**: The conclusion includes too much focus on future plans, which might be better placed elsewhere in the paper.

3. **Appendix Content**: The appendix contains irrelevant materials, which should be organized more effectively.

5. **Few-Shot Testing**: It would be interesting to test if LLMs can solve problems based on few-shot examples without explicit rules, to assess their ability to derive abstract rules.

6. **Knowledge Orthogonality vs. Contradiction**: The concept of knowledge orthogonality is confused with knowledge contradiction, especially in counterfactual questions.

**Questions:**

See questions in Weaknesses.

---

> ### Author Response · Authors · 2024-11-24
> **Response to reviewer YDMq (Part 1/n)**
>
> ## Response to Weakness 1 & 4
>
> Thank you for your insightful comments and feedback. We will clarify the concept of knowledge orthogonality, how orthogonality is ensured in the task design, and the process of constructing KOR-Bench.
> ### Concept of Knowledge Orthogonality：
>   - Knowledge Orthogonality refers to the independence between the background/domain-specific knowledge (e.g., general knowledge or skills) acquired by the model during pre-training and the rules (i.e., explicitly defined core pieces of information) introduced to solve a particular task. Specifically, knowledge orthogonality requires that the successful solution of a task relies on the model's understanding and reasoning about the task rules, rather than on domain-specific knowledge exposed during pretraining. This independence ensures that the task rules are at the center of the problem solving process, while background/domain-specific knowledge serves only as an aid in the reasoning process.Therefore, the core of data construction lies in adapting and adjusting existing domain knowledge through methods such as introducing new symbols, defining new concepts and terminology, adding execution steps, altering constraints, or incorporating new story contexts, among other adjustments. For details, see **Response to reviewer D3fb (Part 1/n) / Response to Weakness 1**.
> ### Data Construction Process：
>   - The entire data construction process is primarily carried out through **manual annotation**, with the large model used solely for quality verification and difficulty screening. The construction of KOR-Bench involved three main phases: **(1) Rule Design**, **(2) Rule-Driven Q&A Design**, and **(3) Quality Validation**, which included adapting, refining, and modifying the original rules and questions to address specific challenges. For details, see **Response to reviewer D3fb (Part 1/n) / Response to Weakness 2**.
> ### Ensuring Orthogonality in Task Design：
>   - **Ensure from data construction**：To ensure knowledge orthogonality, we carefully designed the task rules through an extensive manual data construction process. The natural language descriptions of the rules are logically self-consistent and serve as core references for solving tasks. Additionally, the redefinition and adaptation of these rules further guarantee their independence from domain-specific knowledge.
>   - **Only the Question, Without the Rule**：To investigate the critical role of rules in problem-solving, we conduct a zero-shot experiment where only the questions were provided, without explicit rules. The results, shown in the table below, reveal significantly low performance across models, emphasizing that rules are essential for effectively solving these problems. Results are presented in the figure: [Only_Question_Analysis(Zero_Shot and Three_Shot)](https://anonymous.4open.science/r/kor-bench-rebuttal-repo-44F6/images/only_question.pdf)  and prompt is shown in [here](https://anonymous.4open.science/r/kor-bench-rebuttal-repo-44F6/config/prompt/zero-shot-onlyquestions.yaml).

---

> ### Author Response · Authors · 2024-11-24
> **Response to reviewer YDMq (Part 2/n)**
>
> ## Response to Weakness 2
>
> We are grateful for  your thoughtful comments and feedback regarding the evaluation of base models alongside chat models.The reasons for evaluating base models are as follows:
> ### (1) Exploring Alignment's Impact on Reasoning Abilities
> We aim to explore whether alignment improves the reasoning capabilities of models, which aspects are enhanced, and whether any trade-offs occur. For example, the results show that aligned models exhibit performance improvements across all KOR-Bench subsets, with particularly significant gains in the "operation" category:
> - Qwen2-72B(-Instruct)：34.0(78.0)
> - Yi-1.5-34B(-Chat)：24.8(79.6)
> - Qwen2-7B(-Instruct)：20.4(28.8)
> - MAP-Neo-7B(-Instruct)：7.2(38.4)
>
> This is likely because many of the rules involve newly defined mathematical operations, as similar trends are observed in MATH:
> - Qwen2-72B(-Instruct)：50.9(59.7)
> - Yi-1.5-34B(-Chat)：41.0(50.1)
> - Qwen2-7B(-Instruct)：44.2(49.6)
> - MAP-Neo-7B(-Instruct)：34.9(44.4)
>
> However, such improvements are not observed in benchmarks like MMLU, which emphasize the model's inherent prior knowledge:
> - Qwen2-72B(-Instruct)：84.2(82.3)
> - Yi-1.5-34B(-Chat)：77.1(76.8)
> - Qwen2-7B(-Instruct)：70.3(70.5)
> - MAP-Neo-7B(-Instruct)：58.14(58.28)
>
> We hypothesize that the computational abilities activated during alignment are general-purpose and not strongly dependent on prior knowledge. The computational rules in KOR-Bench are newly defined and are unlikely to exist widely in pretraining corpora. Alignment likely makes models better at identifying relationships between numbers. *The other benckmark results mentioned above are sourced from the official evaluations of the models*. For more relevant results, please see this [table](https://anonymous.4open.science/r/kor-bench-rebuttal-repo-44F6/kor-corr.csv).
> ### (2) Developing a General-Purpose Benchmark
> Our goal is to develop a highly general-purpose benchmark. The ability to evaluate base models using few-shot settings is essential for us, as it allows researchers to perform additional analyses, such as the impact of alignment mentioned in the (1).
>
> ## Response to Weakness 3
>
> Thank you for your valuable feedback regarding the lack of benchmark comparison in our study. We fully understand the significance of comparing KOR-Bench with other benchmarks to better highlight its unique strengths and contributions.
> **Unique Features of KOR-Bench**: Compared to other benchmarks, the most distinctive feature of KOR-Bench lies in its knowledge-orthogonal nature, enabling it to evaluate the fundamental reasoning abilities of models more effectively. Additionally, since all rules and questions are newly designed by humans, the risk of data leakage is minimized. Its construction, with rules and questions designed separately, also gives it strong scalability.
> **Correlations with Other Benchmarks**:
> - **Calculating Correlations**：We further calculate the correlations between KOR-Bench and several benchmarks, including MMLU, MMLU-Pro, GSM8K, MATH, HumanEval, and MBPP. The correlations measure the similarity in score distributions of various models across different benchmarks. We calculate these using 21 models of varying sizes, including GPT4o, Claude-3.5-Sonnet, and the Qwen, Llama, Yi, Mistral, Phi, and Gemma series.
> - **Highest Correlation with MMLU-Pro**：
>   The results, shown in the [figure](https://anonymous.4open.science/r/kor-bench-rebuttal-repo-44F6/images/heatmap_overall.pdf), indicate that KOR-Bench is more closely related to reasoning-focused benchmarks and shows the highest correlation with MMLU-Pro, the newest and most challenging version of MMLU. The main difference between MMLU-Pro and MMLU is that MMLU-Pro places a greater emphasis on reasoning ability[[1]](https://www.reddit.com/r/LocalLLaMA/comments/1du52gf/mmlupro_is_a_math_benchmark/). It includes a large number of computational problems that require strong logical reasoning to solve [[2]](https://www.reddit.com/r/LocalLLaMA/comments/1du52gf/mmlupro_is_a_math_benchmark/).
>     [1] https://huggingface.co/datasets/TIGER-Lab/MMLU-Pro#1-whats-the-difference-between-mmlu-pro-and-mmlu
>     [2] https://www.reddit.com/r/LocalLLaMA/comments/1du52gf/mmlupro_is_a_math_benchmark/
>
> The difference in the correlation between KOR-Bench and these benchmarks aligns with KOR-Bench's stated focus on emphasizing logical reasoning while minimizing reliance on prior knowledge. This demonstrates KOR-Bench’s alignment with reasoning-oriented benchmarks, further validating its design focus. The data used for the correlation calculation can be viewed in this [table](https://anonymous.4open.science/r/kor-bench-rebuttal-repo-44F6/kor-corr.csv).

---

> > ### Comment · Reviewer_YDMq · 2024-11-27
> > **response to authors**
> >
> > These two responses addressed my questions.

---

> ### Author Response · Authors · 2024-11-24
> **Response to reviewer YDMq (Part 3/n)**
>
> ## Response to Weakness 5
>
> Thanks for the heads up on our lack of relevant work.
>
> The paper [1] you mentioned highlights the issue of strong priors in models, where they tend to rely on general patterns learned from large-scale text corpora rather than adjusting to task-specific instructions. While large language models perform well across many tasks, they may still over-rely on prior knowledge from training data and lack flexibility in adapting to new contexts. The task setting described in Section 3.1.3 (Redefine) of [1], which involves redefining common symbols or words, is related to the knowledge-orthogonal task design we propose.
>
> It is precisely because of such strong priors that we introduced the concept of decoupling knowledge and rules, in order to reduce the model's dependency on pre-trained data and enable better adaptation to new definitions and rules.
> We have added a citation to the article in the paper.
>
> [1] McKenzie I R, Lyzhov A, Pieler M, et al. Inverse scaling: When bigger isn't better[J]. arXiv preprint arXiv:2306.09479, 2023.
>
>
> ## Response to Minor comments 1 & 2 & 3
>
> Thank you for your valuable suggestions regarding the organization and content of this article. We have reorganized and optimized the content of these sections, and the changes are highlighted in blue.
>
> ## Response to Minor comments 4
>
> We truly appreciate the insightful suggestion to explore few-shot testing in LLMs. This idea is fantastic and could provide valuable insights into the models' ability to generalize and derive abstract rules from limited examples. Testing LLM's ability to discover and summarize rules on its own is more relevant to the needs of applications in real scenarios.
> We conduct zero-shot and three-shot 'only questions' experiments.
> ### Experimental Setup:
> - **Zero-Shot Setting**: Present the model with a problem without any explicit rules or guidance. Let it rely solely on its prior knowledge and reasoning to generate an answer. Results are presented in the figure: [Only_Question_Analysis(Zero_Shot and Three_Shot)](https://anonymous.4open.science/r/kor-bench-rebuttal-repo-44F6/images/only_question.pdf) and prompt is shown in [here](https://anonymous.4open.science/r/kor-bench-rebuttal-repo-44F6/config/prompt/zero-shot-onlyquestions.yaml).
> - **Three-Shot Setting**: Provide the model with three examples and their corresponding answers. Allow it to infer a pattern or rule from these, then use that rule to solve a new problem. Results are presented in the figure: [Only_Question_Analysis(Zero_Shot and Three_Shot)](https://anonymous.4open.science/r/kor-bench-rebuttal-repo-44F6/images/only_question.pdf) and prompt is shown in [here](https://anonymous.4open.science/r/kor-bench-rebuttal-repo-44F6/config/prompt/three-shot-onlyquestions.yaml).
> ### Result Analysis
> - **In the zero-shot setting**: In this setup, models struggle to answer questions accurately because the information provided in the questions, combined with their prior knowledge, is insufficient.
> - **In the three-shot setting**: When given three example questions and answers, models can infer patterns and solve some problems correctly. The success of this approach depends on the task and how closely related the examples are. For tasks like Counterfactual, Logic, and Operation, where examples are more connected, models can uncover the key rules and apply them effectively, leading to higher scores. However, for tasks like Cipher and Puzzle, with more abstract rules and less correlation between examples, models struggle to deduce the rules, resulting in lower scores.

---

> > ### Comment · Reviewer_YDMq · 2024-11-27
> > **response to authors**
> >
> > These 3 responses addressed my questions.

---

> ### Author Response · Authors · 2024-11-24
> **Response to reviewer YDMq (Part 4/n)**
>
> ## Response to Minor comments 5
>
> We highly appreciate the concept of knowledge conflict you mentioned.
>
> The primary reason for including counterfactual tasks in our evaluation set is that we believe the most critical aspect of designing knowledge-orthogonal tasks lies in **separating the influence of a model's internal prior knowledge** when defining new rules. **Each counterfactual task evaluates the ability to counter prior knowledge**.
>
> However, we must admit that this approach is *somewhat aggressive*, as it does not completely eliminate the influence of internal prior knowledge. Instead, it may introduce conflicts, with the expectation that the model autonomously chooses to follow the rules defined in the task. Despite this, we still consider counterfactual tasks a form of knowledge orthogonality.
>
> Furthermore, we emphasize that large language models (LLMs) should strictly adhere to and rely on the newly defined instructions in KOR-Bench to answer questions, rather than relying on their internal prior knowledge. Therefore, counterfactual tasks are well-suited to testing whether LLMs can rely solely on newly defined instructions to provide answers. We believe the ability of a model to follow instructions is crucial for solving knowledge-orthogonal tasks.

---

> > ### Comment · Reviewer_YDMq · 2024-11-27
> > **response to authors**
> >
> > This response needs to be clarified. I still dont understand why you "still consider counterfactual tasks a form of knowledge orthogonality" given you admit "it does not completely eliminate the influence of internal prior knowledge. Instead, it may introduce conflicts"? what's your logic?
> >
> > Besides, the last paragraph seems was generated by chatgpt automatically because it does not reads logical coherent with text before. I suggest authors to revise this response.

---

> ### Author Response · Authors · 2024-11-26
> **Response to reviewer YDMq (Part 5/n)**
>
> ## Response to Weakness 1
> We have reconsidered the definition of "Knowledge Orthogonality" and refined it based on new insights. Please refer to **Response to Reviewer D3fb (Part 5/n)** for the updated formal definition and further details.

---

> > ### Comment · Reviewer_YDMq · 2024-11-27
> > **response to authors**
> >
> > Thanks for the updating message.
> > 1. I **did not** ask for the clarification of the definition of Knowledge Orthogonality, so please **do not** copy large chunks from your responses to other reviewers.
> > 2. Personally, even I totally understand the information you want to convey, I don't like your formulas in the Formal Definition of Knowledge Orthogonality in your response to Reviewer D3fb (Part 5/n) . Is it necessary to make such a simple definition so complicated just because it looks more sophisticated and fancy? It does not provide more convincing justification for the concept of Knowledge Orthogonality.
> > 3. What I want to know is: How do you quantify the orthogonality of your benchmark with pretrained knowlwdge in LLMs? For example, with an ideal metric of orthogonality, the MMLU would score $0^\circ$, your benchmark would score $90^\circ$, and counterfactual questions would score $180^\circ$ with this metric.
> >
> > I would be appreciate and consider raising my rating if you address the above questions. Thanks.

---

> ### Author Response · Authors · 2024-11-28
> **Response to reviewer YDMq (Part 6/n)**
>
> Thank you for your feedback and engagement! We'd like to further clarify your concerns:
>
> 1. **For an Ideal Metric of Orthogonality:** Ideally, we could conceivably correspond knowledge, reasoning, and counter-knowledge to 0°, 90°, and 180°, respectively. However, in practice, LLMs are pre-trained on billions of documents, which makes entirely decoupling Reasoning from Knowledge a hard nut to crack.
>
> 2. **A Relatively Broad Concept:** The “orthogonality” in our paper is a **relatively broad concept**. We try to construct a benchmark where: 1). We try our best to rule out the influence of pre-trained knowledge of LLMs; 2). The reasoning capability is not very related to pre-trained knowledge; 3). Increasing more pre-training data is not that helpful to improve the reasoning performance on the benchmark. Note that most of the existing widely adopted benchmarks like MMLU/MMLU-Pro, GSM8K/MATH, HumanEval/MBPP cannot meet these requirements. Most of these tasks have strong relevance to knowledge, and most models typically hack a true evaluation of reasoning ability by simply adding more relevant data. To meet these requirements, we craft the redefined and adapted rules and questions through an extensive manual data construction process ensuring that these new questions do not appear in the pre-training data.
>
> 3. **Experimental Results:** In experiments where only questions are provided, we assess the models' performance by relying solely on their intrinsic knowledge. The results in Table 14: [Only_Question_Analysis(Zero_Shot and Three_Shot)](https://anonymous.4open.science/r/kor-bench-rebuttal-repo-44F6/images/only_question.pdf) show that the models struggle to answer correctly, with generally low accuracy and only a few correct answers by chance. As shown in Table 14, the values in parentheses represent the proportion of real-world answers in the counterfactual setting. The high proportion of real-world answers suggests that the model's prior knowledge interferes with accurate responses.
>
> 4. **For the Formal Definition:** We carry out the formal definition of "Knowledge Orthogonality" not to be fancy, but to ensure rigorous precision. To ensure clarity throughout, we begin the setting by defining the informational components of a reasoning task, followed by an explanation of the formal notation. Based on these symbols, we further explain the three core properties in knowledge orthogonality. Finally, the impact of background knowledge (K) on the derivation of the answer (A) is quantified by introducing a β-value, which ensures the rigour and systematicity of the definition.

---

> ### Comment · Reviewer_YDMq · 2024-11-29
> **response to authors**
>
> Thank you for your feedback. Based on your suggestion, the $\beta$-value could indeed serve as a potential metric for orthogonality. However, the authors seem not to have deeply considered these related concepts, which is why they missed the inherent connection.
>
> To illustrate this, let's examine your point in **For an Ideal Metric of Orthogonality**:
> "However, in practice, LLMs are pre-trained on billions of documents, making it challenging to completely separate Reasoning from Knowledge."
>
> Now, compare this with your statement about the $\beta$-value:
> "Background knowledge $K$ may support or interfere with the derivation from $A$ to $Q$... The extent of this influence is quantified by the Knowledge Impact Factor $\beta$."
>
> These two statements clearly contradict each other. Based on this, I will maintain my rating until the authors provide a satisfactory answer.

---

> ### Author Response · Authors · 2024-11-29
> **Response to reviewer YDMq (Part 7/n)**
>
> Thanks for your feedback!
> 1. **Definability versus Non-quantifiability of Orthogonality**：
>     1. In your question, *"How do you quantify the orthogonality of your benchmarks to the pre-trained knowledge in LLMs?"*  we initially understand it as you asking us to define a metric that is both practical and computationally feasible, rather than a purely conceptual metric; in fact, this metric is theoretically definable, but practically uncomputable. Nevertheless, we recognize that our explanation may not fully align with your intent and might have omitted some of the intermediate reasoning steps, which could have led to a misunderstanding. We will provide a more detailed explanation.
>     2. In our response, we point out:
>         > Ideally, we could conceivably correspond knowledge, reasoning, and counter-knowledge to $0°$, $90°$, and $180°$, respectively.
>         - What we mean is that we think this metric is theoretically definable. In KOR Bench, the domain of $\beta$ is extendable, ranging from $(-1,\epsilon]$ to $(-1,1)$, to cover the entire cognitive range from reliance solely on knowledge, and reliance solely on reasoning, to counter-knowledge. We define it here as $\cos\theta$, which aligns with the angle definition you described. This metric can be defined as follows:
>           - $$\cos\theta = \frac{P(Q\rightarrow A\mid R, K) - P(Q\rightarrow A\mid R)}{P(Q\rightarrow A\mid R)}$$
>             - $\cos \theta \in (\epsilon,1), \text{where} \quad \theta \in (0^\circ, 90^\circ)$: In KOR Bench, $\cos\theta$ belongs to the range $(-1, \epsilon)$, which goes beyond the defined range here: This indicates that solving the task will rely on K strongly, and the model's performance will increase due to the hack of pre-trained knowledge.
>             - $\cos \theta \to 0, \text{where} \quad \theta \to 90^\circ$: This indicates that the influence of K on solving the task is minimal, and the task is primarily solved through rule-based reasoning.
>             - $\cos \theta \in (-1,-\epsilon),\text{where} \quad \theta \in (90^\circ,180^\circ)$: This indicates that knowledge interferes with solving the task, and the model's performance actually decreases due to the influence of the model's prior knowledge.
>           - The knowledge orthogonality we propose excludes the portion that strongly depends on knowledge, which aligns with the range defined by $\beta$.
>
>         > However, in practice, LLMs are pre-trained on billions of documents, which makes entirely decoupling Reasoning from Knowledge a hard nut to crack.
>         - In the evaluation of the model, we observe the following points: First, $P(Q\rightarrow A\mid R, K)$ reflects the model's performance when both prior knowledge and rules are available. In contrast, $P(Q\rightarrow A\mid R, K) - P(Q \rightarrow A \mid R)$  can be represented in a manner that does not explicitly rely on the rules (as illustrated in Table 14). However, $P(Q \rightarrow A \mid R)$ cannot be effectively measured, as it is not possible to evaluate the model's performance when it is solely influenced by the rules. In conclusion, although the proposed orthogonality metric is theoretically definable, it is not practically computable.
>
>          > Finally, the impact of background knowledge $(K)$ on the derivation of the answer $(A)$ is theoretically definable through the introduction of a $β$, which ensures the rigor and systematicity of the definition.
>         - We apologize for any confusion caused by our use of the term "quantified". What we intend to convey is that the relationship is theoretically definable, rather than something that can be directly measured in practice.
> 2. **Focus of Our Work**: We do not aim for a completely idealized orthogonality. Instead, we focus on a **loosened knowledge orthogonality**, as we consider the ideal version unrealistic. In our definition, we explicitly state that $β$ ranging from -1 to very small values are acceptable, so our definition is consistent with our motivation.
> 3. **Practical Considerations**: There is no need to measure reasoning ability that is completely independent of knowledge, as this is also impractical. What we aim to measure is reasoning ability that is  **as minimally hacked by pre-trained knowledge as possible and enhanced by reasoning itself**.
> 4. **Experimental Results**: In terms of the internal definitions of our paper, our experimental results, as mentioned above, align with our definitions. The results show that large language models must rely on rules to achieve better performance.
>
> In summary, the key difference in understanding lies in our approach: we are not pursuing an ideal orthogonality but a loosened one that better reflects the generalized reasoning ability of large language models beyond pre-existing knowledge. We believe that KOR Bench sufficiently meets our motivation. We look forward to your careful review of the provided self-consistent definition and await your satisfactory response.

---

> > ### Comment · Reviewer_YDMq · 2024-11-30
> > **response to authors**
> >
> > Thanks for your reponse. I appreciate your efforts to clarify to my quesions. I have no further questions currently. I will raise my score as I have promised.

---

### Official Review · Reviewer_4jdU · 2024-10-27

**Soundness:** 3
**Presentation:** 4
**Contribution:** 4
**Rating:** 8
**Confidence:** 4

**Summary:**

The paper introduces KOR-Bench, a benchmark specifically designed to evaluate the reasoning and problem-solving abilities of LLMs independent of their pre-existing, pretrained knowledge. This benchmark employs a “Knowledge-Orthogonal Reasoning” approach to assess LLMs across five challenging task categories: Operation, Logic, Cipher, Puzzle, and Counterfactual Reasoning. Each task type introduces new rules and problem-solving frameworks that are orthogonal to domain-specific knowledge, focusing on the model’s ability to reason based on novel rules rather than relying on memorized patterns.

**Strengths:**

1. The introduction of knowledge orthogonality is an innovative approach to eliminate biases from pre-existing knowledge, offering a fresh perspective on evaluating true reasoning capabilities in LLMs.
2. By incorporating diverse tasks such as logical puzzles, cryptographic challenges, and hypothetical scenarios, KOR-Bench provides a robust testbed for examining LLMs’ adaptability to unfamiliar rules.
3. This paper conducts comprehensive experiments and provides a thorough analysis of model errors, identifying specific challenges within task sub-steps. This analysis offers valuable insights into model limitations and potential areas for improvement.

**Weaknesses:**

1.	There are already existing benchmarks for evaluating the similar capabilities of current LLMs. For example, as mentioned in the related work, we have logic games [1] for logical reasoning and datasets like [2] for counterfactual reasoning. If the authors argue that these datasets might have been included in current model training data, it would be helpful to explain how KOR-Bench avoids this, such as by concealing the test set or employing other safeguards.
2.	The data scale of KOR-Bench may be a concern, as the current version includes only a small dataset for each task category, potentially limiting the robustness and generalizability of the benchmark findings.

****References****

[1] Gui J, Liu Y, Cheng J, et al. Logicgame: Benchmarking rule-based reasoning abilities of large language models[J]. arXiv preprint arXiv:2408.15778, 2024.

[2] Wu Z, Qiu L, Ross A, et al. Reasoning or Reciting? Exploring the Capabilities and Limitations of Language Models Through Counterfactual Tasks[C]//Proceedings of the 2024 Conference of the North American Chapter of the Association for Computational Linguistics: Human Language Technologies (Volume 1: Long Papers). 2024: 1819-1862.

**Questions:**

I have no further questions; the analysis in the main body and appendix is highly comprehensive.

---

> ### Author Response · Authors · 2024-11-24
> **Response to reviewer 4jdU (Part 1/n)**
>
> ## Response to Weakness 1
>
> Regarding the comparison with existing benchmarks, we will explain the relevance and differences between KOR-Bench and these two efforts, and suggest some avoidance measures for data leakage situations.
> ### Comparison with two efforts：
>   - **For LogicGame[1]**：It focuses on evaluating models' execution and planning reasoning, primarily within predefined game rules and initial conditions. This work belongs to the same period of time as KOR-Bench, and begins with instruction following emphasizing the rule-following ability of the assessment model, but is limited to tasks within the scope of logic games.
>   - **For Counterfactual_evaluation[2]**：This dataset includes 11 counterfactual reasoning tasks, where the same reasoning process is applied between default conditions and counterfactual variations, with a focus on input-output mapping differences, particularly emphasizing counterfactual reasoning. However, it does not assess rule-following abilities.
>   - **Innovative Aspects of KOR-Bench**：The main difference between [1] and [2] and KOR-Bench lies in the type of tasks, where the settings of logic games and counterfactual scenarios should be considered as a subset of KOR-Bench. By introducing the concept of knowledge orthogonality, KOR-Bench aims to incorporate newly defined domain-specific knowledge from different domains, enabling rule-based assessment of reasoning in out-of-distribution scenarios decoupled from the knowledge base. For details, see **Response to reviewer D3fb (Part 1/n) / Response to Weakness 1**.
> ### Measures to prevent data leakage：
>   - **Manual Redefinition of Rules and Questions**: All rules in KOR-Bench are carefully redefined to ensure the originality of the tasks, effectively avoiding overlap with existing training data and thereby reducing the risk of data leakage. The detailed data construction process can be found in **Response to reviewer D3fb (Part 2/n) / Response to Weakness 2**.
>   - **LLM Quality Validation**: During the data construction process, we continuously monitor the quality of generated questions using LLMs. Questions that are easily answered by most LLMs are discarded.
>
> ## Response to Weakness 2
>
> Same as reviewer D3fb's question. For details, see Response to reviewer D3fb (Part 4/n) / Response to Weakness 4 / **Ablation Study on Dataset Size**

---

> > ### Comment · Reviewer_4jdU · 2024-11-26
> >
> > Thanks for your response. I have no further questions and will maintain my score.

---

### Official Review · Reviewer_MyDX · 2024-10-29

**Soundness:** 3
**Presentation:** 3
**Contribution:** 3
**Rating:** 8
**Confidence:** 3

**Summary:**

This paper introduces the concept of Knowledge-Orthogonal Reasoning (KOR), aiming to reduce the interference of existing pre-trained knowledge by introducing new rules that are orthogonal to it, thus allowing for a more precise assessment of a model's intrinsic reasoning capabilities. Based on this concept, the paper proposes the Knowledge-Orthogonal Reasoning Benchmark (KOR-Bench), which covers five task categories: Operation, Logic, Cipher, Puzzle, and Counterfactual. The benchmark is designed to evaluate how well models can apply new rule descriptions to solve novel, rule-driven problems. The study reveals that even state-of-the-art models, such as Claude-3.5-Sonnet and GPT-4o, perform far from optimally on KOR-Bench, achieving accuracies of 58.96% and 58.00%, respectively. The paper also conducts in-depth analyses using methods like Stepwise Prompting to identify performance bottlenecks in specific tasks, such as the Cipher task, and explores the impact of self-correction techniques and tricks on puzzle-solving, along with visualizing rule-focused attention. These contributions provide new perspectives and tools for assessing and enhancing models' reasoning abilities, fostering further research in the field.

**Strengths:**

1. This paper proposes a new method for testing the reasoning capabilities of large models and conducts extensive experiments and tests using both open-source and proprietary large models across multiple aspects. During the assessment, this work reduces the impact of pre-training knowledge. Most existing benchmarks, such as MMLU, GPQA, and CommonsenseQA, primarily evaluate a model's ability to accumulate and recall data, often making it difficult to distinguish whether the model is performing genuine reasoning or simply recalling learned patterns. KOR-Bench reduces this dependency by introducing new rules independent of existing pre-trained knowledge, thereby more accurately assessing the model's intrinsic reasoning and planning capabilities.
2. This assessment is more comprehensive and challenging. KOR-Bench covers a variety of reasoning scenarios and task types, including operations, logic, cryptography, puzzles, and counterfactual reasoning across five categories. These tasks are designed to be novel and challenging for models, testing their ability to handle unseen rules and frameworks. Even the most advanced models achieve relatively low accuracy rates (e.g., Claude-3.5-Sonnet scores 58.96%, and GPT-4o scores 58.00%), demonstrating its significant challenge and discriminative power.
3. The authors provide a comprehensive and detailed introduction to the code and data through appendices and open-source releases.
4. The tables and graphs in this article are clear and beautiful.

**Weaknesses:**

1. There are clerical errors in some parts of the article, such as the model name "Clause-3.5-Sonnet" on lines 308 and 309, and it is critical to be rigorous when doing scientific research.
2. This paper lacks theoretical and experimental evidence on how to ensure the objectivity and impartiality of the assessment.

**Questions:**

1. Can the authors provide more experimental evidence or theoretical proof to ensure the fairness and objectivity of the model evaluation?
2. How does the author intend to implement the plan proposed in line 539, and briefly explain how the results of this paper can be extended to multimodality?

---

> ### Author Response · Authors · 2024-11-24
> **Response to reviewer MyDX (Part 1/n)**
>
> ## Response to Weakness 1
>
> Thank you for pointing out the clerical errors in the model names. We sincerely apologize for the oversight and have addressed these issues in the revised submission. We greatly appreciate your attention to detail and valuable feedback!
>
> ## Response to Weakness 2 & Question 1
>
> Thank you for your feedback and agree that ensuring fairness and objectivity in model assessment is a key aspect. To address your concerns, we ensure consistency in settings across all evaluations during inference and evaluation.
> - **Consistency in Prompts and Model Parameters**: Appendix B outlines the evaluation prompts used for the five task categories in KOR-Bench. We ensure consistency across all evaluation settings, including the prompts and model parameter settings during the reasoning and assessment processes.
> - **Robustness in Answer Extraction and Evaluation**: Appendix A.2 provides detailed specifications regarding the answer format in KOR-Bench, while Section 4.2 and Appendix A.4 explain the evaluation rules in detail, including specific treatments for different types of answers, such as Multiple Answer Handling, Mathematical Expression Handling, and Unordered List Handling.
> - **Availability of Evaluation Code and Results**: Furthermore, the evaluation code, along with all inference and evaluation result files, is available in the anonymous repository for review. We believe that the robustness of our evaluation rules ensures accuracy, particularly in handling various answer types.
>
> In addition, we also conducted data and size ablation experiments to prove the robustness of the current dataset evaluation. For details, see Response to reviewer D3fb (Part 4/n) / Response to Weakness 4 / **Ablation Study on Dataset Size**
>
> ## Response to Question 2
>
> Thank you for your interest in future expansion plans, which is exactly the direction we intend to continue to move in.
>
> We will expand around the notion of knowledge orthogonality. Games are a classic system of rules that are a good embodiment of knowledge orthogonality, but they are better suited to the inclusion of visuals for assessment. In many knowledge disciplines, the combination and interaction of many rules builds a deep theoretical foundation, and these rules themselves can be decoupled and redefined for more complex assessment and analysis. For example, in the field of programming, we can define some new program rules and ask the model to write code corresponding to these rules; in linguistics, we can define some new grammatical rules and ask the model to write sentence structures or grammatical rules conforming to these new rules; in law, we can redefine the existing terms and ask the model to analyze under the new terms. Human society itself has evolved in a continuous process of creating and revising rules. Based on this thinking, our expansion plan includes the following aspects:
> - **Task Category Expansion**: We aim to incorporate new domain knowledge from various fields that introduce the concept of knowledge orthogonality, thereby providing richer perspectives and scenarios for the dataset.
> - **Rule and Rule-Driven Q&A Expansion**: We will introduce more rules under each category, particularly by using formal languages (such as Game Description Language, GDL, and Knowledge Interchange Format, KIF) or by writing code to define and adapt rules, which will assist in question generation and answer validation.
> - **Multimodal Expansion**: Some game rules and interaction designs will serve as excellent data sources. After appropriate modifications, these can support the construction of multimodal datasets. Similarly, graphical knowledge from fields like mathematics and physics, once redefined, can also provide valuable multimodal data. These new rules and definitions not only enrich the semantic layers of the data but also bring innovative perspectives to various types of data processing tasks.
>
> Some examples of potential data sources, primarily focused on game rules (with other knowledge domains still being explored), include:
> https://www.officialgamerules.org/
> https://gamerules.com/index/
> https://www.chess.com/play
> https://www.ultraboardgames.com/category/index.php
> https://www.mastersofgames.com/rules/dominoes-rules.htm
> https://www.pagat.com/

---

> > ### Comment · Reviewer_MyDX · 2024-11-26
> >
> > Thanks for your detailed responses. They have addressed my concerns, and thus I will maintain my score.

---

### Official Review · Reviewer_D3fb · 2024-11-05

**Soundness:** 3
**Presentation:** 3
**Contribution:** 3
**Rating:** 6
**Confidence:** 4

**Summary:**

This paper presents KOR-Bench, a new benchmark that tests LLMs' reasoning abilities across five categories: Operation, Logic, Cipher, Puzzle, and Counterfactual. The key innovation in the design of the benchmark is the concept "knowledge orthogonality", which refers to the property of problems requiring minimal dependence on the domain-specific knowledge the model is exposed to during training. The authors maintain that their knowledge orthogonal tasks can examine how LLMs reason about new rules, concepts, and situations, thus constituting a good testbed for the out-of-distribution reasoning abilities of LLMs. A large number of LLMs, including both chat and base models are evaluated on this benchmark. While SoTA models such as Claude 3.5 and GPT-4o perform the best, all models have large room for improvement. In particular, the cipher and puzzle categories are especially challenging for LLMs based on the evaluation results. The authors also carried out fine-grained and targeted analysis to better understand model behaviors and error patterns. Overall, the experimental results contribute to diagnosing current model limitations, and the benchmark may help assess reasoning capabilities of future models.

**Strengths:**

- The presentation of the paper is generally clear and thorough. It provides good examples of the categories and problems, reasonable details of the benchmark statistics and creation process, and informative tables and figures.

- Benchmarking reasoning in reliable ways is an important topic and is timely, which this work targets.

- The proposed benchmark is fairly diverse with respect to reasoning. Some of the five categories are related, but they together cover a wide range of reasoning tasks.

- The authors conducted comprehensive evaluations across many models. I appreciate the effort of running models of different sizes and different kinds (open vs closed, chat vs base). The results suggest that most of the benchmark is challenging for most of the models.

**Weaknesses:**

- The biggest weakness of the current version of the paper is that the notion of "knowledge orthogonality" is not well articulated enough. There are two aspects. One is about what the notion means. The authors write it "means that the rules within the
benchmark are independent of the domain-specific knowledge the model is exposed to during pre-training" (Lines 76-77). What do you mean by "independent" here? Does it just mean "likely absent"? Could you clarify this central notion more in-depth? The other aspect is whether the five categories satisfy knowledge orthogonality. For example, one could argue that the Puzzle example in Figure 2 is basically the minesweeper game, which is a famous and popular game LLMs know a lot about. I am wondering how you would argue that is "knowledge orthogonal". Moreover, in the Counterfactual category, most of the cover stories are culturally well-known (e.g., Pokemon and Avengers). In what sense are they "independent" of pre-training data?

- Relatedly, in Section 3.2 the authors mention that the raw rules/problems taken from other sources are "adjusted/refined/modified/etc." for this benchmark. How was that process done? What did the authors do to ensure that the end results "meet the specific challenges of KOR-Bench"? It would be good to see more explanations, since this is important given the nature of the proposed benchmark.

- For the evaluation, in terms of prompting strategies this work only studies zero-shot and few-shot settings. We know that prompting matters for LLM reasoning, and there are many methods that can improve reasoning performance (e.g., most notably chain-of-thought/CoT). It would have been a stronger paper if the authors had evaluated the problems with at least one more sophisticated method. Or, if CoT sometimes/often happens automatically and the authors intend that to the case, it should be discussed explicitly in the paper.

- The size of the benchmark is not particularly large, as the authors note. It is still good, and the authors indicate the plan to expand it.

- I think the writing of Section 6 can be improved, especially 6.2 and 6.3. "Self-correction" is introduced rather abruptly with minimal setup/context, and I am not sure that I fully understand what "complex task processing" is about. I recommend that the authors elaborate on those analyses.

**Questions:**

See the "Weaknesses" section.

---

> ### Author Response · Authors · 2024-11-24
> **Response to reviewer D3fb (Part 1/n)**
>
> ## Response to Weakness 1
>
> Thank you for your thoughtful and detailed feedback regarding the concept of "knowledge orthogonality." We sincerely apologize for not providing a sufficiently clear articulation of this core concept in the current version of the paper. Below, we offer a more comprehensive explanation to address your concerns and clarify our intentions.
> ### Clarification of "Knowledge Orthogonality"
>   - Knowledge Orthogonality refers to the independence between background/domain-specific knowledge (e.g., general knowledge or skills acquired during pre-training) and the rules explicitly defined to solve a particular task. It ensures that task-solving relies on understanding and reasoning about the task rules, while background knowledge only aids the reasoning process.
>   - Formal Definition: For a task $T$, the required reasoning information consists of:
>     - $K$: General background/domain-specific knowledge acquired during pre-training, excluding common sense.
>     - $R$: Core rule information designed to solve $T$.
>     - $Q$: A Rule-driven question
>     - $A$: Answer to the question $Q$.
>   - $T$ satisfies knowledge orthogonality under the following conditions:
>     1. Knowledge-Rule Decoupling: Rule $R$ is logically self-contained and independent of $K$.
>       - $R \perp K$
>     2. Knowledge Assistiveness: Background knowledge $K$ supports reasoning but does not determine the answer.
>       - $P(Q→A∣R,K)≫P(Q→A∣K)$
>     3. Rule Centrality: Correctness relies on understanding and applying $R$ with $R$ having significantly greater influence than $K$.
>       - $P(Q→A∣R,K)≈P(Q→A∣R)≫P(Q→A∣K)$
> ### Deeper Interpretation of "Independence"
>   - Independence means that rule $R$ has logical consistency and is clearly defined without reliance on $K$. While rules might resemble patterns in pretraining data, solving the task depends on reasoning with $R$, not recalling prior knowledge.
>   - By decoupling rules from background knowledge, knowledge orthogonality minimizes interference from $K$, ensuring task solutions rely on rules rather than pre-existing knowledge. This approach generalizes across domains, as abstract rule combinations form the foundation for theoretical reasoning.
> ### Determination of Orthogonality in Task Types
>   1. Puzzle Reasoning (e.g., Minesweeper): Although the mechanics resemble Minesweeper, the rules are redefined using symbolic forms such as "X." Solutions depend entirely on these redefined rules, independent of prior Minesweeper knowledge.
>   2. Counterfactual Reasoning (e.g., familiar cover stories): Familiar elements like Pokémon or The Avengers are decorative and irrelevant to reasoning. The task focuses on counterfactual logic and applying introduced rules, independent of background story knowledge.
>   3. While tasks may superficially resemble familiar contexts, solutions rely on reasoning with the provided rules. Background knowledge is neither necessary nor sufficient for solving these tasks. Minimizing similarity between background knowledge and task rules further ensures orthogonality, as demonstrated in examples with minimal overlap.

---

> ### Author Response · Authors · 2024-11-24
> **Response to reviewer D3fb (Part 2/n)**
>
> ## Response to Weakness 2
>
> Thank you for pointing this out. We apologize for the scattered presentation of the data construction process across various task categories in the paper, which resulted in an insufficiently clear explanation of the overall process.
> The process of adapting, refining, and modifying the original rules and questions to address the specific challenges of KOR-Bench followed three main phases: **(1) Rule Design**, **(2) Rule-Driven Q&A Design**, and **(3) Quality Validation**.The entire data process is primarily completed through manual annotation, with large models used only for quality validation and difficulty filtering.
> 1. **Rule Design**
>     - **Rule Extraction**: Core rules are extracted from logic puzzles, textbooks, domain knowledge, or virtual world settings and defined as natural language descriptions.
>     - **Rule Redefinition**: Expand or redefine existing rules by incorporating new symbols, concepts, constraints, execution steps, or introducing novel story contexts.
> 2. **Rule-Driven Q&A Design**
>     - **Q&A Adaptation**: Existing questions are adjusted to align with the extracted rules, and both questions and answers are annotated.
>     - **Q&A Generation**: Questions and answers are either manually crafted (e.g., Counterfactual problems where answers differ from real-world facts) or programmatically generated (e.g., Cipher problems).
>     - **Answer Format Specification**: Answers to different questions are assigned specific formats, including NR (Numerical Response), ME (Mathematical Expression), TR (Textual Response), MC (Multiple Choice), and SD (Structured Data).
> 3. **Quality Validation**
>     - **Human Validation**: Human evaluators assess the quality of rules and Q&A pairs.
>     - **LLM Validation**: We evaluate the dataset using LLMs to assess its difficulty and discriminative power. Tasks where models often fail may indicate excessive difficulty or unclear descriptions, while universally correct answers may suggest overly simple setups or data leakage. Throughout the dataset construction process, we repeatedly revise these issues after each evaluation.

---

> ### Author Response · Authors · 2024-11-24
> **Response to reviewer D3fb (Part 3/n)**
>
> ## Response to Weakness 3
>
> We sincerely appreciate your insightful suggestion regarding the evaluation of more advanced prompting strategies.
> 1. **Zero-shot CoT Experiments**: In fact, we have conducted zero-shot CoT experiments by adding 'let's think step by step' to the prompt, but overall, they do not provide significant insights. The results generally exhibit random fluctuations rather than consistent patterns.
> 2. **Anomalous Observation**: We observe a significant drop in the performance of Claude-3.5-Sonnet following the inclusion of the 'Let's think step by step' prompt. This decline appears to be primarily due to a reduction in the model's mathematical capabilities, particularly in the operation category, where accuracy decreases from 88.40% to 58.40%. Specific [prompt](https://anonymous.4open.science/r/kor-bench-rebuttal-repo-44F6/config/prompt/zero-shot-cot.yaml), [responses](https://anonymous.4open.science/r/kor-bench-rebuttal-repo-44F6/results/1121_zero_shot_cot/) and [detailed experimental results](https://anonymous.4open.science/r/kor-bench-rebuttal-repo-44F6/eval/results_1121_zero_shot_cot.csv) can be found in the anonymized repository.
> 1. **Automatic Emergence of CoT in Large Models**: As you mentioned, we observe that simple chains of thought often emerge automatically, particularly in large models, which could be one reason for the lack of patterns. Another possible reason is that KOR-Bench focuses on knowledge that LLMs have not been trained on, and the Zero-Shot CoT prompt does not activate prior knowledge that the model perceives as similar but is actually unrelated. As a result, performance occasionally declines.
> 2. **Further CoT Experiments in Section 6 and Appendix D**:
> We recognize the importance of testing various methods to improve reasoning performance. Since the Zero-Shot CoT approach does not yield significant improvements, we focus our subsequent experiments on a more comprehensive exploration of CoT. These experiments, detailed in Section 6 and Appendix D. For further elaboration, please refer to our response to Weakness 5, where we discuss these experiments in more detail.
>
> | Model Name                        | Zero Shot CoT - Zero Shot |
> |-----------------------------------|---------------------------|
> | GPT-4o                            | -2.64%                   |
> | Qwen2.5-32B-Instruct              | 0.48%                    |
> | Qwen2.5-72B-Instruct              | 0.96%                    |
> | Mistral-Large-Instruct-2407       | -0.08%                   |
> | DeepSeek-V2.5                     | 4.72%                    |
> | Meta-Llama-3.1-70B-Instruct       | 0.64%                    |
> | yi-large                          | -0.48%                   |
> | Qwen2-72B-Instruct                | 2.16%                    |
> | Qwen2.5-7B-Instruct               | 6.40%                    |
> | claude-3-5-sonnet-20240620        | -14.72%                  |
> | Yi-1.5-9B-Chat                    | 4.00%                    |
> | c4ai-command-r-plus-08-2024       | 5.20%                    |
> | Meta-Llama-3.1-8B-Instruct        | -1.36%                   |
> | c4ai-command-r-08-2024            | 3.44%                    |
> | Qwen2-7B-Instruct                 | 1.20%                    |
> | Mistral-7B-Instruct-v0.3          | 1.36%                    |

---

> ### Author Response · Authors · 2024-11-24
> **Response to reviewer D3fb (Part 4/n)**
>
> ## Response to Weakness 4
>
> We appreciate your comment regarding the size of the benchmark. While the current size of KOR-Bench is limited, we are committed to continuously expanding and enhancing it to improve robustness.
> ### Introducing the KOR Concept
> KOR-Bench emphasizes a unique concept, **"Knowledge Orthogonality"**, which sets it apart from other benchmarks by focusing on tasks where rules are embedded within questions rather than being explicitly separated. This design allows for independent evaluation of reasoning processes. Despite its limited size, the benchmark effectively evaluates model reasoning abilities and provides meaningful differentiation.
> ### Current Benchmark's Differentiation Capability
> The current dataset demonstrates differentiation despite its size. For example, O1-Preview and O1-Mini models achieve scores of 72.88 and 70.16, respectively, surpassing other models and confirming the dataset's ability to distinguish performance effectively.
> | Model                   | Overall | Operation | Logic | Cipher | Puzzle | Counterfactual      |
> |--------------------------|---------|-----------|-------|--------|--------|---------------------|
> | O1-Preview (2024-09-12) | 72.88   | 88.8      | 63.2  | 82.8   | 36.8   | 92.80 (5.20)        |
> | O1-Mini (2024-09-12)    | 70.16   | 82.8      | 61.2  | 79.6   | 35.6   | 91.60 (5.60)        |
>
> ### Ablation Study on Dataset Size
> We conducted an ablation study on dataset size, demonstrating that KOR-Bench is robust to size variations. Results are presented in the figure: [Ablation Study on Dataset Size](https://anonymous.4open.science/r/kor-bench-rebuttal-repo-44F6/images/ablation_study_on_dataset_size.pdf).
> - **Metrics**:
>   1. **Accuracy Error Average**: Measures accuracy loss when models are evaluated on smaller subsets.
>   2. **Accuracy Error Standard Deviation**: Captures consistency across different subset sizes.
>   3. **Gini Coefficient**: Assesses score dispersion, reflecting differentiation within subsets.
>     - The calculation formula is: $G = \frac{\sum_{i=1}^{n} \sum_{j=1}^{n} |y_i - y_j|}{2n^2 \bar{y}}$, where $y = [y_1, y_2, \dots, y_n]$ is the model score.
> - **Sampling Strategies**:
>   1. **Rule-Based Sampling**: Samples a fixed proportion of questions under each rule to preserve diversity.
>   2. **Category-Based Sampling**: Randomly samples from all questions in each category to reduce rule-specific difficulty effects.
> - **Results**:
>   - Differences in model scores are minimally affected by dataset size. Accuracy errors average around 2, with standard deviations remaining low. The Gini coefficient varies by approximately 0.02 at most.
>   - Robustness holds across models regardless of size or fine-tuning status. Detailed results are available in the [eval folder](https://anonymous.4open.science/r/kor-bench-rebuttal-repo-44F6/eval).
> ### Plans for Future Expansion
> We plan to:
> - Add more data categories.
> - Increase problem diversity within each rule.
> - Introduce rule variability.
> - Develop a multimodal benchmark version to enhance reasoning evaluation.
>
> ## Response to Weakness 5
>
> Thank you for your feedback. We agree that Section 6, particularly 6.2 and 6.3, requires further clarification to better explain the motivations and analyses.
>
> The primary goal of the experiments in Section 6 is to explore various Chain-of-Thought (CoT) prompting strategies, both to guide the model in solving KOR-Bench problems and to analyze its behavior in depth.
>
> **Experiment 6.1**: Break down cipher-type problem solutions in advance by human experts, guiding the model step by step. This identifies the specific steps where the model struggles.
> **Experiment 6.2**: Guide the model by pointing out its errors, prompting it to reflect on the causes and correct them.
> **Experiment 6.3**: Investigate how the model handles problems requiring longer, more complex reasoning chains and assess its robustness. Specific setups include:
> - **Multi-Q**: Evaluate the model's ability to extend a single rule to multiple instances, testing whether it can consistently apply the rule's logic across various examples.
> - **Multi-R**: Start with a single problem, requiring the model to filter relevant rules and form a decision chain to select the most effective rule for solving.
> - **Multi-RQ**: Encourage the model to analyze problem requirements deeply and select the best combination of rules across problems, forming an integrated reasoning chain.
>
> **Appendix D.2**: Provide strategic tricks for solving puzzles, which are challenging for the model to discover independently. Emphasize strategy over purely step-based solutions.
> **Appendix D.3**: Visualize the model’s attention on specific tokens in the rules to infer activated knowledge during reasoning. Identify attention focus to reveal the model's thought process.

---

> ### Comment · Reviewer_D3fb · 2024-11-26
>
> Thanks for providing a definition of "knowledge orthogonality". It makes intuitive sense. Suggestions:
> - It'd be helpful to define what probability function you refer to as $P$. Does it represent a language model? Does it represent the belief of an idealized agent? I do not think there has to be a correct answer here, but I recommend thinking about which notion of $P$ best captures what you want the formal concept to be.
> - The meaning of $\rightarrow$ needs to be defined. Is $Q \rightarrow A$ supposed to mean "$A$ is the answer to $Q$"?
> - The formula for 2. does not seem to be about "knowledge $K$ supports reasoning". In fact, I do not think you necessarily want "knowledge supports reasoning", right? For example, $K$ will probably adversarially affect performance in the counterfactual task (knowing that Mendel is a pioneer of genetics in the actual world).
>
> In general, I appreciate the detailed responses, and I think the revision improves the paper. I will raise my score to 6. Personally I still think that the "puzzle" and "counterfactual" categories are not strongly knowledge orthogonal. However, that does not have to prevent the benchmark, with its thorough evaluations, to be a potentially useful resource to the community.

---

> ### Author Response · Authors · 2024-11-26
> **Response to reviewer D3fb (Part 5/n)**
>
> We sincerely appreciate your insightful suggestions and constructive feedback. Your comments have been instrumental in prompting us to reassess and refine the definition of Knowledge Orthogonality. Additionally, we are grateful for your decision to raise your score.
>
> ### Refinement of the Formal Definition of Knowledge Orthogonality:
>
> - **Formal Definition:**
>   - **For a task $T$, the required reasoning information consists of**:
>     - $K$: General background/domain-specific knowledge acquired during pre-training, excluding common sense.
>     - $R$: Core rule information designed to solve $T$.
>     - $Q$: A Rule-driven question.
>     - $A$: Answer to the question $Q$.
>
>   - **Notational Definitions:**
>     - $\rightarrow$: Represents the cognitive process of deriving $A$ from $Q$.
>     - $P$: Represents the belief strength that $A$ is a valid answer to $Q$ based on $R$ and/or $K$.
>       - $P(Q \rightarrow A \mid R)$: Belief in $A$ driven solely by $R$.
>       - $P(Q \rightarrow A \mid K)$: Belief in $A$ based solely on $K$.
>       - $P(Q \rightarrow A \mid R, K)$: Combined belief in $A$, integrating $R$ and $K$.
>
>   - **$T$ satisfies knowledge orthogonality under the following conditions:**
>     1. **Knowledge-Rule Decoupling**: Rule $R$ is logically self-contained and independent of $K$.
>        - $R \perp K$
>     2. **Knowledge Assistiveness**: Background knowledge $K$ may support or interfere with the derivation of $A$ from $Q$, but does not play a central role in reasoning. The extent of this influence is quantified by the Knowledge Impact Factor ($\beta$), defined as:
>        - $\beta = \frac{P(Q \rightarrow A \mid R, K) - P(Q \rightarrow A \mid R)}{P(Q \rightarrow A \mid R)}$
>        - $\beta$ ranges from $(−1, \epsilon]$, where ϵ is a very small positive number.
>          - When $\beta$ is positive and close to 0, $K$ has little impact, with $R$ being dominant.
>          - When $\beta$ is negative, it can range from small negative values to approaching -1, where $K$ increasingly undermines reasoning.
>     3. **Rule Centrality**: Correctness relies on understanding and applying $R$, with $R$ having significantly greater influence than $K$.
>        - $P(Q \rightarrow A \mid R, K) \approx P(Q \rightarrow A \mid R) \gg P(Q \rightarrow A \mid K)$
>     4. **Derivation Adjustment**: This formula adjusts the reasoning process based on $R$, incorporating the influence of $K$ with $\beta$ reflecting its effect.
>        - $P(Q \rightarrow A \mid R, K) = P(Q \rightarrow A \mid R) \cdot (1 + \beta)$
>
> ### On the Puzzle and Counterfactual Categories:
> We appreciate your perspective on the potential lack of strong knowledge orthogonality in the "Puzzle" and "Counterfactual" categories. We acknowledge that this is an area that requires further improvement. We will reconsider these two categories and aim to introduce clearer distinctions in the next version by incorporating new puzzle rule definitions and custom worldviews.
>
> Thank you again for your time and thoughtful review.

---

### Meta-Review · Area_Chair_S8Fu · 2024-12-24

**Metareview:**

This paper presents a reasoning benchmark called KOR-Bench, and it aims to include tasks that do not share knowledge with the pre-training data ("knowledge-orthogonal"). There are five categories of tasks, including Operation, Logic, Cipher, Puzzle, and Counterfactual. The authors tested on a range of models and the results show that the near-SOTA models achieve less than 60% accuracy.

Benchmarking reasoning abilities in a reliable way is a very important problem. The reviewers find the paper mostly clear, comprehensive, sound and novel. However, the reviewers also pointed out that there are similar benchmarks like Logicgame (though concurrent), and shared concerns about data scale, lack of clarity on how "knowledge-orthogonal" is guaranteed and the dataset is constructed, lack of benchmark comparison, and lack of prompting methods. The authors subsequently presented additional experiment results and clarifications.

I don't think the "knowledge-orthogonal" guarantee is reliable. It is difficult to define what exactly knowledge is and what the kind of knowledge present in the training data includes. It is possible that certain tasks in KOR-bench were created by others and added to the training data. For example, replacing certain words in a set of rules can be considered orthogonal by the authors, but LLMs may treat the two set of rules as the same in their own mechanisms. Ideally there needs to be a test to prove "knowledge-orthogonal" is indeed true for the proposed benchmark. The authors do not have substantial evidences to prove their tasks are indeed"knowledge-orthogonal" to training data or any data the LLMs have access to. I think the authors should at least add a section discussing this point to avoid overclaiming.

Overall the strengths outweigh the weaknesses. All reviewers agreed to accept this work and I concur.

**Additional Comments On Reviewer Discussion:**

The reviewers find the paper mostly clear, comprehensive, sound and novel. However, the reviewers also pointed out that there are similar benchmarks like Logicgame (though concurrent), and shared concerns about data scale, lack of clarity on how "knowledge-orthogonal" is guaranteed and the dataset is constructed, lack of benchmark comparison, and lack of prompting methods. The authors subsequently presented additional experiment results and clarifications.

Overall the strengths outweigh the weaknesses.

---

### Decision · Program_Chairs · 2025-01-22

Accept (Poster)